# AutoDV: An End-to-End Deep Learning Model for High-Dimensional Data Visualization

**Wei Dai**
Zhejiang University of Technology, Hangzhou, China,
The Chinese University of Hong Kong, Shenzhen, China,
weidai@zjut.edu.cn

**Jicong Fan**[*]
The Chinese University of Hong Kong, Shenzhen, China,
fanjicong@cuhk.edu.cn

## Abstract

High-dimensional data visualization (HDV) plays an important role in data science and engineering applications. Traditional HDV methods, such as Autoencoder and t-SNE, require hyper-parameter tuning and iterative optimization on every dataset and cannot effectively utilize the knowledge from historical low-dimension representation, which lowers the efficiency, convenience, and accuracy in real applications. In this paper, we present AutoDV, an end-to-end deep learning model, for high-dimensional data visualization. AutoDV is built upon a graph transformer network and an invariant loss function and is trained on a number of diverse datasets converted into multi-weight graphs. Given a new dataset, AutoDV outputs the 2D or 3D embeddings of all data points directly. AutoDV has the following merits: 1) There is no hyper-parameter selection during the data visualization stage; 2) The end-to-end model avoids re-training or iterative optimization when visualizing data; 3) The input dataset can have any number of features and can be from any domain. Our experiments show that AutoDV can successfully generalize to unseen datasets without retraining with $89.37\%$ precision of t-SNE and $91.05\%$ precision of UMAP on the unseen CIFAR10 datasets. Compared with existing parametric data visualization deep models, our method obtains a significant improvement with $86.65\%$ precision gain. AutoDV can perform even better than t-SNE and UMAP on gene and UCI tabular datasets. The project is available at https://github.com/DryDew/AutoDV.

## 1 Introduction

Extracting meaningful insights from high-dimensional and complex datasets (Fan, 2025b) to support informed decision-making and drive innovation remains a challenge in contemporary data analysis. High-dimensional data visualization (HDV) is a special dimensionality reduction (DR) technique that allows humans to intuitively interpret large, complex datasets by projecting them into two or three dimensions. This technique has demonstrated success across diverse scientific domains, such as genomics (Dorrity et al., 2020), remote sensing (Li et al., 2022), and finance (Velykoivanenko & Korchynskyi, 2022). HDV is usually an unsupervised task. Formally, given a high-dimensional dataset $\boldsymbol{X} = \{\boldsymbol{x}_1, \boldsymbol{x}_2, \ldots, \boldsymbol{x}_N\}$, where each data point $\boldsymbol{x}_i \in \mathbb{R}^d$, $d$ is the dimensionality of the dataset, and $N$ represents the number of data points, the goal is to find a corresponding low-dimensional representation $\boldsymbol{Z} = \{\boldsymbol{z}_1, \boldsymbol{z}_2, \ldots, \boldsymbol{z}_N\}$, where each $\boldsymbol{z}_i \in \mathbb{R}^{d'}$, typically $d' = 2$ or $d' = 3$, such that the structural relationships in $\boldsymbol{X}$ are preserved in $\boldsymbol{Z}$.

Over the years, numerous algorithms for DR and HDV have been developed, broadly classified into linear and non-linear methods. Classical linear techniques, including Principal Component Analysis (PCA) (Abdi & Williams, 2010), Multidimensional Scaling (MDS) (Saeed et al., 2018),

---

[*]Corresponding author.

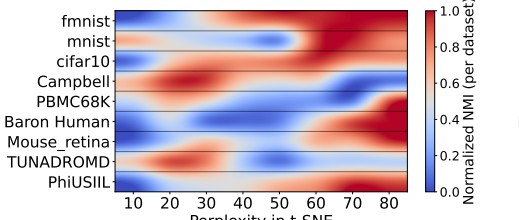 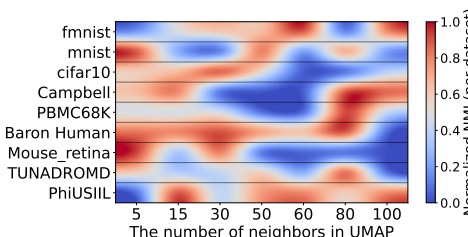

Figure 1: NMI Sensitivity under different hyperparameters for t-SNE (perplexity) and UMAP (n_neighbor). *Note:* The datasets are randomly down-sampled to simulate various real-world unseen datasets. Please see details in Appendix I

and Linear Discriminant Analysis (LDA) (Balakrishnama & Ganapathiraju, 1998), have established foundational approaches but typically fail to uncover complex, non-linear latent structures inherent in the data. Consequently, various non-linear approaches have emerged (Fan et al., 2018), such as Self-Organizing Maps (SOM) (Kohonen, 1982), Isomap (Tenenbaum et al., 2000), Kernel PCA (Schölkopf et al., 1998), Principal Curves (Hastie & Stuetzle, 1989), autoencoders (Wang et al., 2016), Stochastic Neighbor Embedding (SNE) (Hinton & Roweis, 2002), Locally Linear Embedding (LLE) (Roweis & Saul, 2000), and Laplacian Eigenmaps (Belkin & Niyogi, 2003). Despite their advances, these methods are not effective in visualizing real-world data with complex structures. To address this, more sophisticated methods, like t-distributed Stochastic Neighbor Embedding (t-SNE) (Van der Maaten & Hinton, 2008; Sun et al., 2023), Uniform Manifold Approximation and Projection (UMAP) (McInnes et al., 2018), TriMap (Amid & Warmuth, 2019), and PaCMAP (Wang et al., 2021), explicitly optimize low-dimensional embeddings to maintain both local neighborhood structures and global data relationships effectively. Although the traditional methods and recent advances in HDV have facilitated substantial progress in various domains, they face critical limitations in the following.

- **Sensitive to hyper-parameter tuning:** The traditional data visualization methods are sensitive to hyper-parameter selection, such as perplexity in t-SNE and the number of neighbors in UMAP and PaCMAP, also discussed by previous works (Wattenberg et al., 2016; Böhm et al., 2022). Different hyper-parameter selection will significantly change the visualization results. As an empirical experiment presented in Figure 1, a fixed or default hyper-parameter may not always lead to the best performance. The absence of labeled data in unsupervised tasks poses a unique challenge to tuning the HDV hyper-parameter.

- **Re-training overhead:** They need to re-run the training algorithm iteratively, i.e. *re-training*, on every new dataset. It brings significant computational overhead, especially when handling a large number of datasets.

- **Lack of cross-domain and cross-dimension generalization:** Some studies tried to solve re-training overhead using a parametric model by optimizing the following problem: $\min_{\boldsymbol{\theta}} \mathbb{E}_i[\|f_{\boldsymbol{\theta}}(\boldsymbol{x}_i) - \boldsymbol{z}_i\|_2^2]$, where $f : \mathbb{R}^d \rightarrow \mathbb{R}^{d'}$ is the HDV model. They still failed to adapt the model to new datasets with different domains and dimensions and suffered from overfitting to the training datasets. They failed to effectively utilize the historical low-dimensional representations.

To overcome these limitations, we propose AutoDV, a novel end-to-end visualization approach leveraging graph neural networks (GNNs) (Kipf & Welling, 2016) and graph transformers (Rampášek et al., 2022). AutoDV builds a deep learning model using historical visualization results of labeled datasets in a meta-learning manner, effectively capturing the underlying data structure via graph-based representations. Consequently, AutoDV generalizes robustly to unseen datasets without requiring re-training or additional hyper-parameter tuning during inference. Our contributions are summarized as follows:

- We introduce AutoDV, an end-to-end data visualization method that eliminates the need for hyper-parameter tuning and re-training when visualizing new datasets.

- Compared with existing parametric visualization methods such as parametric UMAP and inductive t-SNE, AutoDV can adapt to datasets of varying dimensionality, exhibiting superior generalization performance on unseen datasets from any domain.

- Extensive numerical experiments on real-world datasets demonstrate the effectiveness and superiority of AutoDV.

## 2 RELATED WORKS

As mentioned before, many insightful HDV methods have been proposed in the past decades (Hartono, 2020; Dehghani et al., 2024; Hartono et al., 2014), but they incur high computational costs, particularly when projecting unseen test data. Techniques like Barnes-Hut t-SNE (Van Der Maaten, 2014) and opt-SNE (Belkina et al., 2019) accelerate training but still require re-training on new datasets. Recently, end-to-end visualization models like parametric UMAP (Sainburg et al., 2021) and inductive t-SNE (Roman-Rangel & Marchand-Maillet, 2019) have been developed to avoid re-training overhead. These models use deep neural networks to map high-dimensional data directly to low-dimensional spaces. Auto-encoder (Wang et al., 2016) and its successors, such as Geometric AE (Nazari et al., 2023) and GGAE (Lim et al., 2024), also have the potential to establish an end-to-end HDV model in a self-regression way. However, they still struggle to generalize across datasets with different dimensions or domains. In this paper, we resolve these challenges by utilizing graph neural networks and an affine invariant loss function design.

Another significant challenge in data visualization algorithms is the high sensitivity of hyper-parameter selection (Wattenberg et al., 2016; Böhm et al., 2022). For example, the perplexity in t-SNE critically influences the visualization outcomes. Choosing an excessively large perplexity can result in meaningless spherical visualizations, while an excessively small perplexity might lead to clustering inconsistent with the true data labels, rendering the visualization practically meaningless. Research addressing hyper-parameter optimization in visualization algorithms is still limited. Indeed, tuning hyper-paremters in unsupervised learning is always challenging (Fan et al., 2022). Liao et al. (2023) introduced a Bayesian optimization-based framework to identify optimal hyper-parameters across various visualization performance metrics. However, these metrics rely heavily on ground truth labels, such as classification accuracy, area under the ROC curve, and normalized mutual information (NMI), which are impractical to obtain for large, unlabeled datasets typical in real-world scenarios. Our proposed method does not intend to optimize the hyper-parameters without accessing the ground truth label. Instead, we eliminate the hyper-parameter selection when visualizing new unlabeled datasets to further increase the efficiency.

## 3 PROPOSED METHOD

### 3.1 TASK DEFINITION OF AUTODV

**Definition 1 (AutoDV)** *Suppose we have $L$ labeled historical datasets $(\boldsymbol{X}_1, \boldsymbol{y}_1)$, $(\boldsymbol{X}_2, \boldsymbol{y}_2)$, ..., $(\boldsymbol{X}_L, \boldsymbol{y}_L)$, where $\boldsymbol{X}_i \in \mathbb{R}^{N_i \times d_i}$ is the data matrix with $N_i$ samples and $d_i$ features, and $\boldsymbol{y}_i \in \mathbb{R}^{N_i}$ are the labels, $i = 1, \ldots, L$. Let $\mathcal{A}_\theta$ be an effective data visualization algorithm, with hyper-parameters $\theta$. For each $\boldsymbol{X}_i$, an optimal low-dimensional embedding $\boldsymbol{Z}_i^* \in \mathbb{R}^{N_i \times d'}$ is obtained by $\boldsymbol{Z}_i^* = \mathcal{A}_{\theta_i^*}(\boldsymbol{X}_i)$, where $\theta_i^*$ denotes the optimal hyper-parameters selected using $\boldsymbol{y}_i$ with respect to some metric $\mathcal{M}$. The goal of AutoDV is to train an end-to-end model $f_\phi : \mathbb{R}^{N_i \times d_i} \to \mathbb{R}^{N_i \times d'}$ from $\{(\boldsymbol{X}_i, \boldsymbol{Z}_i^*)\}_{i=1}^L$, such that, for any unseen dataset $\boldsymbol{X}_{new} \in \mathbb{R}^{N_{i'} \times d_{i'}}$, with $\boldsymbol{Z}_{new}^* = \mathcal{A}_{\theta_{i'}^*}(\boldsymbol{X}_{new})$, it holds that*

$$dist(f_\phi(\boldsymbol{X}_{new}), \boldsymbol{Z}_{new}^*) \leq \varepsilon \tag{1}$$

*where $dist(\cdot, \cdot)$ denotes some distance metric and $\varepsilon > 0$ is a small constant.*

In Definition 1, examples of $\mathcal{A}_\theta$ include t-SNE and UMAP. The evaluation metric $\mathcal{M}$ could be the normalized mutual information (NMI), which is a widely used index in clustering and dimensionality reduction. $dist(\cdot, \cdot)$ could be the Frobenius norm. In most real scenarios, we hope to use HDV methods to observe the potential cluster structures in the data and it is not difficult to achieve a large number of labeled datasets from diverse domains to train our model $f_\phi$. That's why we use labeled historical datasets to establish AutoDV.

Nevertheless, there are two challenges when solving problem 1.

C1 Designing the end-to-end model $f_\phi$ is non-trivial, especially when the model is expected to handle input datasets with different sizes and dimensions, i,e., $N_i \neq N_j$ and $d_i \neq d_j$ for some $i$ and $j$.

C2 There exist multiple optimal low-dimension embeddings $\boldsymbol{Z}_i^*$ for the same input $\boldsymbol{X}_i$ if we apply translation, rotation, and scaling to $\boldsymbol{Z}_i^*$, e.g., $\boldsymbol{Z}_i^*$ and $\boldsymbol{Z}_i^* \boldsymbol{Q}$ with any orthonormal matrix $\boldsymbol{Q}$ are equivalent in terms of visualization. This will result in a one-to-many problem and will be difficult to converge (Arridge et al., 2019).

To tackle these two challenges, we present a graph neural network (GNN) based model design and an affine invariant loss design in the following, respectively.

## 3.2 AutoDV Model Design

Most deep neural networks are designed for a fixed input dimensionality and therefore cannot be directly applied to datasets with different numbers of features, which is a main limitation of existing parametric data visualization models. To address this issue, AutoDV additionally exploits the graph structure of each dataset. Given a historical dataset $\boldsymbol{X}_i$, we first construct a weighted graph whose adjacency matrix is a pairwise similarity matrix between samples. The similarity matrix is computed using a Gaussian kernel function (Wasserman, 2006). To preserve as much structural information as possible, we further generate *k-scale graphs* for each dataset by using different bandwidth parameters in the Gaussian kernel, following (Long et al., 2015). The resulting graphs, $\boldsymbol{S}_i^{(1)}, \boldsymbol{S}_i^{(2)}, \ldots, \boldsymbol{S}_i^{(k)}$, capture the data structure at different scales and are computed as follows.

$$\boldsymbol{S}_i^{(j)}[u, v] = \exp\left( -\frac{\|\boldsymbol{X}_i[u] - \boldsymbol{X}_i[v]\|_2^2}{\gamma^{(j)}} \right), \ \ \forall j = 1, 2, \ldots, k, \tag{2}$$

where $\boldsymbol{S}_i^{(j)}[u, v]$ denote the entry in the $u$-th row and $v$-th column of $\boldsymbol{S}_i^{(j)}$, $\|\boldsymbol{X}_i[u] - \boldsymbol{X}_i[v]\|_2^2$ represents the squared Euclidean distance between $u$-th data point and the $v$-th data point in dataset $\boldsymbol{X}_i$, and $\gamma^{(j)}$ is the bandwidth parameter of the $j$-th scale graph. Consequently, a dataset in $\mathbb{R}^{N_i \times d_i}$ is transformed into multiple graphs represented by weighted adjacency matrices in $\mathbb{R}^{N_i \times N_i}$. This graph representation mitigates the constraints imposed by varying dataset dimensions on end-to-end visualization methods.

We employ Graph Neural Networks (GNNs) (Kipf & Welling, 2016) to extract intrinsic structural features from the input graphs. GNNs, extensively studied in graph learning literature (Wu et al., 2020), leverage message-passing and node-weight sharing mechanisms to handle graphs with varying node counts effectively. Unlike typical graph learning tasks such as node classification (Yifan et al., 2020), graph classification (Wang & Fan, 2024), and graph clustering (Sun & Fan, 2024; Fan, 2025a), our input graphs do not inherently include node features, while GNNs often require node features for node aggregation. To solve this, we derive graph positional encodings (PE) (Grötschla et al., 2024) from the weighted adjacency matrices as node features, utilizing techniques such as Laplacian positional encoding (Belkin & Niyogi, 2003) or random walk encoding (Perozzi et al., 2014). We denote the positional encoding as $\boldsymbol{P}_i \in \mathbb{R}^{N_i \times d_e}$, where $d_e$ is the dimensionality of the positional encoding. Then, for each dataset $\boldsymbol{X}_i$, we have

$$\boldsymbol{P}_i^{(i)} = h(\boldsymbol{S}_i^{(j)}), \ \ \forall j = 1, \ldots, k \tag{3}$$

where $h(\cdot)$ is the function for extracting the positional encoding. In our experiment, we adopt singular value decomposition (SVD) positional encoding (Sium et al., 2023).

Our end-to-end AutoDV model extensively utilizes the popular GNN architectures, GIN (Xu et al., 2018) and Graph Transformer (Rampášek et al., 2022), as the model backbone. To utilize the $k$-scale graphs spawned from Eq. (2), we propose a multi-graph GNN. For each graph, there is an individual GNN to process it. Then, we have $k$ sets of node embeddings extracted by GNNs. By concatenating them together, we get the final hidden node embedding for the input dataset. Finally, a graph transformer is introduced to explore the inter-node relationship, and the final output of the model is produced by a multi-layer perceptron (MLP). The whole AutoDV framework is illustrated in Figure 2. It processes an input dataset $\boldsymbol{X}_i$ by first computing $k$ similarity matrices $\{\boldsymbol{S}_i^{(j)}\}_{j=1}^k$ and extracting graph positional encodings $\{\boldsymbol{P}_i^{(j)}\}_{j=1}^k$. Then, $(\boldsymbol{P}_i^{(j)}, \boldsymbol{S}_i^{(j)}), j = 1, 2, \ldots, k$, serve as inputs to the GINs, which perform feature extraction. The $k$ outputs are concatenated as new node

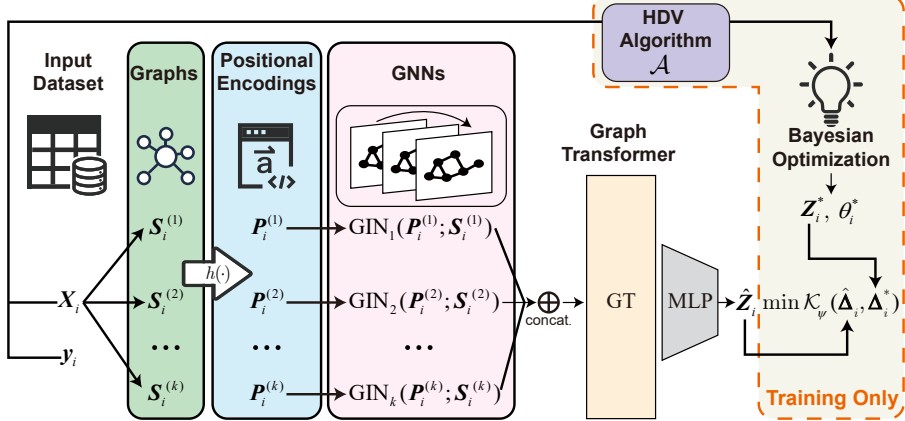

Figure 2: AutoDV framework. The input datasets will first be converted into $k$ graphs. Then, positional encodings are extracted from the graph. Each graph is processed by an independent GIN network. The output node features of GINs are concatenated and sent to a graph transformer. Finally, an MLP head is responsible for outputting the low-dimensional embedding. Ground truth label $\boldsymbol{y}_i$, HDV algorithm $\mathcal{A}$, and Bayesian optimization are only used at the training stage to produce $\boldsymbol{Z}_i^*$.

features and are fed into a graph transformer and an MLP to output a low-dimensional visualization result $\hat{\boldsymbol{Z}}_i$. The AutoDV model $f_\phi$ is expressed as follows:

$$
f_\phi \left( \left\{ (\boldsymbol{P}_i^{(j)}, \boldsymbol{S}_i^{(j)}) \right\}_{j=1}^k \right) = \mathrm{MLP} \left( \mathrm{GT} \left( \mathrm{GIN}_1(\boldsymbol{P}_i^{(1)}, \boldsymbol{S}_i^{(1)}) \oplus \cdots \oplus \mathrm{GIN}_k(\boldsymbol{P}_i^{(k)}, \boldsymbol{S}_i^{(k)}) \right) \right), \quad (4)
$$

where $\oplus$ represents the concatenation along the feature dimension, $\mathrm{GIN}_j(\cdot)$ is the $j$-th GIN network, $j = 1, 2, \ldots, k$, and $\mathrm{GT}(\cdot)$ is the graph transformer. Hence, we let $f_\phi$ to be a GNN model, $\mathbb{R}^{k \times N_i \times d_e} \to \mathbb{R}^{N_i \times d'}$, parametrized by $\phi$ accepting unified dimensionality of node features.

### 3.3 MODEL TRAINING

**Affine Invariant Loss Function**  To tackle the second challenge in problem 1, we introduce an affine invariant loss function for the AutoDV task. Instead of direct alignment between the output low-dimensional embeddings, we align the geometric structure between the low-dimensional embeddings. Specifically, we construct pairwise squared similarity matrices $\hat{\boldsymbol{\Delta}}_i$ and $\boldsymbol{\Delta}_i^*$ for $\boldsymbol{Z}_i$ and $\boldsymbol{Z}_i^*$, i.e.,

$$
\hat{\boldsymbol{\Delta}}_i[u, v] = \sigma \left( \|\hat{\boldsymbol{Z}}_i[u] - \hat{\boldsymbol{Z}}_i[v]\|_2^2 \right), \qquad \boldsymbol{\Delta}_i^*[u, v] = \sigma \left( \|\boldsymbol{Z}_i^*[u] - \boldsymbol{Z}_i^*[v]\|_2^2 \right), \quad (5)
$$

where $\sigma(\cdot)$ is a transformation function turning a distance to a similarity. Then we present a general training loss function as follows:

$$
\mathcal{L} = \frac{1}{L} \sum_{i=1}^{L} \sum_{u=1}^{N_i} \sum_{v=1}^{N_i} \mathcal{K}_\psi(\hat{\boldsymbol{\Delta}}_i[u, v], \boldsymbol{\Delta}_i^*[u, v]), \quad (6)
$$

where $\mathcal{K}_\psi(\cdot, \cdot)$ denotes the Bregman divergence (Bregman, 1967) with a strictly convex function $\psi$ and $\mathcal{K}_\psi(x, y) = \psi(x) - \psi(y) - \langle \nabla \psi(y), x - y \rangle$ for any two scalers $x$ and $y$. By aligning the pairwise distance matrices, the one-to-many problem brought by translation and rotation is eliminated. To further realize the scaling invariant, we pre-process $\boldsymbol{Z}_i^*$ using $z$-score normalization. Then, the training loss function is an affine invariant function that effectively solves the one-to-many problem.

For the choice of $\sigma$ and $\psi$, it depends on the selection of $\mathcal{A}$ spawning $\boldsymbol{Z}^*$. In this paper, we consider t-SNE and UMAP since they are the most popular HDV methods currently, although other HDV methods can be used in our AutoDV. For t-SNE, we use Kullback-Leibler divergence (KLD) loss

similar to the original t-SNE training process as follows:

$$\mathcal{L}_{\text{tsne}} = \frac{1}{L} \sum_{i=1}^{L} \sum_{u=1}^{N_i} \sum_{v=1}^{N_i} p_{u,v}^{(i)} \log \left( \frac{p_{u,v}^{(i)}}{q_{u,v}^{(i)}} \right),$$

$$p_{u,v}^{(i)} = \frac{(1 + \|\boldsymbol{Z}_i^*[u] - \boldsymbol{Z}_i^*[v]\|_2^2)^{-1}}{\sum_{u',v'}(1 + \|\boldsymbol{Z}_i^*[u'] - \boldsymbol{Z}_i^*[v']\|_2^2)^{-1}}, \quad q_{u,v}^{(i)} = \frac{(1 + \|\hat{\boldsymbol{Z}}_i[u] - \hat{\boldsymbol{Z}}_i[v]\|_2^2)^{-1}}{\sum_{u',v'}(1 + \|\hat{\boldsymbol{Z}}_i[u'] - \hat{\boldsymbol{Z}}_i[v']\|_2^2)^{-1}}.$$

(7)

Compared with t-SNE, Eq. 7 does not have access to the original high-dimensional data, avoiding the selection of perplexity. It guides the model $f_\phi$ to mimic the optimal low-dimensional embedding from t-SNE using KLD to ensure the outputs and ground truth $\boldsymbol{Z}^*$ are from the same distribution.

As for $\boldsymbol{Z}^*$ coming from UMAP, we consider both local structure and global structure consistency between $\hat{\boldsymbol{Z}}$ and $\boldsymbol{Z}^*$ since the original UMAP algorithm focuses more on the local structure using "n_neighbors" hyper-parameter, leading to a sparse similarity matrix in the original data space. We present the training loss for UMAP as follows:

$$\mathcal{L}_{\text{umap}} = \frac{1}{L} \sum_{i=1}^{L} \left( \sum_{(u,v)\in\mathcal{N}_i} \ell_{(u,v)} + \lambda(t) \sum_{(u,v)\notin\mathcal{N}_i} \ell_{(u,v)} \right),$$

(8)

where $\ell_{(u,v)} = \left( \frac{1}{1+a\|\hat{\boldsymbol{Z}}_i[u'] - \hat{\boldsymbol{Z}}_i[v']\|_2^{2b}} - \frac{1}{1+a\|\boldsymbol{Z}_i^*[u'] - \boldsymbol{Z}_i^*[v']\|_2^{2b}} \right)^2$, $a$ and $b$ are two hyper-parameters mapping the distance value into a family of student-t distribution, $\lambda(t)$ is a regularization coefficient decreasing as the training step $t$ growing, $\mathcal{N}_i$ is the index set of the k-nearest pairs in dataset $\boldsymbol{X}_i$, and the number of neighbors is determined by $\boldsymbol{Z}^*$ when searching the optimal hyper-parameters, i.e. "n_neighbors". We set $a = 1.93$ and $b = 0.79$, the same as the default hyper-parameters of UMAP. We linearly decay the $\lambda(t)$, i.e., $\lambda(t) = (1 - t/T)$, where $T$ is the maximum number of iterations. This term makes sure that the model focuses more on the local structure consistency. We show that Eq. (7) and Eq. (8) are special cases of Eq. (6) in Appendix C.

**Eliminating Sign Ambiguity in PE** In the proposed method, positional embeddings (PE) are derived using Eq. (3) via spectral techniques such as SVD or eigenvalue decomposition. A well-known challenge with these methods is sign ambiguity (Lim et al., 2022). If $v$ is an eigenvector of $\boldsymbol{S}$, then $-v$ is also an eigenvector. This inherent ambiguity can cause structurally similar graphs to produce significantly different positional embeddings simply due to inconsistent eigenvector signs, which is clearly unreasonable. However, there is currently no principled way to resolve this ambiguity or determine an optimal sign convention. To address the sign ambiguity, we propose a sign count-based method to decide whether to flip the sign of each dimension of PE or not. Specifically, if the majority value in one column of $\boldsymbol{P}$ is negative, we flip the sign of this column. Otherwise, we keep the sign. The process is illustrated in Appendix D. Although the proposed sign flipping strategy may not completely solve the sign ambiguity problem, it is lightweight to implement and shows good performance in our empirical experiments.

## 3.4 COMPUTATIONAL COMPLEXITY ANALYSIS AND LARGE-SCALE EXTENSION

As shown by Table 1, AutoDV is more efficient than t-SNE and UMAP because it does not require iterative optimization on a new dataset. More details are in Appendix F due to limited space.

Table 1: Computation complexity comparison for effectively visualizing a new dataset.

| HDV Method | AutoDV | t-SNE | UMAP |
|---|---|---|---|
| Complexity | $\mathcal{O}(N_i^2 w_{\max})$ | $\mathcal{O}(N_i^2 BT)$ | $\mathcal{O}(N_i^2 d_i + N_i r BT)$ |

As shown by the table, the quadratic complexity in terms of $N_i$ hinders the application of AutoDV to very large datasets. One can use sparsification or other techniques to accelerate the computation of self-attention and graph convolution. We propose a batch-based method to allow AutoDV scale to datasets with large sizes at the inference stage. Specifically, we split $\boldsymbol{X}_i$ into $\{\boldsymbol{X}_i^{(1)}, \boldsymbol{X}_i^{(2)}, \ldots, \boldsymbol{X}_i^{(q_i)}\}$, where each subset is in $\mathbb{R}^{M \times d_i}$. Then, we construct an anchor subset $\boldsymbol{A}$

with a size smaller than $M$, where the intra-distance among points is small, i.e., points in $\boldsymbol{A}$ belong to the same cluster. We simultaneously input $\boldsymbol{A} \cup \boldsymbol{X}_i^{(j)}$, $j = 1, 2, \ldots, q_i$, for each $j$. The final low-dimensional embedding is the corresponding $\hat{\boldsymbol{Z}}_i^{(1)} \cup \cdots \cup \hat{\boldsymbol{Z}}_i^{(q_i)}$. We utilize a fixed anchor set to calibrate $\hat{\boldsymbol{Z}}_i^{(j)}$ from different outputs. The details are shown in Appendix E.

## 3.5 THEORETICAL ROBUSTNESS ANALYSIS FOR AUTODV

In this section, we analyze the theoretical robustness of AutoDV. It is to test how stable its outputs are when the input data contains noise, outliers, or small perturbations. A robust model will produce very similar low-dimensional visualizations even from slightly corrupted data, ensuring its results are reliable. This is crucial because real-world data is often messy.

**Theorem 1** *Given a dataset $\boldsymbol{X}$ with $n$ samples, let $\{\boldsymbol{P}^{(j)}, \boldsymbol{S}^{(j)}\}_{j=1}^k$ with $\boldsymbol{P}^{(j)} \in \mathbb{R}^{n \times d_e}$ be the input of AutoDV model $f_\phi$, where the output is $\boldsymbol{Z} = f_\phi(\{\boldsymbol{P}^{(j)}, \boldsymbol{S}^{(j)}\}_{j=1}^k)$. Denote $\tilde{\boldsymbol{X}}$ as a perturbed counterpart of $\boldsymbol{X}$, where the output is $\tilde{\boldsymbol{Z}} = f_\phi(\{\tilde{\boldsymbol{P}}^{(j)}, \tilde{\boldsymbol{S}}^{(j)}\}_{j=1}^k)$. Suppose $f_\phi$ is $L_\phi$-Lipschitz continuous, then*

$$\|\boldsymbol{Z} - \tilde{\boldsymbol{Z}}\|_F \le 2k L_\phi \sqrt{\frac{2n}{e}} \max_j \left( \frac{c^{(j)} + 1}{\sqrt{\gamma^{(j)}}} \right) \|\boldsymbol{X} - \tilde{\boldsymbol{X}}\|_F, \tag{9}$$

*where $c^{(j)} = \frac{2\sqrt{2\lambda_{\max}^{(j)}}}{\delta^{(j)}} + \frac{1}{2} \sqrt{\frac{d_e}{\lambda_{\min}^{(j)}}}$, $\delta^{(j)}$ is the eigen gap between $\boldsymbol{S}^{(j)}$ and $\tilde{\boldsymbol{S}}^{(j)}$, $\lambda_{\max}^{(j)}$ is the maximum eigenvalue of $\boldsymbol{S}^{(j)}$, and $\lambda_{\min}^{(j)}$ is the minimum the $d_e$-th eigenvalue between $\boldsymbol{S}^{(j)}$ and $\tilde{\boldsymbol{S}}^{(j)}$.*

The Lipschitz constant $L_\phi$ is determined by the network structure and is usually $\mathcal{O}(\prod_{i=1}^D \|\mathbf{W}_i\|_2)$, where $D$ denotes the maximum deepth of the neural networks and $\mathbf{W}_i$ denotes the weight matrix of layer $i$. As shown by the theorem, proved in Appendix B, if $\delta^{(j)}$ is large, $d_e / \lambda_{min}^{(j)}$ is small, and $\gamma^{(j)}$ is large, our AutoDV is stable, provided that the neural network is not too complex. The result also indicates that, if a new dataset is similar to a training dataset, the visualizations should be similar too. This guarantees the generalization ability of AutoDV to some extent.

## 4 EXPERIMENTS

### 4.1 EXPERIMENTAL SETTINGS

**Datasets** We consider 3 types of datasets from the real world. 1. **Image Data:** We visualize image datasets via CLIP-extracted features (Radford et al., 2021), a common practice in deep learning (Hohman et al., 2018; Huff et al., 2021). We use MNIST (Deng, 2012), Fashion-MNIST (Xiao et al., 2017), and CIFAR-10 (Krizhevsky et al., 2009); each image is mapped to a 512-dim vector. MNIST and FMNIST are used for training, CIFAR-10 for testing. 2. **Gene Data:** We utilize four commonly used gene datasets in bioinformatics research. Baron Human (Baron et al., 2016), Mouse Retina[1], Campbell (Campbell et al., 2017), and PBMC68k[2]. We set Mouse Retina as the testing set and the rest for training. We drop out the rare cells following (Wang et al., 2024) to ensure the class balance. 3. **Tabular Data:** [3]We collect 26 real-world tabular datasets from UCI Machine Learning Library (Kelly et al.). These tabular datasets are labeled for classification tasks. Categorical features are one-hot encoded. We drop the auto-increment attributes, such as date, time, and ID. We randomly split 30% for the testing. The detailed information of the datasets is summarized in Appendix G.

**Baselines and Evaluation Metrics** We compare the proposed AutoDV with two existing deep learning based parametric visualization methods, including parametric UMAP (**p-UMAP**)(Sainburg

---

[1]https://www.ncbi.nlm.nih.gov/geo/query/acc.cgi?acc=GSE201402

[2]https://www.10xgenomics.com/datasets/fresh-68-k-pbm-cs-donor-a-1-standard-1-1-0

[3]Although the gene data also belongs to tabular data, we in this work treat them separately because almost all papers on gene data analysis use HDV, and gene data are often in higher dimensionality with higher sparsity than daily life tabular data.

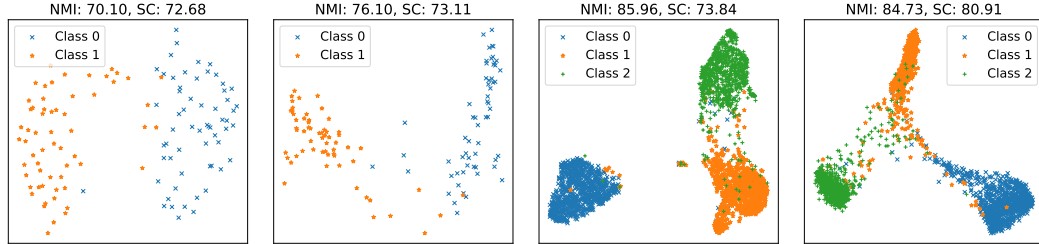

(a) UMAP* on CIFAR10 with 120 points. (b) AutoDV-UMAP on CIFAR10 with 120 points. (c) UMAP* on CIFAR10 with 2759 points. (d) AutoDV-UMAP on CIFAR10 with 2759 points.

Figure 3: (a) and (b) are results of UMAP* and AutoDV-UMAP on the same small test subset of CIFAR10, where AutoDV is better. (c) and (d) are results of UMAP* and AutoDV-UMAP on the same large test subset of CIFAR10.

et al., 2021)[4] and inductive TSNE (**i-tSNE**) (Roman-Rangel & Marchand-Maillet, 2019)[5]. We manually extend i-tSNE to UMAP as inductive UMAP (**i-UMAP**) for fair comparison. We also compare the proposed method with **PCA** (Abdi & Williams, 2010) and autoencoder (**AE**) (Wang et al., 2016). We also report the performance of the optimal low-dimensional embedding from t-SNE and UMAP searched by Bayesian optimization as the ground truth performance of our method, denoted as **t-SNE**\* and **UMAP**\*, respectively. The default (D.) t-SNE and UMAP are also reported. We evaluate the output low-dimensional embeddings using **NMI** with ground-truth labels and the silhouette coefficient (**SC**) (Rousseeuw, 1987), following (Qiao et al., 2025), detailed in Appendix H. To validate whether our method can finish the AutoDV task in problem 1, we also report the relative precision of NMI and SC, which is calculated as $\mathcal{M}(\hat{Z}; y)/\mathcal{M}(Z^*; y)$. $\mathcal{M}$ is NMI or SC. Note that the precision can be larger than 1 since the performance of $\hat{Z}$ can be better.

**Training Data Preparation** Due to the limited video memory size, directly inputting a graph with a large number of nodes is infeasible. To mitigate this, we randomly sample many subsets (sub-graphs) from a very large dataset (graph) similar to (Zeng et al., 2020). The detailed sampling process is in Appendix I. The maximum size of the subset is set to 3000. For each subset, we use Bayesian optimization (Jones et al., 1998) and its ground truth label to search for the optimal hyper-parameter and the corresponding low-dimensional embedding $Z^*$ setting $\mathcal{M}$ to NMI. Note that AutoDV is not limited by the choice of $\mathcal{M}$, and can accommodate various metrics for HDV depending on specific analytical objectives. We adopt NMI and SC, as they reveal cluster structures that are more interpretable and perceptually aligned with human intuition in high-dimensional spaces. Finally, we drop out the subset with NMI less than $10\%$ to ensure the quality of the training data. For t-SNE, we search for the optimal perplexity. For UMAP, we search for the optimal "n_neighbors". Detailed implementation of our method is in Appendix J.

## 4.2 RESULTS

**Comparative Study with Baselines** In the image data experiment, the feature dimensionality of training data and testing data is the same. So, we can compare the performance of our method and the baseline methods. We randomly sample 500 subsets from MNIST and FMNIST for training, and sample 50 subsets from CIFAR10 for testing. The comparison results are reported in Table 2. It is seen that almost no baseline method can produce an effective low-dimensional embedding for unseen new datasets, indicating poor generalizability. For i-tSNE and i-UMAP, due to the one-to-many problem, they are even underfitting on the training sets. In contrast, our method demonstrates superior performance on unseen new datasets using historical results from t-SNE or UMAP.

**Cross-Dimension Results on Gene and Tabular Data** Since the dimensionalities of the training datasets and the test datasets are different, baseline methods cannot directly perform training. For

---

[4]https://github.com/fcarli/parametric_umap
[5]https://github.com/AlexMour92/inductive-tsne

Table 2: Comparative data visualization results on image datasets. Average NMI, NMI precision, SC, and SC precision are reported on the test set. 500 subsets of MNIST and FMNIST are sampled for training. 50 subsets of CIFAR-10 are sampled for testing. The best performance on test sets is **bold**. Selected visualization results are plotted in Figure 3.

| Methods | t-SNE* | UMAP* | D. t-SNE | D. UMAP | PCA | AE | p-UMAP | i-tSNE | i-UMAP | AutoDV-tSNE | AutoDV-UMAP |
|---|---|---|---|---|---|---|---|---|---|---|---|
| Test NMI | $77.04_{\pm8.8}$ | $80.45_{\pm6.6}$ | $71.71_{\pm13.5}$ | $79.72_{\pm6.7}$ | $15.97_{\pm6.9}$ | $6.08_{\pm3.8}$ | $15.58_{\pm7.6}$ | $5.38_{\pm2.1}$ | $3.63_{\pm1.7}$ | $68.70_{\pm9.0}$ | $\mathbf{73.28_{\pm7.6}}$ |
| Test NMI Prec. | 100 | 100 | - | - | - | - | $18.54_{\pm5.4}$ | $10.41_{\pm25.4}$ | $4.40_{\pm2.1}$ | $89.37_{\pm7.8}$ | $\mathbf{91.05_{\pm5.3}}$ |
| Test SC | $63.32_{\pm9.5}$ | $68.54_{\pm6.6}$ | $47.24_{\pm9.9}$ | $67.61_{\pm6.79}$ | $34.51_{\pm2.5}$ | $42.85_{\pm8.7}$ | $43.84_{\pm8.5}$ | $39.34_{\pm5.2}$ | $40.50_{\pm6.5}$ | $55.15_{\pm6.6}$ | $\mathbf{70.41_{\pm7.1}}$ |
| Test SC Prec. | 100 | 100 | - | - | - | - | $62.63_{\pm7.2}$ | $62.65_{\pm8.6}$ | $58.10_{\pm11.1}$ | $88.60_{\pm14.4}$ | $\mathbf{103.0_{\pm8.6}}$ |

Table 3: Data visualization results on gene datasets. Average NMI, NMI precision, SC, and SC precision are reported on the test set. 2918 subsets of Baron Human, Campbel, and PBMC68k are sampled for training. 113 subsets of the Mouse Retina dataset are sampled for testing. The best performance on test sets is **bold**.

| Methods | t-SNE* | UMAP* | D. t-SNE | D. UMAP | PCA | AE | p-UMAP | i-tSNE | i-UMAP | AutoDV-tSNE | AutoDV-UMAP |
|---|---|---|---|---|---|---|---|---|---|---|---|
| Test NMI | $32.73_{\pm30.3}$ | $28.85_{\pm34.7}$ | $15.67_{\pm21.9}$ | $22.45_{\pm33.2}$ | $9.27_{\pm5.0}$ | $5.21_{\pm5.9}$ | $24.78_{\pm15.1}$ | $4.80_{\pm3.5}$ | $13.98_{\pm7.9}$ | $\mathbf{33.22_{\pm28.7}}$ | $33.03_{\pm25.0}$ |
| Test NMI Prec. | 100 | 100 | - | - | - | - | $93.38_{\pm72.9}$ | $35.42_{\pm29.7}$ | $52.50_{\pm61.1}$ | $102.7_{\pm36.7}$ | $\mathbf{111.9_{\pm60.2}}$ |
| Test SC | $34.43_{\pm4.6}$ | $47.98_{\pm20.1}$ | $35.84_{\pm13.3}$ | $57.00_{\pm24.9}$ | $34.10_{\pm7.6}$ | $\mathbf{73.18_{\pm27.3}}$ | $65.58_{\pm15.3}$ | $54.70_{\pm1.6}$ | $61.51_{\pm6.0}$ | $47.11_{\pm10.8}$ | $47.98_{\pm12.2}$ |
| Test SC Prec. | 100 | 100 | - | - | - | - | $151.5_{\pm53.8}$ | $\mathbf{161.1_{\pm17.8}}$ | $160.3_{\pm26.2}$ | $110.4_{\pm30.1}$ | $112.5_{\pm22.1}$ |

these methods, we randomly project the input data into 512-dimension space. In the gene data experiment, we sampled 2918 subsets of Baron Human, Campbel, and PBMC68k for training and sampled 113 subsets of Mouse Retina for testing. All subsets have optimal NMIs higher than $10\%$. The results are reported in Table 3. In the tabular data experiment, we sampled 519 subsets in total from 31 real-world tabular datasets. We split $30\%$, 156 subsets, for testing. The results are reported in Table 4. In the results, we see that AutoDV has the best NMI on the test sets, and are even better than the optimal low-dimensional embeddings from the original t-SNE and UMAP. It indicates our methods can better capture the structural information from the high-dimensional space. We observe that deep-learning baselines often yield high SC but very low NMI under our setting, suggesting the output representation collapses into a dense region due to optimization difficulties.

**Cross-Domain Transferability of AutoDV** To further validate the generalizability of the proposed AutoDV model, we conducted cross-domain experiments across diverse data types. Figure 4 summarizes the average NMI, with source domains represented along the columns and target domains along the rows. The results of SC are given in Appendix K. AutoDV exhibits strong transferability, especially when transferring from image or tabular domains to the gene domain. It surpasses even the optimal within-domain embeddings produced by t-SNE* and UMAP*. Transfers into the image domain result in only a slight reduction in performance, which may be attributed to the higher structural complexity or noise inherent in gene and tabular data.

## 4.3 RUNNING TIME COMPARISON

We compare the wall-clock running time among t-SNE, UMAP, AutoDV, and AutoDV with pre-computed PEs when visualizing a new dataset in $\mathbb{R}^{3000 \times 512}$ using 1 CPU core. The number of optimization iterations of t-SNE and UMAP are set to 2000. All runs are repeated 10 times. The results are in Table 5. It is seen that AutoDV obtains significant acceleration compared with t-SNE. Although UMAP also consumes less time, it suffers from the hyper-parameter selection involving multiple retrainings, which results in unacceptable overhead.

Table 4: Data visualization results on UCI tabular datasets. Average NMI, NMI precision, SC, and SC precision are reported on the test set. 519 subsets are sampled from 26 datasets, of which 156 subsets are split for testing. The best performance on test sets is **bold**.

| Methods | t-SNE* | UMAP* | D. t-SNE | D. UMAP | PCA | AE | p-UMAP | i-tSNE | i-UMAP | AutoDV-tSNE | AutoDV-UMAP |
|---|---|---|---|---|---|---|---|---|---|---|---|
| Test NMI | $30.92_{\pm12.2}$ | $24.80_{\pm16.2}$ | $23.53_{\pm11.2}$ | $15.13_{\pm12.1}$ | $10.96_{\pm11.56}$ | $5.17_{\pm11.0}$ | $14.11_{\pm11.8}$ | $8.00_{\pm10.6}$ | $10.56_{\pm12.6}$ | $33.45_{\pm21.6}$ | $\mathbf{35.15_{\pm34.5}}$ |
| Test NMI Prec. | 100 | 100 | - | - | - | - | $57.08_{\pm30.8}$ | $27.96_{\pm32.2}$ | $50.89_{\pm51.7}$ | $121.3_{\pm40.3}$ | $\mathbf{129.0_{\pm93.2}}$ |
| Test SC | $44.42_{\pm10.0}$ | $64.46_{\pm15.9}$ | $43.87_{\pm10.6}$ | $61.81_{\pm19.1}$ | $63.87_{\pm25.46}$ | $75.4_{\pm6.6}$ | $75.26_{\pm9.7}$ | $\mathbf{77.11_{\pm16.9}}$ | $76.68_{\pm16.5}$ | $48.25_{\pm10.7}$ | $64.28_{\pm9.7}$ |
| Test SC Prec. | 100 | 100 | - | - | - | - | $171.2_{\pm22.0}$ | $178.1_{\pm44.2}$ | $\mathbf{179.7_{\pm40.9}}$ | $110.0_{\pm12.5}$ | $106.4_{\pm32.3}$ |

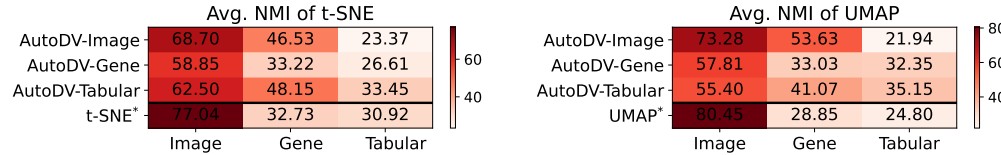

Figure 4: Cross-domain transferability performance using t-SNE and UMAP. Each heatmap shows average NMI results for (left) t-SNE and (right) UMAP. Within each map, columns denote the source domain and rows the target domain for the first three rows; the fourth row ("TSNE*" or "UMAP*") reports the within-domain performance of the optimal low-dimensional embedding. Test sets for each domain are the same as in Tables 2, 3, and 4.

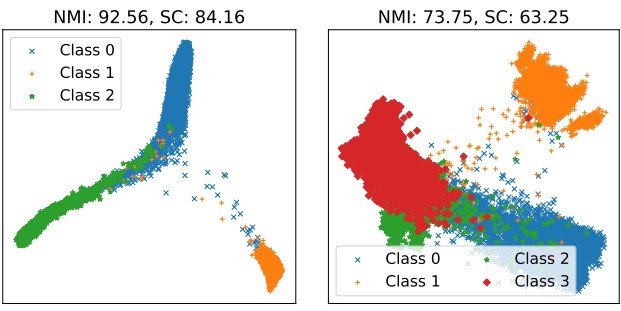

(a) AutoDV-UMAP on CIFAR10 with 10000 points. (b) AutoDV-UMAP on CIFAR10 with 20000 points.

Figure 5: (a) is the result of extending AutoDV to a dataset with 10000 points by direct input. (b) is the result of extending AutoDV to a dataset with 20000 points by the batch-based method in 3.4.

Table 5: Running time comparison for visualizing a dataset.

| HDV | AutoDV | AutoDV (precomputed PE) | t-SNE | UMAP |
|---|---|---|---|---|
| Time (s) | 101.71±10.1 | 92.67±6.2 | 763.30±7.01 | 103.32±9.63 |

## 4.4 EXTENSION TO LARGE DATASET AND MORE RESULTS

We present results of two ways to visualize large datasets in Figure 6a and 6b. We show more results and an ablation study in Appendix L.

## 5 CONCLUSION

This paper proposed a new end-to-end parametric HDV method, called AutoDV. AutoDV avoids hyper-parameter selection and retraining when visualizing new datasets. Furthermore, it can be generalized to datasets with any number of features from any domain. Experiments on three types of datasets demonstrate superior performance in terms of NMI and SC.

**Limitations** Although our proposed AutoDV tackles two challenges when designing parametric HDV models and avoids the hyper-parameter selection at the inference stage, it still has some limitations, such as historical dataset dependency.

- It requires historical datasets and their optimal low-dimensional embeddings. This is a common limitation of all inductive learning models.
- In our empirical results, if the number of ground-truth classes of the dataset is large, the NMI will decrease compared with the optimal. This may be because the number of training sets is small in our current experiments.
- The strategy of extending to the large dataset using our method in Section 3.4 may not be optimal. We leave it as the future works.

## ACKNOWLEDGMENTS

This work was partially supported by the National Natural Science Foundation of China under Grant No.62376236, the Shenzhen Stability Science Program 2023, and the Natural Science Foundation of Zhejiang province, China, under Grant No. LMS26F030015. The authors would like to thank the group members and Prof. Tongkai Ji in the Institute of Data Intelligence at Zhejiang University of Technology for their support.

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

## A LIMITATION OF AUTODV

- It requires historical datasets and their optimal low-dimensional embeddings. This is a common limitation of all inductive learning models.

- In our empirical results, if the number of ground-truth classes of the dataset is large, the NMI will decrease compared with the optimal. This may be because the number of training sets is small in our current experiments.

- The strategy of extending to the large dataset using our method in Section 3.4 may not be optimal. We leave it as the future works.

## B PROOF FOR THEOREM 1

Before we prove Theorem 1, we provide 2 lemmas.

**Lemma 1** *The Gaussian kernel function in Eq. 2 is $L_g^{(j)}$-Lipschitz continuous with $L_g^{(j)} = \sqrt{\frac{2}{e\gamma^{(j)}}}$ depending on different choice of $\gamma^{(j)}$. We have*

$$\|\boldsymbol{S}^{(j)} - \tilde{\boldsymbol{S}}^{(j)}\|_F \leq 2\sqrt{\frac{2n}{e\gamma^{(j)}}}\|\boldsymbol{X} - \tilde{\boldsymbol{X}}\|_F. \tag{10}$$

*Proof:* Here we repeat some key notations and definitions.

$$\mathbf{S}^{(j)}[u,v] = \exp\left(-\frac{\|\boldsymbol{X}[u] - \boldsymbol{X}[v]\|_2^2}{\gamma^{(j)}}\right), \quad \tilde{\mathbf{S}}^{(j)}[u,v] = \exp\left(-\frac{\left\|\tilde{\boldsymbol{X}}[u] - \tilde{\boldsymbol{X}}[v]\right\|_2^2}{\gamma^{(j)}}\right). \tag{11}$$

Consider the Gaussian kernal function $g(\mathbf{x}) = \exp\left(-\frac{\|\mathbf{x}\|^2}{\gamma}\right)$, the gradient $\nabla g(\mathbf{x}) = -2/\gamma \exp\left(\|\mathbf{x}\|_2^2/\gamma\right)\mathbf{x}$. Then, the norm of the gradient

$$\|\nabla g(\mathbf{x})\| \leq 2/\gamma\|\mathbf{x}\|_2 \exp\left(-\|\mathbf{x}\|_2^2/\gamma\right) \leq \sqrt{\frac{2}{e\gamma}} \tag{12}$$

with the maximum value of $\nabla g(\mathbf{x})$ obtained when $\|\mathbf{x}\|_2^2 = \frac{\gamma}{2}$. Hence, $g(\cdot)$ is Lipschitz continuous with $L_g^{(j)} = \sqrt{\frac{2}{e\gamma^{(j)}}}$.

Then, for a graph $j$,

$$\|\boldsymbol{S}^{(j)} - \tilde{\boldsymbol{S}}^{(j)}\|_F^2 = \sum_{u,v}|\boldsymbol{S}^{(j)}[u,v] - \tilde{\boldsymbol{S}}^{(j)}[u,v]|^2 \leq L_g^{(j)}L_g^{(j)}\sum_{u,v}(\|\boldsymbol{X}[u] - \tilde{\boldsymbol{X}}[u]\|_2 + \|\boldsymbol{X}[v] - \tilde{\boldsymbol{X}}[v]\|_2)^2$$

$$\leq 2L_g^{(j)}L_g^{(j)}\sum_{u,v}(\|\boldsymbol{X}[u] - \tilde{\boldsymbol{X}}[u]\|_2^2 + \|\boldsymbol{X}[v] - \tilde{\boldsymbol{X}}[v]\|_2^2)$$

$$= 4nL_g^{(j)}L_g^{(j)}\sum_{u}\|\boldsymbol{X}[u] - \tilde{\boldsymbol{X}}[u]\|_2^2 = 4nL_g^{(j)}L_g^{(j)}\|\boldsymbol{X} - \tilde{\boldsymbol{X}}\|_F^2.$$

Hence,

$$\|\boldsymbol{S}^{(j)} - \tilde{\boldsymbol{S}}^{(j)}\|_F \leq 2\sqrt{n}L_g^{(j)}\|\boldsymbol{X} - \tilde{\boldsymbol{X}}\|_F = 2\sqrt{\frac{2n}{e\gamma^{(j)}}}\|\boldsymbol{X} - \tilde{\boldsymbol{X}}\|_F. \tag{13}$$

Q.E.D.

**Lemma 2** *For the perturbed positional encoding $\tilde{\boldsymbol{P}}^{(j)}$, if the SVD positional encoding is adopt, it satisfies*

$$\|\boldsymbol{P}^{(j)} - \tilde{\boldsymbol{P}}^{(j)}\|_F \leq \left(\frac{2\sqrt{2\lambda_{\max}^{(j)}}}{\delta^{(j)}} + \frac{\sqrt{d_e}}{2\sqrt{\lambda_{\min}^{(j)}}}\right)\|\boldsymbol{S}^{(j)} - \tilde{\boldsymbol{S}}^{(j)}\|_F, \tag{14}$$

where $\delta^{(j)}$ is the eigen gap between $\boldsymbol{S}^{(j)}$ and $\tilde{\boldsymbol{S}}^{(j)}$, $\lambda_{\max}^{(j)}$ is the maximum eigenvalue of $\boldsymbol{S}^{(j)}$, and $\lambda_{\min}^{(j)}$ is the minimum the $d_e$-th eigenvalue between $\boldsymbol{S}^{(j)}$ and $\tilde{\boldsymbol{S}}^{(j)}$

*Proof:* Let $\boldsymbol{S}^{(j)}$ and $\tilde{\boldsymbol{S}^{(j)}}$ perform truncated SVD with first $d_e$ singular value.

$$\boldsymbol{S}^{(j)} = \boldsymbol{U}^{(j)}\boldsymbol{\Sigma}^{(j)}\boldsymbol{V}^{(j)}, \tilde{\boldsymbol{S}}^{(j)} = \tilde{\boldsymbol{U}}^{(j)}\tilde{\boldsymbol{\Sigma}}^{(j)}\tilde{\boldsymbol{V}}^{(j)}, \tag{15}$$

Then,

$$\boldsymbol{P}^{(j)} = \boldsymbol{U}^{(j)}[:d_e]\sqrt{\boldsymbol{\Sigma}^{(j)}[:d_e,:d_e]} \triangleq \boldsymbol{U}_{d_e}^{(j)}\sqrt{\boldsymbol{\Sigma}_{d_e}^{(j)}}, \tilde{\boldsymbol{P}}^{(j)} = \tilde{\boldsymbol{U}}^{(j)}[:d_e]\sqrt{\tilde{\boldsymbol{\Sigma}}^{(j)}[:d_e,:d_e]} \triangleq \tilde{\boldsymbol{U}}_{d_e}^{(j)}\sqrt{\tilde{\boldsymbol{\Sigma}}_{d_e}^{(j)}}, \tag{16}$$

We have

$$
\begin{aligned}
\|\boldsymbol{P}^{(j)} - \tilde{\boldsymbol{P}}^{(j)}\|_F &= \|\boldsymbol{U}_{d_e}^{(j)}\sqrt{\boldsymbol{\Sigma}_{d_e}^{(j)}} - \tilde{\boldsymbol{U}}_{d_e}^{(j)}\sqrt{\tilde{\boldsymbol{\Sigma}}_{d_e}^{(j)}}\|_F \\
&\leq \left\|\boldsymbol{U}_{d_e}^{(j)}\sqrt{\boldsymbol{\Sigma}_{d_e}^{(j)}} - \tilde{\boldsymbol{U}}_{d_e}^{(j)}\sqrt{\boldsymbol{\Sigma}_{d_e}^{(j)}}\right\|_F + \left\|\tilde{\boldsymbol{U}}_{d_e}^{(j)}\sqrt{\boldsymbol{\Sigma}_{d_e}^{(j)}} - \tilde{\boldsymbol{U}}_{d_e}^{(j)}\sqrt{\tilde{\boldsymbol{\Sigma}}_{d_e}^{(j)}}\right\|_F \\
&= \left\|(\boldsymbol{U}_{d_e}^{(j)} - \tilde{\boldsymbol{U}}_{d_e}^{(j)})\sqrt{\boldsymbol{\Sigma}_{d_e}^{(j)}}\right\|_F + \left\|\sqrt{\boldsymbol{\Sigma}_{d_e}^{(j)}} - \sqrt{\tilde{\boldsymbol{\Sigma}}_{d_e}^{(j)}}\right\|_F
\end{aligned} \tag{17}
$$

For the first term,

$$\left\|(\boldsymbol{U}_{d_e}^{(j)} - \tilde{\boldsymbol{U}}_{d_e}^{(j)})\sqrt{\boldsymbol{\Sigma}_{d_e}^{(j)}}\right\|_F \leq \left\|(\boldsymbol{U}_{d_e}^{(j)} - \tilde{\boldsymbol{U}}_{d_e}^{(j)})\right\|_F\left\|\sqrt{\boldsymbol{\Sigma}_{d_e}^{(j)}}\right\|_2 = \sqrt{\lambda_{\max}^{(j)}}\left\|\boldsymbol{U}_{d_e}^{(j)} - \tilde{\boldsymbol{U}}_{d_e}^{(j)}\right\|_F, \tag{18}$$

where $\lambda_{\max}^{(j)}$ is the maximum eigenvalue of $\boldsymbol{S}^{(j)}$.

Here, we consider a simplified case that SVD process follows a consistent rule where no sign or rotation ambiguity exists. It is possible to realize as we propose a sign-flipping strategy. Based on Davis-Kahan theorem (Stewart & Sun, 1990; Yu et al., 2015), we have

$$\|\boldsymbol{U}_{d_e}^{(j)} - \tilde{\boldsymbol{U}}_{d_e}^{(j)}\|_F \leq \frac{2\sqrt{2}\|\boldsymbol{S}^{(j)} - \tilde{\boldsymbol{S}}^{(j)}\|_F}{\delta^{(j)}}, \tag{19}$$

where $\delta^{(j)}$ is the eigen gap between $\boldsymbol{S}^{(j)}$ and $\tilde{\boldsymbol{S}}^{(j)}$. The eigen gap is formally defined as $\delta^{(j)} = \inf\left\{\left|\tilde{\lambda}^{(j)} - \lambda^{(j)}\right| \ \middle|\ \lambda^{(j)} \in [\lambda_{d_e}^{(j)}, \lambda_1^{(j)}],\ \tilde{\lambda}^{(j)} \in (-\infty, \tilde{\lambda}_{d_e+1}^{(j)}] \cup (\tilde{\lambda}_1, \infty)\right\}.\tilde{\lambda}_i^{(j)}$ and$\lambda_i^{(j)}$ are the $i$-th largest eigenvalue of $\boldsymbol{S}^{(j)}$ and $\tilde{\boldsymbol{S}}^{(j)}$. Then, we have

$$\left\|(\boldsymbol{U}_{d_e}^{(j)} - \tilde{\boldsymbol{U}}_{d_e}^{(j)})\sqrt{\boldsymbol{\Sigma}_{d_e}^{(j)}}\right\|_F \leq \frac{2\sqrt{2\lambda_{\max}^{(j)}}}{\delta^{(j)}}\|\boldsymbol{S}^{(j)} - \tilde{\boldsymbol{S}}^{(j)}\|_F \tag{20}$$

For the second term,

$$\left\|\sqrt{\boldsymbol{\Sigma}_{d_e}^{(j)}} - \sqrt{\tilde{\boldsymbol{\Sigma}}_{d_e}^{(j)}}\right\|_F \triangleq \sqrt{\sum_{i=1}^{d_e}(\sqrt{s_i^{(j)}} - \sqrt{\tilde{s}_i^{(j)}})^2}.$$

For each $i$,

$$\left|\sqrt{s_i^{(j)}} - \sqrt{\tilde{s}_i^{(j)}}\right| = \frac{|s_i^{(j)} - \tilde{s}_i^{(j)}|}{\sqrt{s_i^{(j)}} + \sqrt{\tilde{s}_i^{(j)}}} \leq \frac{|s_i^{(j)} - \tilde{s}_i^{(j)}|}{2\sqrt{\lambda_{\min}^{(j)}}},$$

where $\lambda_{\min}^{(j)} = \min(\lambda_{d_e}^{(j)}, \tilde{\lambda}_{d_e}^{(j)})$, i.e. $\lambda_{\min}^{(j)}$ is the minimum the $d_e$-th eigenvalue between $\boldsymbol{S}^{(j)}$ and $\tilde{\boldsymbol{S}}^{(j)}$. Hence,

$$\left\|\sqrt{\boldsymbol{\Sigma}_{d_e}^{(j)}} - \sqrt{\tilde{\boldsymbol{\Sigma}}_{d_e}^{(j)}}\right\|_F \leq \frac{1}{2\sqrt{\lambda_{\min}^{(j)}}}\left\|\boldsymbol{\Sigma}_{d_e}^{(j)} - \tilde{\boldsymbol{\Sigma}}_{d_e}^{(j)}\right\|_F. \tag{21}$$

According to Weyl theorem(Franklin, 2012), $|s_i^{(j)} - \tilde{s}_i^{(j)}| \leq \|\boldsymbol{S}^{(j)} - \tilde{\boldsymbol{S}^{(j)}}\|_2$. Then,

$$\left\|\sqrt{\boldsymbol{\Sigma}_{d_e}^{(j)}} - \sqrt{\tilde{\boldsymbol{\Sigma}}_{d_e}^{(j)}}\right\|_F \leq \frac{\sqrt{d_e}}{2\sqrt{\lambda_{\min}^{(j)}}}\|\boldsymbol{S}^{(j)} - \tilde{\boldsymbol{S}^{(j)}}\|_2 \leq \frac{\sqrt{d_e}}{2\sqrt{\lambda_{\min}^{(j)}}}\|\boldsymbol{S}^{(j)} - \tilde{\boldsymbol{S}^{(j)}}\|_F. \tag{22}$$

Now, we combine Eq. 20 and Eq. 22 and plug them in Eq. 17, we have

$$\|\boldsymbol{P}^{(j)} - \tilde{\boldsymbol{P}}^{(j)}\|_F \leq \left( \frac{2\sqrt{2\lambda_{\max}^{(j)}}}{\delta^{(j)}} + \frac{\sqrt{d_e}}{2\sqrt{\lambda_{\min}^{(j)}}} \right) \|\boldsymbol{S}^{(j)} - \tilde{\boldsymbol{S}}^{(j)}\|_F.$$

Then, we finish the proof. Q.E.D.

**Proof of Theorem 1.**

*Proof:* Since AutoDV model consists $k$ separate GINs with the same network structures, we assume the GIN is $L_{\phi 1}$-Lipschitz continuous. Denote $\boldsymbol{H}^{(j)} = \text{GIN}_j(\boldsymbol{P}^{(j)}, \boldsymbol{S}^{(j)})$ and $\tilde{\boldsymbol{H}}^{(j)} = \text{GIN}_j(\tilde{\boldsymbol{P}}^{(j)}, \tilde{\boldsymbol{S}}^{(j)})$, we have

$$\left\| \left[ \boldsymbol{H}^{(1)}, ..., \boldsymbol{H}^{(k)} \right] - \left[ \tilde{\boldsymbol{H}}^{(1)}, ..., \tilde{\boldsymbol{H}}^{(k)} \right] \right\|_F \leq \sum_{j=1}^{k} L_{\phi 1} \left\| \begin{bmatrix} \boldsymbol{P}^{(j)} \\ \boldsymbol{S}^{(j)} \end{bmatrix} - \begin{bmatrix} \tilde{\boldsymbol{P}}^{(j)} \\ \tilde{\boldsymbol{S}}^{(j)} \end{bmatrix} \right\|_F$$

$$\leq k L_{\phi 1} \max_j \left( \|\boldsymbol{P}^{(j)} - \tilde{\boldsymbol{P}}^{(j)}\|_F + \|\boldsymbol{S}^{(j)} - \tilde{\boldsymbol{S}}^{(j)}\|_F \right)$$

Suppose GT and MLP together are $L_{\phi 2}$-Lipschitz continuous, we can have

$$\|\boldsymbol{Z} - \tilde{\boldsymbol{Z}}\|_F \leq L_{\phi 2} \left\| \left[ \boldsymbol{H}^{(1)}, ..., \boldsymbol{H}^{(k)} \right] - \left[ \tilde{\boldsymbol{H}}^{(1)}, ..., \tilde{\boldsymbol{H}}^{(k)} \right] \right\|_F$$

$$\leq k L_{\phi 1} L_{\phi 2} \max_j \left( \|\boldsymbol{P}^{(j)} - \tilde{\boldsymbol{P}}^{(j)}\|_F + \|\boldsymbol{S}^{(j)} - \tilde{\boldsymbol{S}}^{(j)}\|_F \right)$$

Let $L_\phi = L_{\phi 1} L_{\phi 2}$ and use the results in Lemma 2. We have

$$\|\boldsymbol{Z} - \tilde{\boldsymbol{Z}}\|_F \leq k L_\phi \max_j \left( \frac{2\sqrt{2\lambda_{\max}^{(j)}}}{\delta^{(j)}} + \frac{\sqrt{d_e}}{2\sqrt{\lambda_{\min}^{(j)}}} + 1 \right) \|\boldsymbol{S}^{(j)} - \tilde{\boldsymbol{S}}^{(j)}\|_F. \tag{23}$$

Now, plug in the results of Lemma 1. We have

$$\|\boldsymbol{Z} - \tilde{\boldsymbol{Z}}\|_F \leq 2k L_\phi \max_j \left( \frac{2\sqrt{2\lambda_{\max}^{(j)}}}{\delta^{(j)}} + \frac{\sqrt{d_e}}{2\sqrt{\lambda_{\min}^{(j)}}} + 1 \right) \sqrt{\frac{2n}{e\gamma^{(j)}}} \|\boldsymbol{X} - \tilde{\boldsymbol{X}}\|_F. \tag{24}$$

Let $c^{(j)} = \frac{2\sqrt{2\lambda_{\max}^{(j)}}}{\delta^{(j)}} + \frac{1}{2}\sqrt{\frac{d_e}{\lambda_{\min}^{(j)}}}$, we finish the proof. Q.E.D.

**Remark B.1** *From the intermediate result in Theorem 1, we can have the following result.*

$$\|\boldsymbol{Z} - \tilde{\boldsymbol{Z}}\|_F \leq k L_\phi \max_j \left( \frac{2\sqrt{2\lambda_{\max}^{(j)}}}{\delta^{(j)}} + \frac{\sqrt{d_e}}{2\sqrt{\lambda_{\min}^{(j)}}} + 1 \right) \|\boldsymbol{S}^{(j)} - \tilde{\boldsymbol{S}}^{(j)}\|_F. \tag{25}$$

*It indicates that if two input graphs are similar to graphs in the training set, the visualization results should be similar.*

It is possible that very different $\boldsymbol{X}$ can produce similar $\boldsymbol{S}$ because datasets can have similar internal relation structures. Hence, it guarantees the cross-domain generalization ability to some extent.

## C  PROOF OF LOSS DESIGN

We first show Eq. (7) is a special case of the Bregman divergence.

*Proof:* Let $\sigma(\|\boldsymbol{Z}_i[u] - \boldsymbol{Z}_i[v]\|_2^2) = \frac{(1+\|\boldsymbol{Z}_i[u]-\boldsymbol{Z}_i[v]\|_2^2)^{-1}}{\sum_{u',v'}(1+\|\boldsymbol{Z}_i[u']-\boldsymbol{Z}_i[v']\|_2^2)^{-1}}$, which is a normalized similarity value, applying to $\hat{\boldsymbol{Z}}_i$ and $\boldsymbol{Z}_i^*$. Let $\psi(x) = x\log(x)$, we have

$$\mathcal{K}(x, y) = x\log(x) - y\log(y) - (1 + \log(x))(x - y) = x\log(\frac{x}{y}).$$

We see that the Bregman divergence is reduced to the KLD loss. Hence, Eq. (7) is a special case of Eq. (6).                                                                                            Q.E.D.

Then, we show Eq. (8) is a special case of the Eq. 6.

*Proof:*    Let $\sigma(x) = \frac{1}{1+ax^b}$. Let $\psi(x) = \frac{1}{2}x^2$, which is strictly convex. We prove $\ell_{u,v}$ is a reduced form of Bregman divergence in the following. Let $\psi(x) = \frac{1}{2}x^2$, we have

$$\mathcal{K}(x, y) = \frac{1}{2}x^2 - \frac{1}{2}y^2 - \langle y, \ x - y \rangle = \frac{1}{2}(x - y)^2.$$

We see that the Bregman divergence is reduced to the MSE loss. Hence, $\ell_{u,v}$ is a reduced form of Bregman divergence. Since $\lambda(t)$ is a constant during the loss calculation. Eq. (8) is a linear combination of Bregman divergence by using properties in (Nock et al., 2016).                      Q.E.D.

## D    SIGN-COUNT BASED SIGN FLIPPING ALGORITHM FOR POSITIONAL ENCODING

The detailed process is shown in Algorithm 1.

---

**Algorithm 1** Sign-Count based Sign Flipping for Positional Encoding

---

**Input:** positional encoding $\boldsymbol{P} \in \mathbb{R}^{N \times d_e}$ from Eq. (3);
1: **for** $i \in \{1, ..., d_e\}$ **do**
2:     $\boldsymbol{v} \leftarrow \boldsymbol{P}[:, i]$;
3:     n_pos $\leftarrow$ sum(bool($\boldsymbol{v} > 0$));  *//count the number of positive elements in $\boldsymbol{v}$*
4:     n_neg $\leftarrow$ sum(bool($\boldsymbol{v} < 0$));  *//count the number of negative elements in $\boldsymbol{v}$*
5:     **if** n_pos $<$ n_neg **then**
6:         $\boldsymbol{v} \leftarrow -1 \cdot \boldsymbol{v}$;
7:     **end if**
8:     $\boldsymbol{P}[:, i] \leftarrow \boldsymbol{v}$;
9: **end for**
**Output:** Sign Flipped Positional Encoding $\boldsymbol{P}$.

---

## E    EXTENDING TO LARGE DATASETS

There are two methods to utilize our AutoDV on large datasets. The first one is directly inputting the large dataset into the model. This method is limited by the memory space. An example result is in Figure 6a. The second method is the proposed batch-based method. The method is scalable in large datasets. As described in Section 3.4. The process is detailed in Algorithm 2. An example of visualizing CIFAR10 with 20000 points is in Figure 6b.

---

**Algorithm 2** Batch-based AutoDV for Extending to Large Datasets

---

**Input:** Trained AutoDV model $f_\phi$, input dataset $\boldsymbol{X}$, anchor size $M_A$;
1: $\{\boldsymbol{X}^{(1)}, \boldsymbol{X}^{(2)}, \ldots, \boldsymbol{X}^{(q)}\} \leftarrow$ split($\boldsymbol{X}, q$); *// split dataset into q chunks.*
2: $c \leftarrow \boldsymbol{X}$.size$//M_A$;
3: $\boldsymbol{A} \leftarrow$ KMeans($\boldsymbol{X}, c$)[0];
4: $\hat{\boldsymbol{Z}} \leftarrow \emptyset$;
5: **for** $i \in \{1, ..., q\}$ **do**
6:     $\boldsymbol{X}_{in} = \boldsymbol{A} \cup \boldsymbol{X}^{(i)}$;
7:     $\{\boldsymbol{P}\}, \{\boldsymbol{S}\} \leftarrow$ Extract PEs and similarity matrix from $\boldsymbol{X}_{in}$;
8:     $\hat{\boldsymbol{Z}}_{out} \leftarrow f_\phi(\{\boldsymbol{P}\}, \{\boldsymbol{S}\})$
9:     $\hat{\boldsymbol{Z}} \leftarrow \hat{\boldsymbol{Z}} \cup \hat{\boldsymbol{Z}}_{out}[M_A :]$
10: **end for**
**Output:** Low-dimensional embedding for large dataset $\hat{\boldsymbol{Z}}$.

---

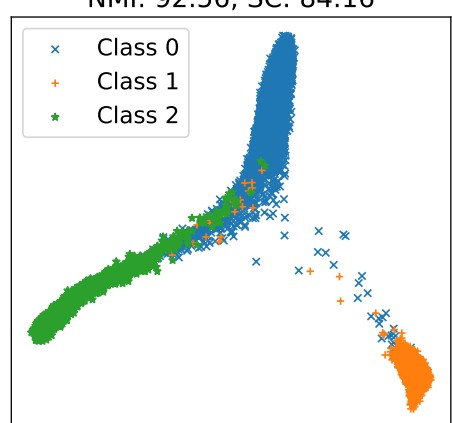
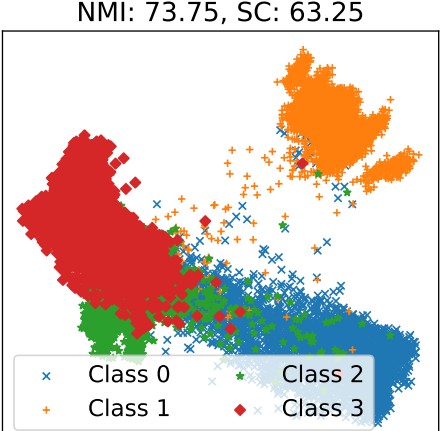

(a) AutoDV-UMAP on CIFAR10 with 10000 points.

(b) AutoDV-UMAP on CIFAR10 with 20000 points.

Figure 6: (a) is the result of extending AutoDV to a dataset with 10000 points by direct input. (b) is the result of extending AutoDV to a dataset with 20000 points by the batch-based method in section 3.4.

## F  DETAILS OF COMPUTATIONAL COMPLEXITY ANALYSIS AND COMPARISON

In the training stage, the process involves Bayesian optimization for searching the optimal hyper-parameter of the HDV algorithm, $k$ graph conversion, PE calculation, and model training. For t-SNE, the complexity for the searching is $\mathcal{O}(N_i^2 d_i + BT N_i^2 d')$, where $B$ is the number of runs for searching the optimal hyper-parameters and $T$ is the number of iterations for training. For UMAP, the complexity is $\mathcal{O}(N_i^2 d_i + BT N_i r d')$, where $r$ is the number of neighbors used in UMAP. The time complexity for $k$ PEs calculation is $\mathcal{O}(N_i^2 d_i + k N_i^2 d_e)$ by utilizing randomized SVD (Halko et al., 2011). Let $w_{\max}$ represent the maximum width of the hidden layers in the $L$-layer AutoDV model. The complexity of AutoDV forwarding for one dataset is at most $\mathcal{O}(N_i^2 d_e w_{\max} L)$ since the transformer block has complexity in $\mathcal{O}(N_i^2)$. The complexity of training loss calculation is $\mathcal{O}(N_i^2 d')$. Hence, the final training complexity of AutoDV is $\mathcal{O}(N_i^2(BT + w_{\max}))$ as $k$, $d_e$, $d'$, and $L$ are usually set as fixed constant.

In the inference stage, AutoDV only involves PE calculation from $k$-scale graphs and model forwarding. Then, the inference complexity is $\mathcal{O}(N_i^2 w_{\max})$. A comparison between AutoDV, t-SNE, and UMAP when processing a new dataset is summarized in Table 1. Furthermore, the complexity of our method can be lowered by a batch operation, i.e., splitting a large graph into $q_i$ sub-graphs with $M$ nodes and visualizing them one by one. The complexity will be $\mathcal{O}(k N_i M w_{\max})$. More importantly, PEs in our method can be pre-computed, which further accelerates our method in practice.

## G  EXPERIMENT DATASETS SUMMARY

Table 6 lists the details of datasets used in this paper, including name, size, number of classes, and dimensionality. Here, as mentioned in Section 4.1, MNIST, FMNIST, and CIFAR10 are features in latent space extracted by CLIP. Campbell, PBMC68K, Baron Human, and Mouse_retina are filtered out the rare cells.

## H  EVALUATION METRIC DETAILS

**Normalized Mutual Information (NMI):**  NMI evaluates how well the clustering results match the true labels. It quantifies the amount of mutual information shared between the cluster assign-

Table 6: Dataset Summary.

| Name | size | dims | n_class |
|---|---|---|---|
| MNIST* | 60000 | 512 | 10 |
| FMNIST* | 60000 | 512 | 10 |
| CIFAR10* | 50000 | 512 | 10 |
| Campbell* | 13289, | 26774 | 6 |
| PBMC68K* | 66332 | 32738 | 8 |
| Baron Human* | 7771 | 20125 | 6 |
| Mouse_retina* | 8352 | 6198 | 5 |
| Palmer_Penguins | 333 | 10 | 3 |
| Forty_Soybean_Cultivars_from_Subsequent_Harvests | 320 | 9 | 40 |
| Cirrhosis_Patient_Survival_Prediction | 312 | 776 | 3 |
| PIRvision_FoG_presence_detection | 15302 | 56 | 3 |
| Auction_Verification | 2043 | 7 | 2 |
| Period_Changer | 90 | 1177 | 2 |
| SUPPORT2 | 332 | 64 | 2 |
| Wine | 178 | 13 | 3 |
| Iris | 150 | 4 | 3 |
| Cryotherapy | 90 | 5 | 2 |
| Phishing_Websites | 11055 | 30 | 2 |
| Sirtuin6_Small_Molecules | 100 | 6 | 2 |
| Land_Mines | 338 | 3 | 5 |
| Z-Alizadeh_Sani | 606 | 78 | 2 |
| NHANES | 2278 | 7 | 2 |
| Differentiated_Thyroid_Cancer_Recurrence | 383 | 55 | 2 |
| RT-IoT2022 | 123117 | 94 | 12 |
| Regensburg_Pediatric_Appendicitis | 780 | 3158 | 2 |
| DARWIN | 174 | 624 | 2 |
| TUNADROMD | 4465 | 241 | 3 |
| Toxicity | 171 | 1203 | 2 |
| Drug_induced_Autoimmunity_Prediction | 597 | 196 | 2 |
| MetroPT-3 | 1516948 | 15 | 2 |
| Accelerometer_Gyro_Mobile_Phone | 31991 | 6 | 2 |
| Caesarian_Section_Classification_Dataset | 80 | 5 | 2 |
| NPHA | 714 | 14 | 3 |

ments and the ground truth. To measure the quality of the HDV results, we first perform K-Means clustering upon the output low-dimension embeddings. Then, we calculate the NMI of the clustering results. The higher NMI, the better, because an objective of HDV is to show the clustering structure within the high-dimensional dataset.

Let $U$ be the set of ground truth labels and $V$ be the clustering assignments. The Normalized Mutual Information is defined as:

$$\text{NMI}(U, V) = \frac{2 \cdot I(U; V)}{H(U) + H(V)}$$

where $I(U; V)$ is the mutual information between $U$ and $V$, and $H(U)$ and $H(V)$ denote the entropy of $U$ and $V$, respectively. Concretely,

$$I(U; V) = \sum_{u \in U} \sum_{v \in V} P(u, v) \log \frac{P(u, v)}{P(u)P(v)}$$

$$H(U) = -\sum_{u \in U} P(u) \log P(u), \quad H(V) = -\sum_{v \in V} P(v) \log P(v)$$

**Silhouette Coefficient (SC):** SC is also a metric of clustering. It quantifies how well a point is clustered, i.e., how similar it is to its own cluster compared to other clusters.

For each data point $i$, define $a(i)$ as the average intra-cluster distance (distance to other points in the same cluster), and $b(i)$ as the average nearest-cluster distance (distance to the closest neighboring cluster). The silhouette coefficient $s(i)$ is given by:

$$s(i) = \frac{b(i) - a(i)}{\max\{a(i), b(i)\}}$$

The overall silhouette coefficient for the dataset is the mean value of $s(i)$ over all samples:

$$\text{SC} = \frac{1}{n} \sum_{i=1}^{n} s(i)$$

## I  DATA PREPARATION DETAILS

Due to the limited video memory during the training, we randomly downsample the datasets to generate several subsets if the number of points in the dataset is less than 3000. To increase the diversity of the training subsets, we first randomly select a number of classes, then randomly sample 10 subsets with different sizes from the selected classes. The detailed process is in Algorithm 3. Note that the maximum number of points of subsets is 3000. Since the total number of classes used in the experiments is smaller than 10, we go through all combinations of the selected classes. For image data experiments, we randomly select 500 subsets from MNIST and FMNIST for training, and randomly select 50 subsets for testing. For gene data experiment, we filtered out the NMI less than 10 of the optimal low-dimensional embedding for training and testing. For tabular data experiments, we do the same filtering. Then, we randomly split the whole sampled subsets into training and testing, where 30% is for testing.

**Other training details and justifications:** We list additional training details not mentioned in the main text in the following.

- At the training stage, we pre-computed PEs for all $k$-scale graphs in advance to saving the processing times.
- We perform 50 times Bayesian optimization searches for both t-SNE and UMAP.

We list the following justifications for potential questions:

- *Why filter out NMI less than 10% when generating $\mathbf{Z}^*$?* We observe that if the NMI is very small, the results of t-SNE and UMAP will collapse into one dense area. It usually means the training fails at this dataset for t-SNE or UMAP. These datasets are harmful for the AutoDV training since they are meaningless.

- *Why filter out the rare cell in gene data experiments?* The rare cell is usually present in a small portion of the gene data. The portion is usually smaller than 3% according to (Wang et al., 2024). It could be considered as noise information and hence harmful for the AutoDV training. So, we filter out them. We leave a AutoDV training that is robust to the noised structure or dataset in the future work.

---

**Algorithm 3** Dataset Downsampling in Training Data Preparation

---

**Input:** dataset $\boldsymbol{X}$, the number of class $C$, maximum subset size $N_{max}$;
1: $\mathcal{D}_{\text{train}} \leftarrow \emptyset$;
2: **for** $i \in \{1, \ldots, C\}$ **do**
3:     $\mathcal{C} \leftarrow \texttt{combination}(C, i)$;
4:     $\boldsymbol{X}_c \leftarrow$ select all points with label in $\mathcal{C}$;
5:     $\boldsymbol{s} \leftarrow \texttt{linspace}(100, N_{max}, 10)$; *// generate 10 subsets with different sizes.*
6:     **for** $j \in \{1, \ldots, 10\}$ **do**
7:         $N' \leftarrow \texttt{random}(\boldsymbol{s}[j], \boldsymbol{s}[j+1])$;
8:         $\boldsymbol{X}' \leftarrow$ randomly downsample $N'$ samples from $\boldsymbol{X}_c$;
9:         $\mathcal{D}_{\text{train}} \leftarrow \mathcal{D}_{\text{train}} \cup \{\boldsymbol{X}'\}$;
10:     **end for**
11: **end for**
**Output:** All sampled subsets $\mathcal{D}_{\text{train}}$.

---

## J    IMPLEMENTATION DETAILS

All experiments are implemented by Pytorch (Paszke et al., 2017) with Deep Graph Library (Wang et al., 2019) on NVIDIA GeForce RTX 4090 and Intel Xeon Gold 5117 platform. We set $k = 5$ and set $\gamma^{(j)}$ as $[0.1, 0.5, 1, 2, 5]$ times the median value of the pairwise distance matrix. We utilize SVD as the positional encoding and set $d_e = 64$. For GNNs, we utilize a 3-layer GIN with 128 hidden dimensions and an 8-layer graph transformer with 4 heads and a 2048-dimensional feed-forward layer. A 2-layer MLP head is appended for the dimensionality reduction. All layers are activated by SeLu (Klambauer et al., 2017) and layer normalization (Xiong et al., 2020). Unless specified, the model is trained for 100 epochs by AdamW (Loshchilov & Hutter, 2017). The learning rate is set $1 \times 10^{-4}$ with cosine anneal decay. During each batch, there are $18000$ nodes processed. The results of t-SNE and UMAP are trained separately. For baseline methods with deep neural networks, we use a 10-layer MLP network for fair comparison with ours.

## K    MORE TRANSFERABILITY RESULTS

We report the SC results in Figure 7. It consistently shows a good transferability of AutoDV.

## L    ABLATION STUDY

**Effectiveness of PE**    Due to limited resources, we design a simplified experiment setting, where only 5 training subsets are randomly sampled from MNIST, FMNIST, CIFAR10. They are MNIST with class labels in [0, 1, 2, 3, 4], MNIST with class labels in [5, 6, 7, 8, 9], FMNIST with class labels in [0, 1, 2, 3, 4], FMNIST with class labels in [5, 6, 7, 8, 9], and CIFAR10 with class labels in [0, 1, 2, 3, 4]. There is only 1 subset sampled from CIFAR10, which is CIFAR10 with class labels in [5, 6, 7, 8, 9]. All subsets are with 3000 points. This is the simplest toy experiment setting, where the test dataset remains unseen during the training. We test different types of PE in AutoDV-tSNE, including Laplacian PE (LapPE), non-negative matrix factorization (NMF), random walk (RRWP) (Ma et al., 2023), SignNet (Lim et al., 2022), and SVD. In this line of testing, there is no sign flipping strategy for SVD and LapPE. We test two variants of LapPE, using only eigenvectors of graph Laplacian matrix (LapPE-V) or both eigenvectors and eigenvalues (LapPE-VS). The results are reported in Figure 8a, where SVD PE performs the best. Although SignNet claims it can handle the sign ambiguity problem well, it instead is harmful to our AutoDV model training and is inefficient when $N_i$ and $d_e$ is large.

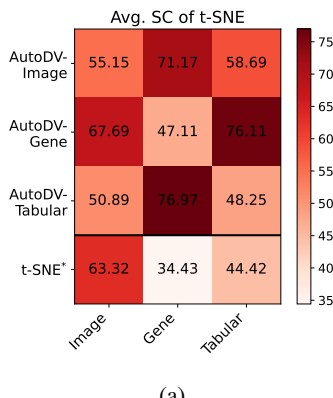 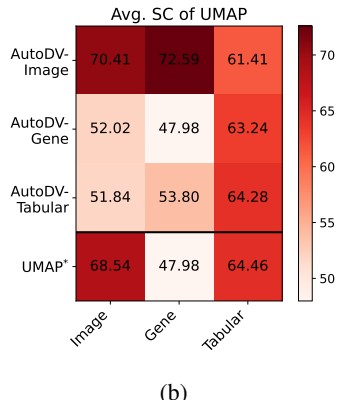

|     |     |
| --- | --- |
| (a) | (b) |

Figure 7: Cross-domain transferability performance using t-SNE and UMAP. Each heatmap shows average SC results for (a) t-SNE and (b) UMAP. Within each map, columns denote the source domain and rows the target domain for the first three rows; the fourth row ("TSNE*" or "UMAP*") reports the within-domain performance of the optimal low-dimensional embedding. Test sets for each domain are the same as in Tables 2, 3, and 4.

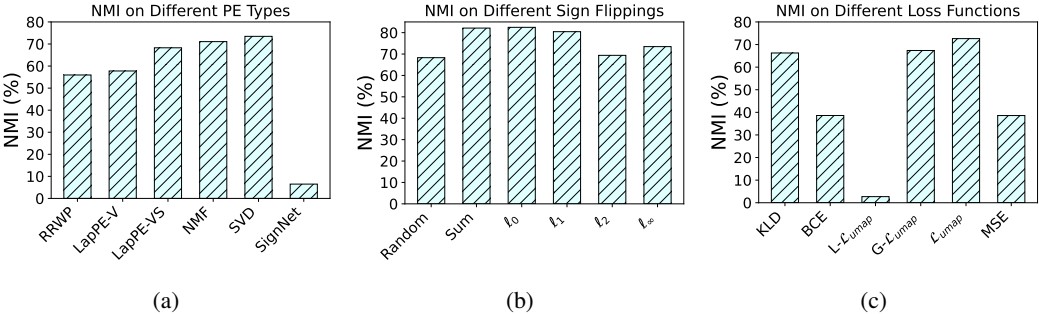

|     |     |     |
| --- | --- | --- |
| (a) | (b) | (c) |

Figure 8: Results of ablation studies using different (a) PEs, (b) sign flipping strategies, and (c) training loss functions.

**Effectiveness of PE Sign Flipping**   We fix the PE as SVD and test various sign flipping strategies in AutoDV-tSNE, including random, summation, $\ell_0$, $\ell_1$, $\ell_2$, and $\ell_\infty$. $\ell_\infty$ is the default SVD result by numpy in Python. Random means that we randomly flip the sign of each dimension of PE. Summation decides the sign based on the sign of the column sum of $S$. $\ell_0$ is the adopted strategy in Algorithm 1. $\ell_1$, $\ell_2$, and $\ell_\infty$ are variant of $\ell_0$ by using different norms in line 3 and 4. The test setting is the same as above. The results are presented in Figure 8b. Among the strategies compared, the sign-count-based ($\ell_0$) flipping method achieves the highest NMI. Although the summation approach yields comparable performance, we adopt the $\ell_0$ strategy due to its intuitive interpretation that the majority sign dominates the overall direction.

**Effectiveness of Loss Design**   We test the performance of AutoDV-UMAP under different loss designs, including KLD, binary cross entropy (BCE), local only (L-$\mathcal{L}_{\mathrm{umap}}$), global only (G-$\mathcal{L}_{\mathrm{umap}}$), and mean square error (MSE). MSE is loss similar to i-tSNE (Roman-Rangel & Marchand-Maillet, 2019). KLD is the same as $\mathcal{L}_{\mathrm{tsne}}$. BCE is similar to KLD but utilizing the cross entropy. The test setting is the same as Table 2. The results are shown in Figure 8c. The proposed loss function $\mathcal{L}_{\mathrm{umap}}$ in Eq. 8 achieves the highest NMI. In contrast, using only the local loss results in poor performance, as the initialized model tends to collapse all points into close proximity. This collapse leads to an artificially small local loss value, ultimately causing training failure.

**Sensitivity of PE Dim. $d_e$:**   Due to limited time, we keep the experiment settings the same as we described in "Effectiveness of PE", where there are only 5 datasets for training and 1 dataset for testing. The results of AutoDV-tSNE are listed below.

Table 7: Results of different PE dimensions compared with t-SNE.

| PE dim. | 8 | 16 | 32 | 64 | 128 | t-SNE* |
|---|---|---|---|---|---|---|
| train NMI | 76.83 | 77.70 | 78.52 | 78.18 | 78.12 | 80.39 |
| train SC | 57.43 | 57.75 | 57.10 | 57.34 | 57.24 | 54.23 |
| test NMI | 78.41 | 77.15 | 81.78 | 82.48 | 73.24 | 86.28 |
| test SC | 64.61 | 61.50 | 62.30 | 62.43 | 60.07 | 67.25 |

It is seen that setting PE dim. to 32 or 64 is a good choice. If PE dim. is small, the structural information may loss. If PE dim. is large, the model could face the problem of underfitting or receive more noisy structural information.

**Sensitivity of the number of graphs ($k$):** Since the choice of $k$ will directly affect the model running cost, we tested $k = [1, 3, 5, 7, 9]$ to show the effectiveness of multi-scale graphs. The detailed selection of sigmas is here.

Table 8: Sigmas used for different $k$ values.

| $k$ | 1 | 3 | 5 | 7 | 9 |
|---|---|---|---|---|---|
| sigmas | [1] | [0.5, 1, 2] | [0.1, 0.5, 1, 2, 5] | [0.05, 0.1, 0.5, 1, 2, 5, 10] | [0.01, 0.05, 0.1, 0.5, 1, 2, 5, 10, 20] |

The results of AutoDV-tSNE varing different k are listed below.

Table 9: Results of different $k$ values compared with t-SNE.

| $k$ | 1 | 3 | 5 | 7 | 9 | t-SNE* |
|---|---|---|---|---|---|---|
| train NMI | 38.32 | 76.75 | 78.18 | 77.86 | 36.23 | 80.39 |
| train SC | 42.20 | 58.32 | 57.34 | 57.25 | 53.64 | 54.23 |
| test NMI | 59.83 | 75.54 | 82.48 | 74.08 | 53.27 | 86.28 |
| test SC | 45.79 | 57.60 | 62.43 | 56.41 | 48.12 | 67.25 |

It is seen that less or more k will lead to a very low training NMI. If k is small, the structural information is limited to train the model. If k is large with too large sigmas and too small sigmas, the structural information will become noisy leading to difficult convergence.

**Justification of Hyper-parameter Selection during AutoDV Training:** As shown above, although there still are hyper-parameter selections for AutoDV, two core hyper-parameters, $d_e$ and $k$, can be reasonably decided. For other hyper-parameters such as learning rate, hidden dimension, etc. All deep learning-based methods need to tune these hyper-parameters. In AutoDV, luckily, we can also tune these hyper-parameters, including $d_e$ and $k$, by spliting the original training set into training and validation sets. Then, the hyper-parameters can be tuned according to the model performance on the validation set, which is the same as any supervised learning. Hence, at the **training** stage, AutoDV will not meet the hyper-parameter tuning challenge in the **unsupervised** setting as discussed in Section 2

## M    MORE QUALITATIVE VISUALIZATION RESULTS

We here present more visualization results on training set and test set. For image data from CLIP, the results on the training and testing set are in Figure 9 to 10 and Figure 11 to 12, respectively. We show the results that contains 2 to 8 classes. It is seen that AutoDV is a little underfitting when the number of classes becomes large. It may be because training data with large number of classes is small. Such imbalance leads a biased training. This problem can be solved by carefully generating the subsets for training. Nevertheless, the performance of AutoDV on dataset with a small number of classes is visually acceptable and can be well generalized to the unseen test dataset.

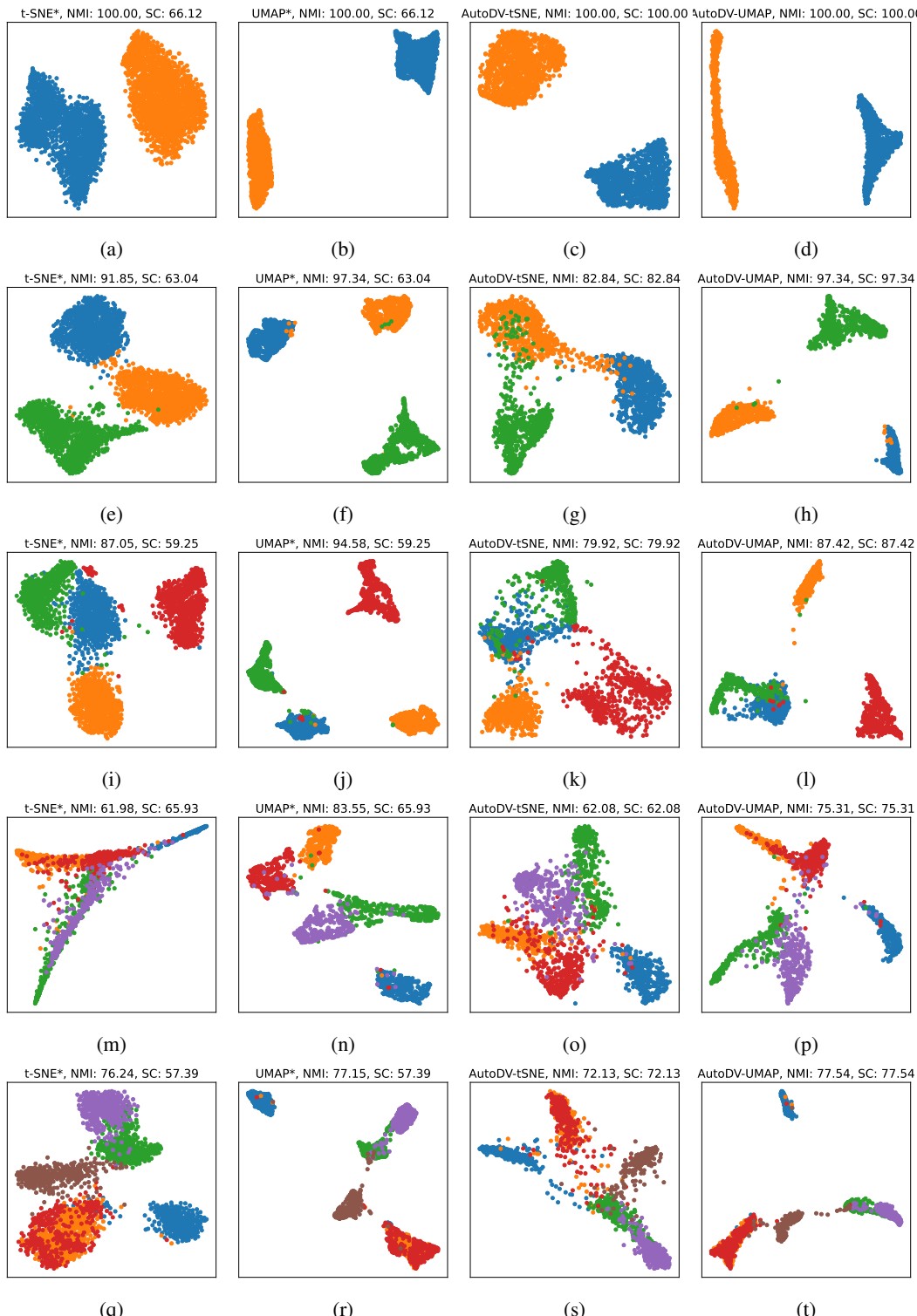

Figure 9: Visualization of t-SNE*, UMAP*, AutoDV-tSNE, and AutoDV-UMAP on **training** datasets of image data experiments. Each row represents the different number of classes.

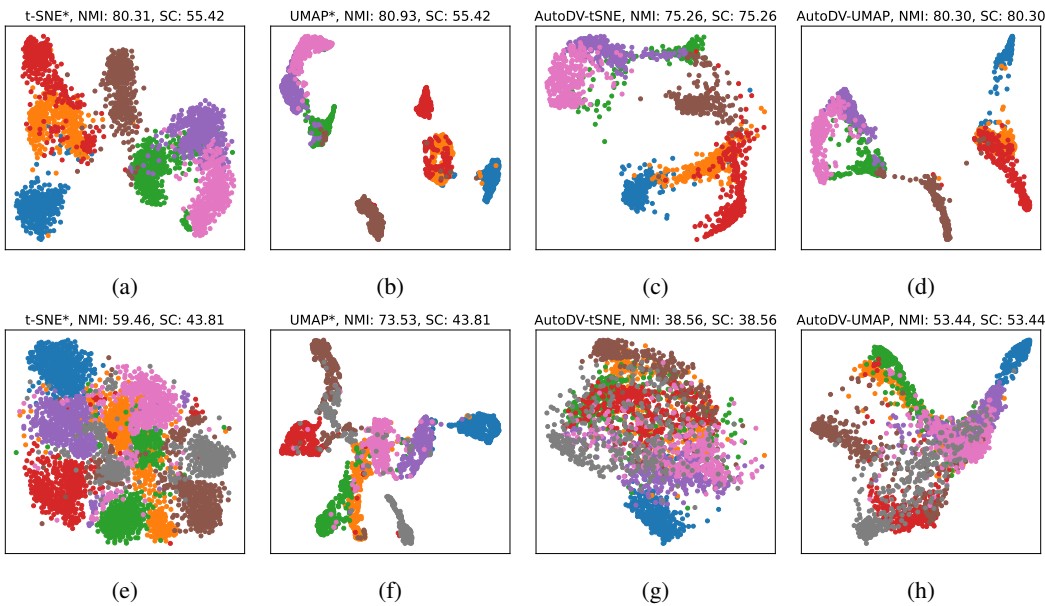

Figure 10: Continued visualization of t-SNE*, UMAP*, AutoDV-tSNE, and AutoDV-UMAP on **training** datasets of image data experiments for 7-class data and 8-class data.

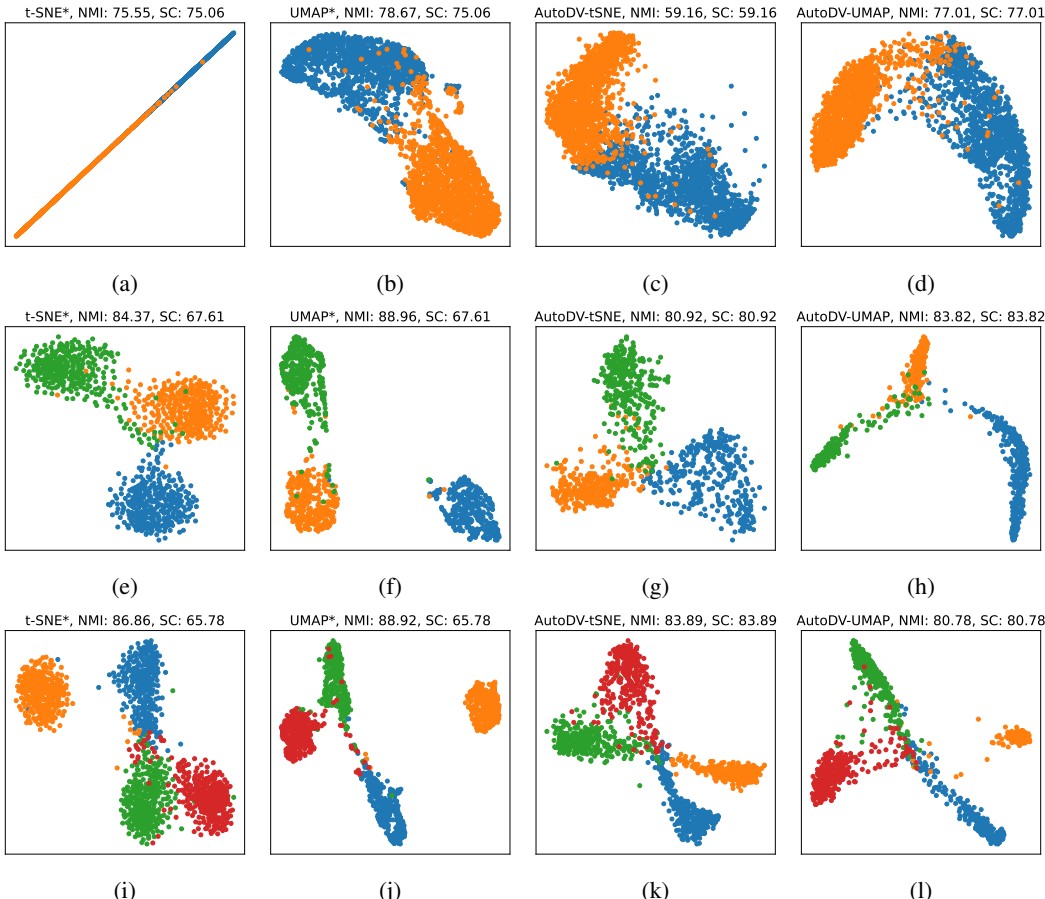

Figure 11: Visualization of t-SNE*, UMAP*, AutoDV-tSNE, and AutoDV-UMAP on **testing** datasets of image data experiments. Each row represents the different number of classes.

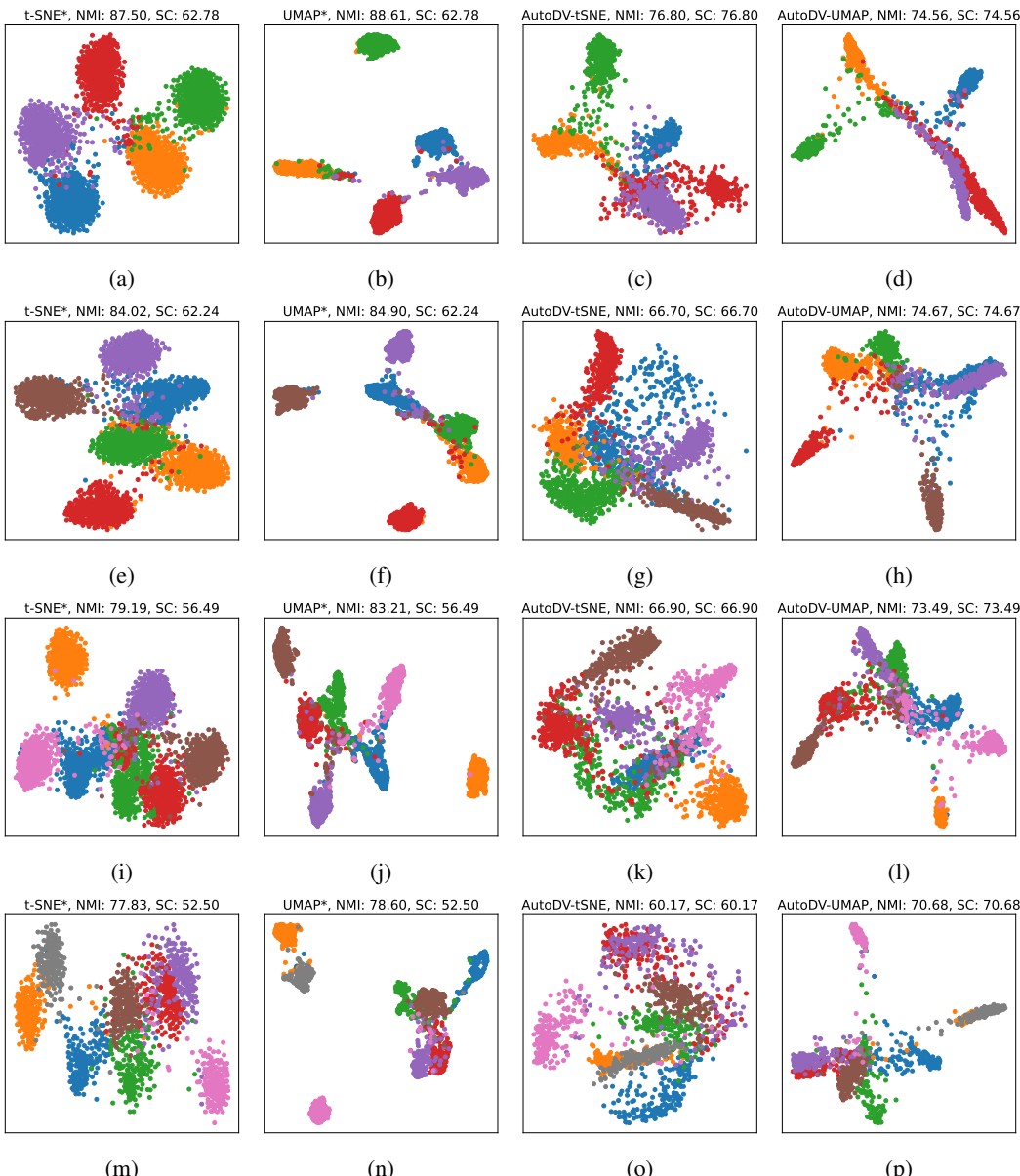

Figure 12: Continued visualization of t-SNE*, UMAP*, AutoDV-tSNE, and AutoDV-UMAP on **test** datasets of image data experiments from 5-class data to 8-class data.

# N    USE OF LARGE LANGUAGE MODELS

We use LLM to polish our verbal writing as it is really useful.

## O    TRAINING DATA DIVERSITY STUDY

We conduct a series of experiment to study how the training data diversity affect the model performance and generalization ability. It includes structural diversity, training sample size, dimensionality, domain diversity.

**How do the number of training data and the diversity jointly affect the performance?**    We keep the same setting as that in the Table 2. Then, we vary the number of training data in [10, 50, 500, 1000], and see how the performance changes on the AutoDV-UMAP. Note that if we increase the number of training data under such setting, both training sample and training diversity will be increased because the training data are uniformly subsampled. The results on the same test data are shown below in Table 10

Table 10: Training Sample Size and Diversity v.s. test NMI/SC

| training size | 10 | 50 | 500 | 1000 | UMAP* |
|---|---|---|---|---|---|
| test NMI | 57.19 | 58.72 | 73.28 | 74.17 | 80.45 |
| test SC | 51.11 | 52.01 | 70.41 | 71.3 0 | 68.54 |

In the results, we see that if the training sample size becomes large, the performance on a fixed testing set will become better.

To further understand how the training sample size and training sample diversity affect the model performance separately, we conduct the following studies.

**How does training-data structural diversity affect performance?**    We define training-data diversity in terms of geometric structural diversity. We construct a family of synthetic datasets with controlled structures, where the geometric structural diversity is controlled by the variety of cluster structures present in the training set.

Concretely, we generate datasets from mixtures of $c$ Gaussian components in 20-dimension space, so that each dataset contains exactly $c$ well-separated clusters. We vary $c$ from 1 to 9, and refer to these datasets as *c-cluster datasets*. Thus, each value of $c$ corresponds to a distinct geometric structure.

To study the effect of structural diversity, we gradually enlarge the set of structures used for training. Let the *structural coverage* of the training data be the set of cluster configurations included in it. We consider a sequence of training regimes where the coverage increases from

$$\{1\text{-cluster}\},\ \{1\text{-cluster}, 2\text{-cluster}\},\ \ldots,\ \{1\text{-cluster}, \ldots, 8\text{-cluster}\}.$$

In other words, the $k$-th regime ($k = 1, \ldots, 8$) uses all datasets whose number of clusters $c$ satisfies $1 \leq c \leq k$, so that the structural diversity of the training data increases with $k$. We then evaluate how the model's generalization ability changes as this structural diversity grows. The training sample size is fixed at 10. For the testing data, we use a dataset from 9-cluster datasets. We use AutoDV-UMAP with 10 epoch training. Qualitative and quantitative results are reported in Figure 13.

It is seen that AutoDV-UMAP training with 1-cluster data can generalize a lot on 9-cluster data but is not perfect. As the coverage or structural diversity increased, the generalization ability gets better and better. On the one hand, it indicates AutoDV cannot generalize well if the geometric structure of the test data is never seen. One the other hand, the results indicate the more diverse in the training data, the better the generalization.

**How does the number of training data solely affect the performance?**    Here, we use training data from $\{1\text{-cluster}, 2\text{-cluster}\}$ with different number of training samples in [10, 50, 100, 250, 500, 1000]. The test data is the same as above, i,e, a dataset from 9-cluster. The performance curve in terms of test NMI is illustrated in Figure 14. The result indicates that increasing the number of training sample can help the generalization, but it is very limited. Compared with the result in the bottom right in Figure 13, the performance can easily increase to 1.0 by increasing the training data diversity. However, here the number of training data is increased to 1000, the performance is still lower than 0.98. It means solely increasing the training sample cannot efficiently help the model performance.

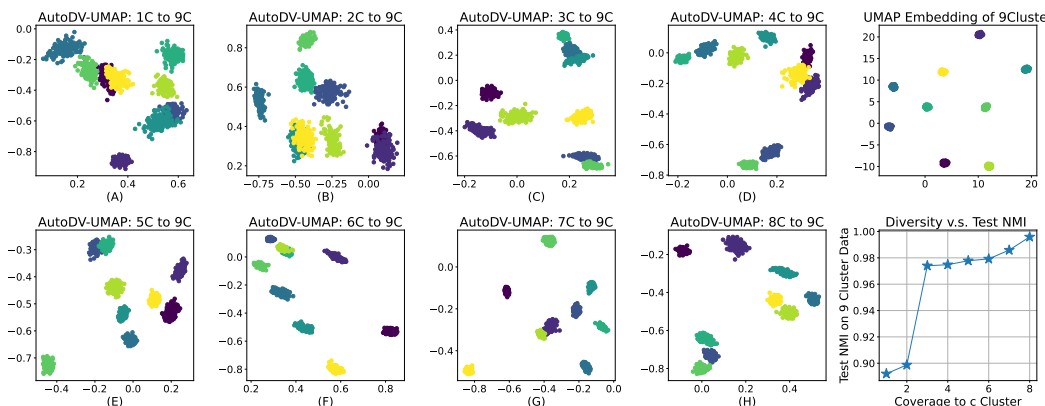

Figure 13: Figure (A)-(H) are qualitative visualization of different coverage. The upper right corner is the UMAP visualization of the test datasets. The bottom right corner is an NMI performance curve varying training data diversity.

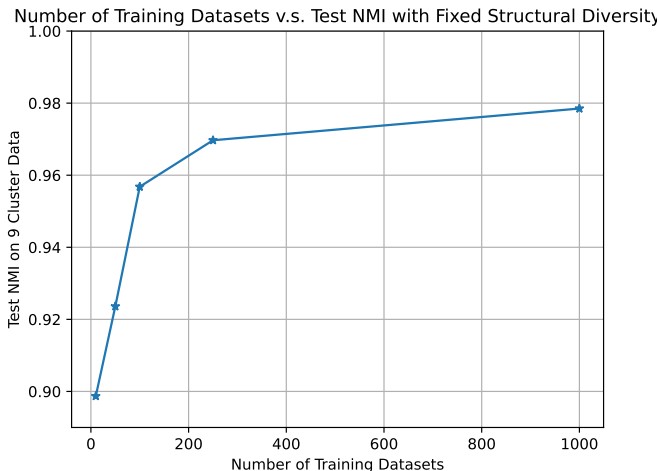

Figure 14: Performance curve of varying the different training sample size with fixed structural diversity. Training data coverage is {1-cluster, 2-cluster}.

**How does the dimension of training data affect the performance?** Here, we study the problem by varying the dimension of 2-cluster datasets. We test the performance of AutoDV-UMAP trained on 2-cluster datasets in [2, 4, 8, 16, 32, 64, 128, 256, 512] dimensional space. The test data is from 2-cluster and 9-cluster in 2-dimension space and 1024-dimension space, separately. The results are shown in Figure 15. In the results, as expected, the dimensionality of training data will not affect the performance if the test data share the same geometric structure with the training data though they are in different dimensions (Figure 15(a) and 15(c)). In addition, it is of interest to find that when the training data come from a higher-dimensional space, the model generalizes poorly to lower-dimensional test data (Figure 15 (b)). In contrast, when the training data are low-dimensional and the model is evaluated on higher-dimensional data, the generalization performance remains strong (Figure 15 (d)). This asymmetry may be related to the fact that, as the dimensionality increases, pairwise distances between samples drawn from a Gaussian distribution tend to grow, altering the geometry of the data manifold.

**How does the domain diversity of training data affect the performance?** We argue that the domain diversity is not the key to affect the performance, while the structural diversity is still the most important factor. To understand this, we illustrate two results of AutoDV-tSNE on image data.

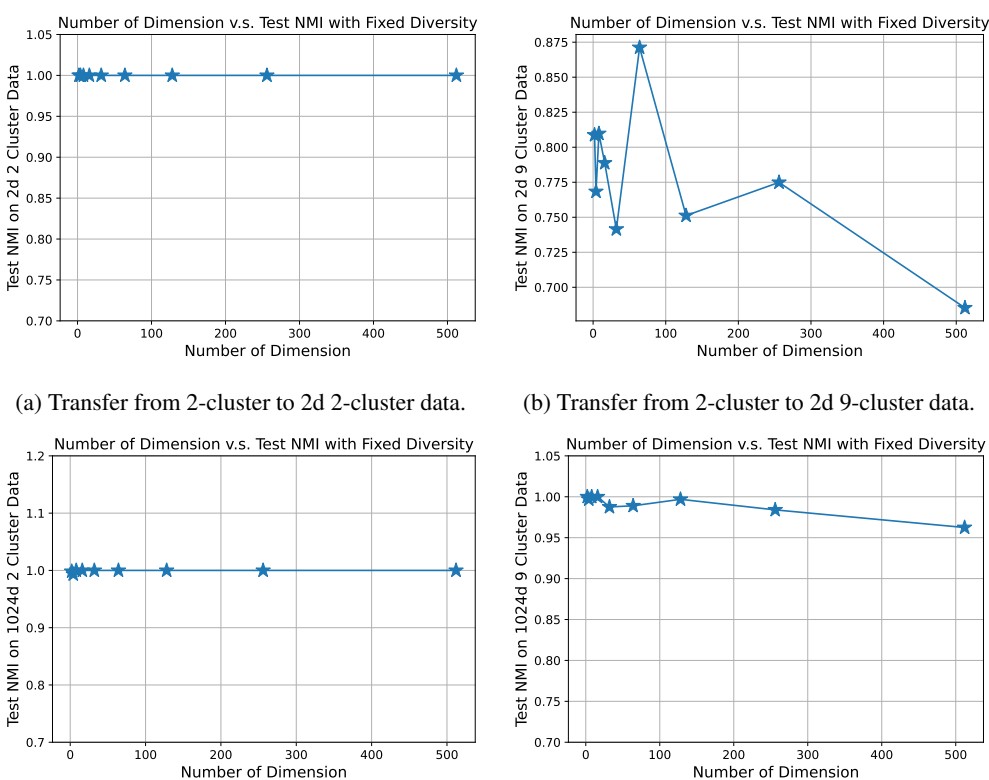

(a) Transfer from 2-cluster to 2d 2-cluster data.  (b) Transfer from 2-cluster to 2d 9-cluster data.

(c) Transfer from 2-cluster to 1024d 2-cluster data.  (d) Transfer from 2-cluster to 1024d 9-cluster data.

Figure 15: Performance on different dimensions and geometric structures varying training data dimensions with fixed training diversity.

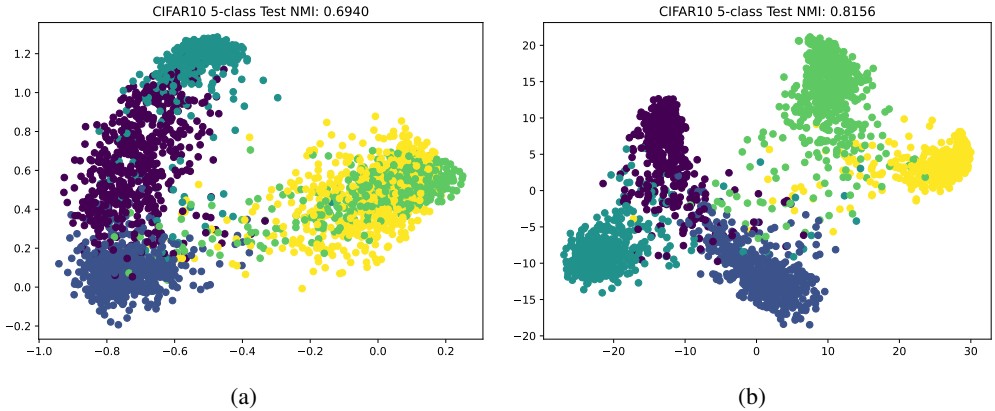

Figure 16: Visualization and test NMI on a 5-class CIFAR10 dataset. (a) Performance of model trained on 2-class data from MNIST and FMNIST. (b) Performance of model trained on 5-class data from MNIST.

Due to limited time, we conduct a toy example. First, we randomly sample 5 2-class datasets from MNIST and FMNIST for training. The training data is with high domain diversity. Second, we randomly sample 5 5-class solely from MNIST to train another model. The corresponding training data is with low domain diversity. Finally, we compare the testing results on a 5-class dataset from CIFAR10. The results are shown in Figure 16. It is seen that increasing the domain diversity does not help with model generalization ability.

## P  ADDITIONAL METRICS FOR EVALUATION

We provide results using more evaluation metrics of Table 2. The distance rank correlation is calculated by spearman rank correlation. The triplet preservation is consistent with Wang et al. (2021). The result are in Table 11. It is seen that our method can obtain a comparable performance with t-SNE* and UMAP*.

Table 11: Performance comparison on Image data (values are mean, subscript indicates $\pm$ variance, all in %).

| Metric | t-SNE* | UMAP* | Default t-SNE | Default UMAP | AutoDV-tSNE | AutoDV-UMAP |
|---|---|---|---|---|---|---|
| Trust@1 | $93.18_{\pm5.91}$ | $94.32_{\pm1.61}$ | $93.11_{\pm0.47}$ | $95.06_{\pm1.19}$ | $87.34_{\pm2.23}$ | $88.62_{\pm1.96}$ |
| Trust@10 | $92.32_{\pm4.91}$ | $93.92_{\pm1.62}$ | $91.95_{\pm0.91}$ | $94.46_{\pm1.30}$ | $87.62_{\pm1.84}$ | $88.78_{\pm1.95}$ |
| Trust@50 | $92.24_{\pm3.89}$ | $92.84_{\pm2.33}$ | $91.35_{\pm1.30}$ | $93.31_{\pm1.59}$ | $88.26_{\pm2.04}$ | $89.41_{\pm1.83}$ |
| Continuity@1 | $95.58_{\pm2.02}$ | $97.53_{\pm1.20}$ | $94.79_{\pm0.47}$ | $98.03_{\pm0.51}$ | $93.34_{\pm1.32}$ | $93.97_{\pm1.17}$ |
| Continuity@10 | $94.59_{\pm2.02}$ | $95.35_{\pm1.44}$ | $93.68_{\pm0.88}$ | $95.77_{\pm0.96}$ | $92.10_{\pm1.50}$ | $92.92_{\pm1.32}$ |
| Continuity@50 | $93.84_{\pm1.96}$ | $93.21_{\pm2.51}$ | $92.89_{\pm1.27}$ | $93.73_{\pm1.39}$ | $91.38_{\pm1.77}$ | $92.21_{\pm1.41}$ |
| Rank Correlation | $65.58_{\pm5.74}$ | $62.17_{\pm8.86}$ | $62.68_{\pm4.78}$ | $64.86_{\pm7.40}$ | $62.90_{\pm10.15}$ | $65.19_{\pm6.27}$ |
| Triplet | $74.21_{\pm2.44}$ | $72.48_{\pm3.37}$ | $73.06_{\pm1.98}$ | $73.49_{\pm2.89}$ | $73.19_{\pm3.67}$ | $74.20_{\pm2.22}$ |
| CA-knn@1 | $87.83_{\pm10.60}$ | $90.45_{\pm4.35}$ | $86.95_{\pm3.99}$ | $89.99_{\pm4.56}$ | $82.99_{\pm10.37}$ | $86.23_{\pm7.56}$ |
| CA-knn@10 | $89.77_{\pm9.27}$ | $92.41_{\pm3.72}$ | $89.05_{\pm3.73}$ | $92.69_{\pm3.79}$ | $86.61_{\pm8.60}$ | $89.29_{\pm6.47}$ |
| CA-knn@50 | $88.63_{\pm9.87}$ | $91.49_{\pm5.25}$ | $88.54_{\pm4.12}$ | $91.53_{\pm3.81}$ | $86.08_{\pm9.04}$ | $88.32_{\pm7.14}$ |
| NMI | $77.04_{\pm8.81}$ | $80.45_{\pm6.64}$ | $71.71_{\pm13.52}$ | $79.72_{\pm6.68}$ | $68.70_{\pm9.04}$ | $73.28_{\pm7.64}$ |
| SC | $63.32_{\pm9.52}$ | $68.54_{\pm6.58}$ | $47.24_{\pm9.91}$ | $67.61_{\pm6.8}$ | $55.15_{\pm6.57}$ | $70.41_{\pm7.14}$ |
| KNN Overlap@1 | $8.17_{\pm7.95}$ | $8.91_{\pm3.50}$ | $10.67_{\pm1.83}$ | $10.96_{\pm4.18}$ | $2.05_{\pm1.73}$ | $2.44_{\pm2.50}$ |
| KNN Overlap@10 | $21.61_{\pm8.54}$ | $26.43_{\pm8.24}$ | $24.00_{\pm5.62}$ | $29.62_{\pm7.71}$ | $10.85_{\pm6.05}$ | $12.44_{\pm6.78}$ |
| KNN Overlap@50 | $41.36_{\pm11.02}$ | $42.00_{\pm9.63}$ | $41.51_{\pm9.60}$ | $43.29_{\pm9.82}$ | $29.33_{\pm11.54}$ | $31.84_{\pm12.41}$ |
| Jaccard@1 | $8.17_{\pm7.95}$ | $8.91_{\pm3.50}$ | $10.67_{\pm1.83}$ | $10.96_{\pm4.18}$ | $2.05_{\pm1.73}$ | $2.44_{\pm2.50}$ |
| Jaccard@10 | $13.33_{\pm6.07}$ | $16.60_{\pm6.04}$ | $15.02_{\pm4.87}$ | $18.88_{\pm5.93}$ | $6.27_{\pm3.97}$ | $7.28_{\pm4.54}$ |
| Jaccard@50 | $27.96_{\pm9.55}$ | $28.08_{\pm8.51}$ | $27.93_{\pm9.25}$ | $29.21_{\pm9.16}$ | $18.59_{\pm9.81}$ | $20.57_{\pm10.62}$ |

The results of Gene data with more metrics are in Table 12. It is seen that the results are basically consistent with the results in Table 3. AutoDV-tSNE and AutoDV-UMAP outperforms t-SNE and UMAP in terms of many metrics such as Rank Correlation and Triplet preservation.

Table 12: Performance comparison on Gene data (values are mean, subscript indicates $\pm$ variance, all in %).

| Metric | t-SNE* | UMAP* | Default t-SNE | Default UMAP | AutoDV-tSNE | AutoDV-UMAP |
|---|---|---|---|---|---|---|
| Trust@1 | $56.46_{\pm5.27}$ | $53.67_{\pm4.88}$ | $53.74_{\pm4.96}$ | $54.34_{\pm4.79}$ | $53.63_{\pm4.72}$ | $53.75_{\pm4.74}$ |
| Trust@10 | $55.88_{\pm5.54}$ | $53.78_{\pm4.88}$ | $53.59_{\pm4.62}$ | $54.01_{\pm4.92}$ | $54.47_{\pm4.80}$ | $54.32_{\pm4.73}$ |
| Trust@50 | $58.76_{\pm7.50}$ | $55.20_{\pm5.51}$ | $55.65_{\pm6.48}$ | $55.12_{\pm5.64}$ | $57.72_{\pm5.55}$ | $56.68_{\pm5.50}$ |
| Continuity@1 | $74.24_{\pm6.80}$ | $48.20_{\pm20.28}$ | $68.71_{\pm14.57}$ | $63.53_{\pm11.52}$ | $69.06_{\pm3.12}$ | $55.27_{\pm9.90}$ |
| Continuity@10 | $74.52_{\pm5.65}$ | $56.79_{\pm11.57}$ | $69.52_{\pm13.70}$ | $66.48_{\pm7.27}$ | $70.33_{\pm2.73}$ | $60.01_{\pm6.04}$ |
| Continuity@50 | $78.03_{\pm6.60}$ | $61.82_{\pm6.78}$ | $72.26_{\pm14.42}$ | $63.74_{\pm5.56}$ | $73.79_{\pm3.00}$ | $62.73_{\pm5.92}$ |
| Rank Correlation | $38.75_{\pm12.30}$ | $10.58_{\pm16.44}$ | $23.28_{\pm19.96}$ | $5.07_{\pm18.83}$ | $48.67_{\pm11.14}$ | $29.37_{\pm13.36}$ |
| Triplet | $63.29_{\pm5.24}$ | $54.02_{\pm5.21}$ | $58.07_{\pm7.30}$ | $52.63_{\pm5.92}$ | $64.04_{\pm3.10}$ | $58.03_{\pm2.95}$ |
| CA-knn@1 | $90.34_{\pm6.24}$ | $85.54_{\pm7.02}$ | $81.16_{\pm8.31}$ | $85.47_{\pm6.78}$ | $91.37_{\pm5.31}$ | $92.89_{\pm4.28}$ |
| CA-knn@10 | $92.81_{\pm4.16}$ | $90.35_{\pm4.94}$ | $87.08_{\pm6.20}$ | $90.20_{\pm4.79}$ | $92.58_{\pm4.16}$ | $94.27_{\pm3.37}$ |
| CA-knn@50 | $91.02_{\pm4.51}$ | $90.01_{\pm4.87}$ | $86.31_{\pm7.78}$ | $89.89_{\pm4.65}$ | $91.23_{\pm4.47}$ | $92.15_{\pm4.69}$ |
| NMI | $32.73_{\pm30.76}$ | $28.85_{\pm34.72}$ | $15.67_{\pm21.9}$ | $22.45_{\pm33.23}$ | $33.22_{\pm28.72}$ | $33.03_{\pm24.99}$ |
| SC | $34.43_{\pm4.63}$ | $47.98_{\pm20.10}$ | $35.84_{\pm13.33}$ | $47.98_{\pm24.91}$ | $47.11_{\pm10.82}$ | $47.98_{\pm12.19}$ |
| KNN Overlap@1 | $0.70_{\pm0.46}$ | $0.91_{\pm0.74}$ | $0.32_{\pm0.34}$ | $1.55_{\pm0.93}$ | $0.21_{\pm0.22}$ | $0.14_{\pm0.18}$ |
| KNN Overlap@10 | $2.83_{\pm1.65}$ | $3.06_{\pm2.44}$ | $1.97_{\pm1.43}$ | $3.87_{\pm2.64}$ | $2.07_{\pm1.37}$ | $1.74_{\pm1.38}$ |
| KNN Overlap@50 | $13.24_{\pm9.26}$ | $10.16_{\pm7.58}$ | $9.96_{\pm7.81}$ | $10.12_{\pm7.27}$ | $11.05_{\pm7.84}$ | $9.02_{\pm7.26}$ |
| Jaccard@1 | $0.70_{\pm0.46}$ | $0.91_{\pm0.74}$ | $0.32_{\pm0.34}$ | $1.55_{\pm0.93}$ | $0.21_{\pm0.22}$ | $0.14_{\pm0.18}$ |
| Jaccard@10 | $1.82_{\pm1.18}$ | $1.70_{\pm1.40}$ | $1.19_{\pm0.96}$ | $2.16_{\pm1.54}$ | $1.36_{\pm0.89}$ | $0.99_{\pm0.81}$ |
| Jaccard@50 | $9.76_{\pm6.92}$ | $5.65_{\pm4.58}$ | $6.44_{\pm5.48}$ | $5.62_{\pm4.37}$ | $8.43_{\pm6.25}$ | $5.31_{\pm4.62}$ |

The results of the Tabular data is in Table 13. Similarly, it is seen that AutoDV-tSNE/UMAP outperfoms t-SNE/UMAP not only in terms of NMI and SC, but also many metrics such as CA-KNN, Rank Correlation, and Triplet preservation.

We noticed that despite AutoDV can obtain a good CA-knn performance indicating a fairly good local preservation, the KNN Overlap and Jaccard are not good enough. This problem can be solved by adding a new regularization term during the training to further pay attention to the local structure. We leave this for a future improvement.

Table 13: Performance comparison on Tabular data (values are mean, subscript indicates $\pm$ variance, all in %).

| Metric | t-SNE* | UMAP* | Default t-SNE | Default UMAP | AutoDV-tSNE | AutoDV-UMAP |
|---|---|---|---|---|---|---|
| Trust@1 | $98.41_{\pm1.13}$ | $91.11_{\pm9.86}$ | $98.23_{\pm2.74}$ | $97.78_{\pm1.21}$ | $91.31_{\pm9.57}$ | $86.50_{\pm11.88}$ |
| Trust@10 | $97.56_{\pm1.82}$ | $90.18_{\pm9.81}$ | $97.39_{\pm3.11}$ | $97.26_{\pm1.65}$ | $90.22_{\pm9.66}$ | $84.17_{\pm11.88}$ |
| Trust@50 | $95.47_{\pm3.55}$ | $85.95_{\pm10.07}$ | $95.77_{\pm3.87}$ | $94.12_{\pm3.90}$ | $89.39_{\pm9.71}$ | $81.98_{\pm12.18}$ |
| Continuity@1 | $98.51_{\pm1.05}$ | $93.22_{\pm6.31}$ | $98.52_{\pm1.27}$ | $98.62_{\pm0.81}$ | $95.95_{\pm5.10}$ | $93.11_{\pm6.17}$ |
| Continuity@10 | $97.90_{\pm1.54}$ | $89.99_{\pm7.96}$ | $97.94_{\pm1.78}$ | $97.60_{\pm1.33}$ | $94.75_{\pm5.96}$ | $90.10_{\pm7.56}$ |
| Continuity@50 | $96.30_{\pm3.04}$ | $84.67_{\pm10.20}$ | $96.85_{\pm2.83}$ | $93.58_{\pm3.67}$ | $93.25_{\pm6.76}$ | $86.29_{\pm9.41}$ |
| Rank Correlation | $55.10_{\pm22.41}$ | $35.19_{\pm32.46}$ | $55.43_{\pm24.03}$ | $46.99_{\pm28.54}$ | $58.59_{\pm25.58}$ | $40.10_{\pm32.26}$ |
| Triplet | $72.33_{\pm9.52}$ | $64.17_{\pm11.94}$ | $73.27_{\pm10.63}$ | $68.53_{\pm12.16}$ | $73.93_{\pm9.73}$ | $67.32_{\pm12.16}$ |
| CA-knn@1 | $94.61_{\pm9.29}$ | $94.26_{\pm8.30}$ | $94.34_{\pm10.74}$ | $93.73_{\pm10.50}$ | $94.49_{\pm10.25}$ | $94.69_{\pm6.79}$ |
| CA-knn@10 | $93.88_{\pm10.01}$ | $94.25_{\pm9.91}$ | $93.59_{\pm11.14}$ | $93.70_{\pm10.61}$ | $94.37_{\pm10.09}$ | $94.39_{\pm6.56}$ |
| CA-knn@50 | $91.95_{\pm10.47}$ | $91.36_{\pm11.56}$ | $91.81_{\pm10.85}$ | $90.81_{\pm11.10}$ | $92.67_{\pm11.10}$ | $93.20_{\pm7.12}$ |
| NMI | $30.92.80_{\pm12.20}$ | $24.80_{\pm16.2}$ | $23.53_{\pm11.21}$ | $15.13_{\pm12.08}$ | $33.45_{\pm21.60}$ | $35.15_{\pm34.54}$ |
| SC | $44.42_{\pm10.00}$ | $64.46_{\pm15.91}$ | $43.87_{\pm10.57}$ | $61.81_{\pm19.10}$ | $48.25_{\pm10.70}$ | $64.28_{\pm9.72}$ |
| KNN Overlap@1 | $30.01_{\pm15.82}$ | $9.45_{\pm6.63}$ | $31.18_{\pm16.95}$ | $17.30_{\pm9.08}$ | $21.76_{\pm26.01}$ | $19.51_{\pm25.13}$ |
| KNN Overlap@10 | $46.81_{\pm17.56}$ | $28.56_{\pm17.47}$ | $47.46_{\pm18.45}$ | $43.09_{\pm15.82}$ | $31.64_{\pm23.58}$ | $24.31_{\pm21.00}$ |
| KNN Overlap@50 | $58.94_{\pm14.60}$ | $38.40_{\pm15.17}$ | $59.72_{\pm14.60}$ | $54.81_{\pm12.41}$ | $46.30_{\pm19.66}$ | $33.16_{\pm16.68}$ |
| Jaccard@1 | $30.01_{\pm15.82}$ | $9.45_{\pm6.63}$ | $31.18_{\pm16.95}$ | $17.30_{\pm9.08}$ | $21.76_{\pm26.01}$ | $19.51_{\pm25.13}$ |
| Jaccard@10 | $35.74_{\pm17.10}$ | $20.23_{\pm15.01}$ | $36.50_{\pm18.00}$ | $31.59_{\pm14.67}$ | $24.49_{\pm23.83}$ | $18.44_{\pm19.73}$ |
| Jaccard@50 | $45.82_{\pm16.03}$ | $26.97_{\pm13.23}$ | $46.71_{\pm15.90}$ | $41.26_{\pm12.93}$ | $35.07_{\pm20.06}$ | $23.48_{\pm14.45}$ |

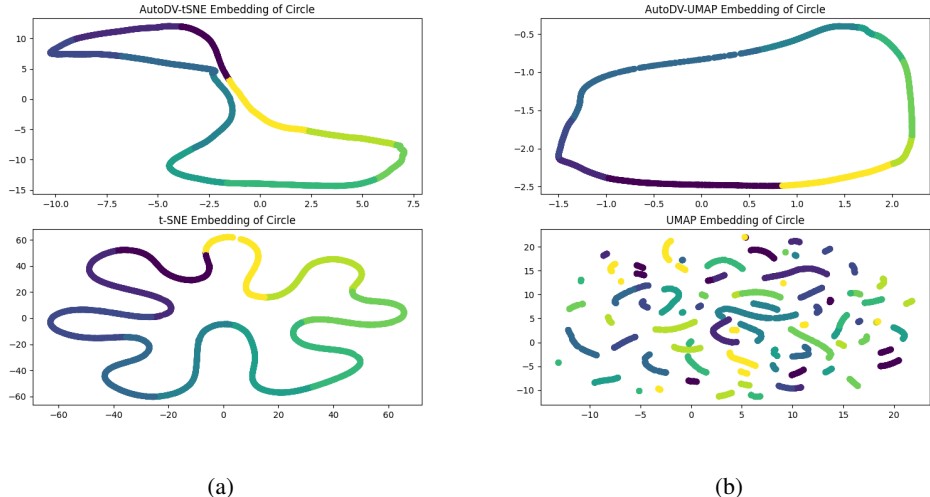

(a)                                    (b)

Figure 17: Visualization on Circle dataset. Circle samples 5000 points uniformly on the unit circle and then bin them by angle into 10 equal angular sectors (arcs). (a) Performance of AutoDV-tSNE and default t-SNE. (b) Performance of AutoDV-UMAP and default UMAP.

## Q   GENERALIZATION BOUNDARY AND FAILURE CASE ANALYSIS

We analyze how far away our model can generalize to a new unseen dataset. The experiment reported so far are using classification centric datasets, where these dataset may share some intrinsic structurally similarity. So, it can generalize well. We would like to see whether it has any failure case. Here, we further explore the generalization boundary of our models by answering two questions.

**Q1:  Can AutoDV trained on image data generalize to non-classification centric dataset?**
Rather than testing AutoDV-UMAP and AutoDV-tSNE on classification centric datasets, we test AutoDV-UMAP and and AutoDV-tSNE trained on image data on some dataset focusing on manifold structure, such as Circle, Lineage, Mammoth, which is also used in Wang et al. (2021). The results are compared with default UMAP and default t-SNE. The results of Circle is in Figure 17. It is seen that AutoDV can successfully generalize to the circled structure. The results of Lineage is in Figure 18. It is seen that AutoDV can allocate the point in a line though they are not in a straight line. In contrast, t-SNE and UMAP fail to do so. The results of Mammoth is in Figure 19. It is seen that our model finally fails to generalize to datasets emphasizing manifold structure and local patterns.

AutoDV fails to generalize to Mamooth because the training data is classification centric. The model does not learn the manifold structure during the training. The problem can be solved by training an AutoDV model using data emphasizing manifold structure such as Swiss Roll and Swiss Hole. Then, we further conduct the experiment where an AutoDV model is trained by default t-SNE results of Swiss Roll and Swiss Hole. We report the test results on newly sampled Swiss Roll and Swiss Hole, and Mammoth in Figure 20, 21, and 22. It is seen that AutoDV can successfully showing the roll and hole in the Swiss Roll and Swiss Hole. In addition, AutoDV can correctly allocate the legs and body of the mammoth with local structure preserved. Note that Mammoth is totally new to the AutoDV model showing a strong generalization ability of our method.

**Q2: Can AutoDV model generalize to dataset with high-noise, or overlapping clusters, or even on random noise?**   We select a AutoDV-UMAP model trained on 20 2-cluster datasets for the analysis. The definition of 2-cluster data is described in Appendix O.

- **High-noise:** We study global noise and local noise. Specifically, we randomly sample noise from a uniform distribution in the test data as the global noise. We randomly perturb

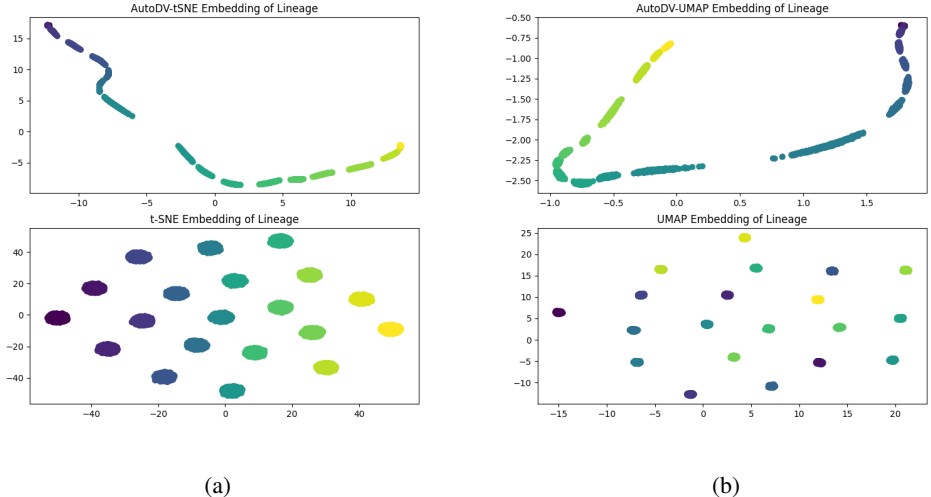

(a) (b)

Figure 18: Visualization on Lineage dataset. Circle is bunch of Gaussian blobs arranged in a straight line in high-dimensional space. (a) Performance of AutoDV-tSNE and default t-SNE. (b) Performance of AutoDV-UMAP and default UMAP.

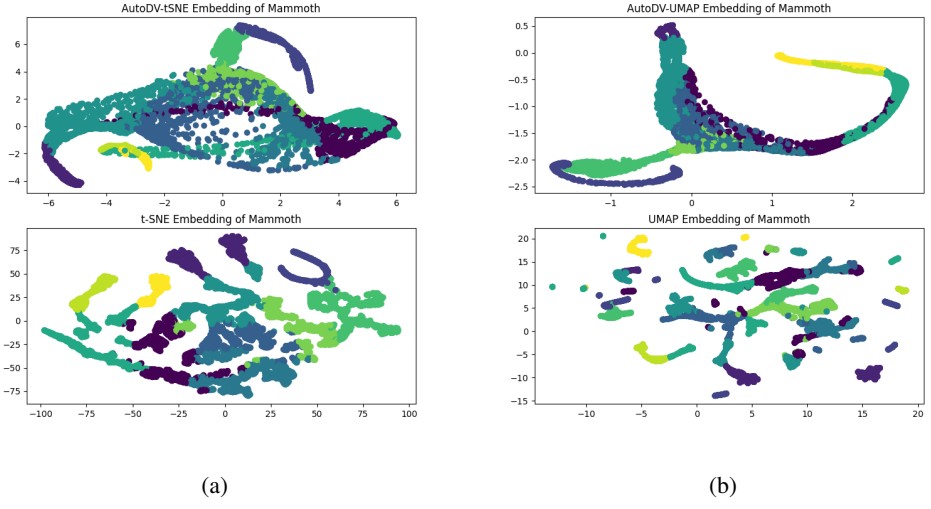

(a) (b)

Figure 19: Visualization on Mammoth dataset. (a) Performance of AutoDV-tSNE and default t-SNE. (b) Performance of AutoDV-UMAP and default UMAP.

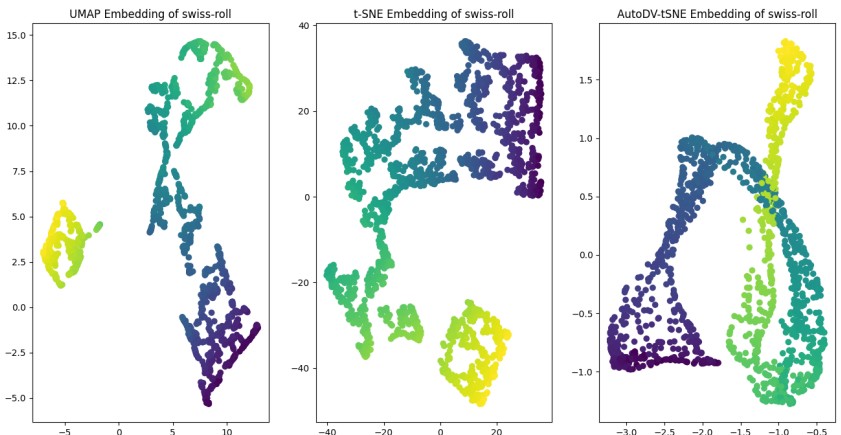

Figure 20: Visualization on a newly sampled Swiss Roll dataset. (left) Performance of default UMAP. (middle) Performance of default t-SNE. (right) Performance of AutoDV-tSNE trained on Swiss Roll and Swiss Hole.

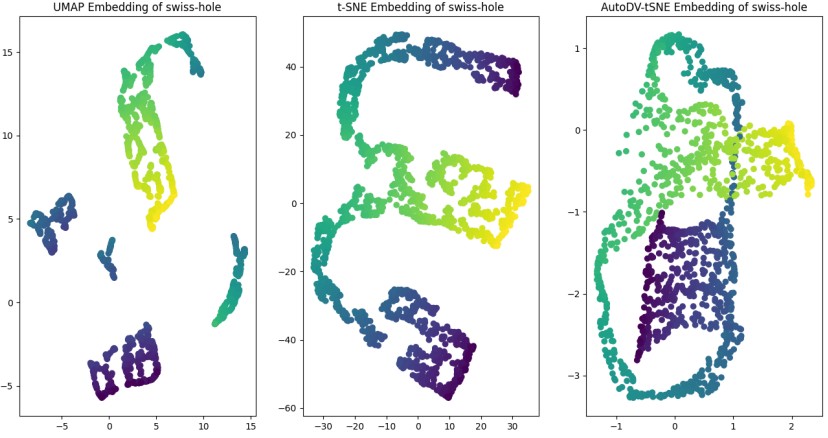

Figure 21: Visualization on a newly sampled Swiss Hole dataset. (left) Performance of default UMAP. (middle) Performance of default t-SNE. (right) Performance of AutoDV-tSNE trained on Swiss Roll and Swiss Hole.

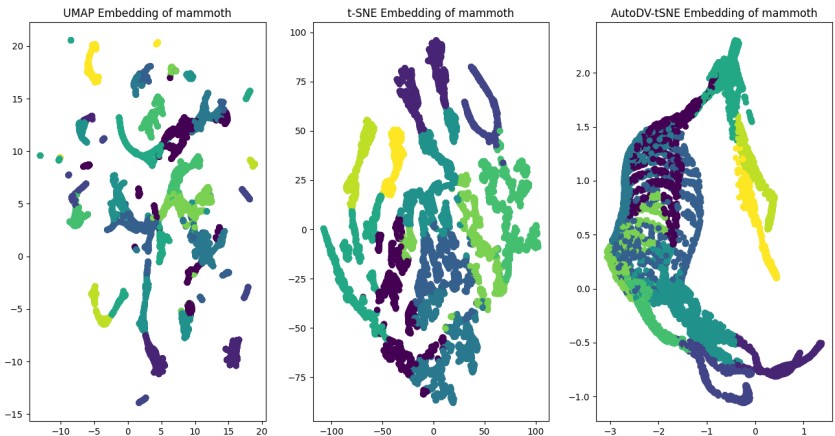

Figure 22: Visualization on Mammoth dataset. (left) Performance of default UMAP. (middle) Performance of default t-SNE. (right) Performance of AutoDV-tSNE trained on Swiss Roll and Swiss Hole.

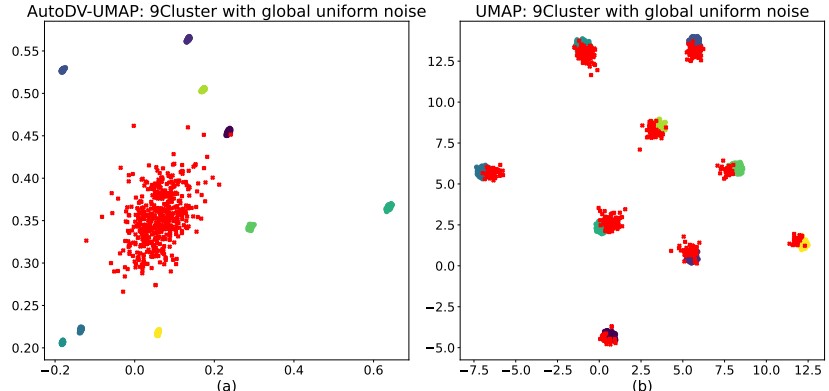

Figure 23: Visualization of 9-Cluster dataset with global noise using AutoDV-UMAP trained on 2-Cluster data (a) and UMAP (b). Red points indicate the noise.

each points within a radius using a uniform noise for the local noise. The test set is a 9-cluster dataset with 1500 points in 1024 demension space. The results of global noise is in Figure 23. The results of local noise with small radius and large radius are in Figure 24 and Figure 25, respectively. It is seen that if the data contains global noise, our AutoDV model can effectively recognize the noise and keep the original cluster structure. For the local noise, the AutoDV model can correctly show the location of the noise.

- **Overlap cluster:** We show results of the overlap cluster in 3 degree, low, middle, and high. For better understanding, we use 2-dimension 2-Cluster data for testing. The original data is also illustrated. The results are presented in Figure 26, 27, and 28. In the results, AutoDV can still illustrate the 2-cluster structure while UMAP tend to split the cluster due to the overlapping.

- **Random noise:** We directly visualize 3 types of noise with zero mean, Gaussian noise, uniform noise, and Laplacian noise. The results are shown in Figure 29, 30, and 31. It is seen that AutoDV tends to form a 2-cluster structure. It is because the model is trained with 2-cluster datasets. However, it is better than UMAP in terms of showing a randomness.

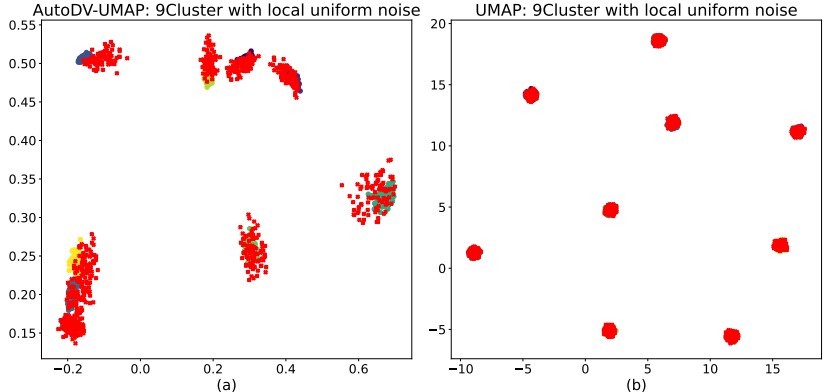

Figure 24: Visualization of 9-Cluster dataset with small radius local noise using AutoDV-UMAP trained on 2-Cluster data (a) and UMAP (b). Red points indicate the noise.

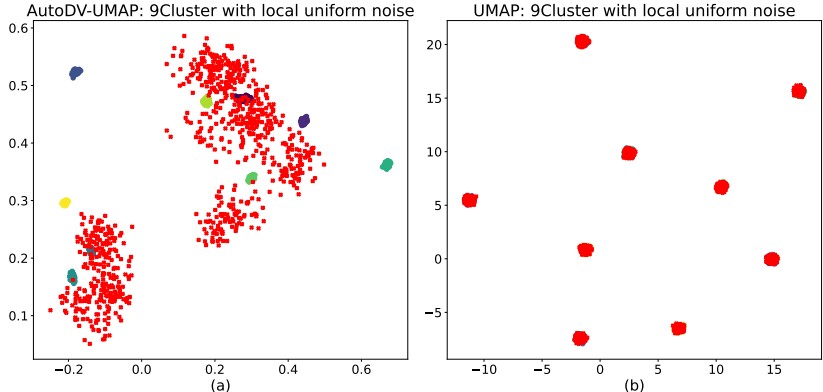

Figure 25: Visualization of 9-Cluster dataset with large radius local noise using AutoDV-UMAP trained on 2-Cluster data (a) and UMAP (b). Red points indicate the noise.

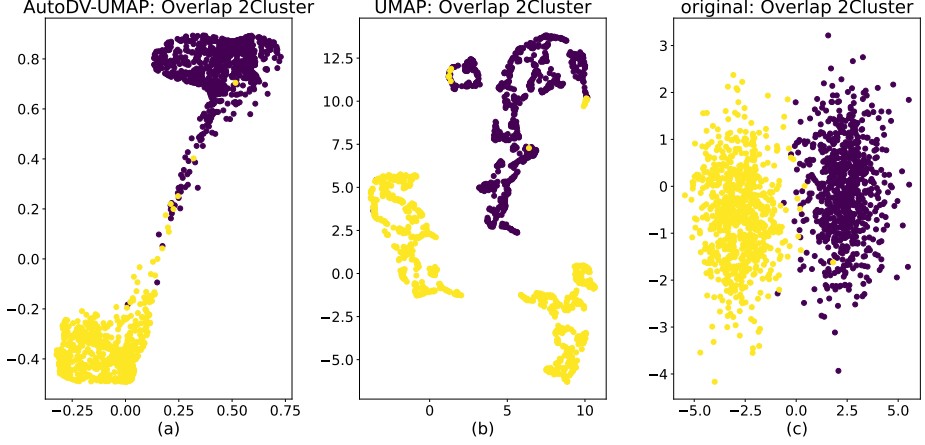

Figure 26: Visualization of 2-Cluster dataset with low level overlap using AutoDV-UMAP trained on 2-Cluster data (a) and UMAP (b). (c) is the original cluster allocation.

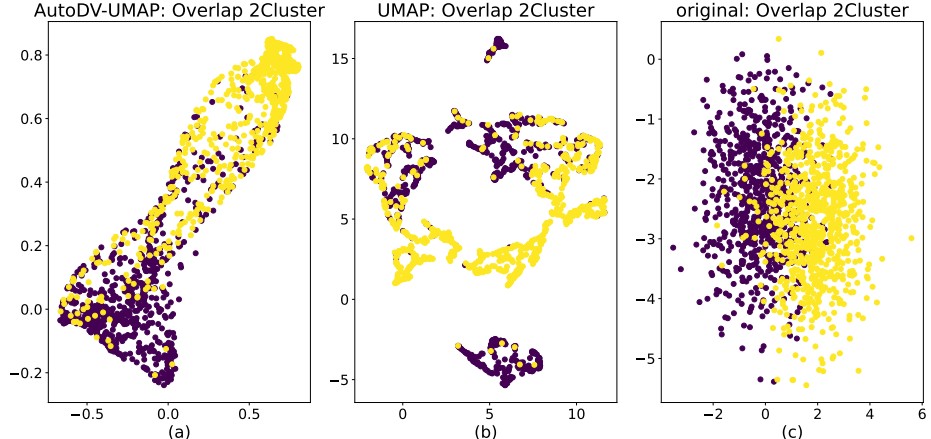

Figure 27: Visualization of 2-Cluster dataset with middle level overlap using AutoDV-UMAP trained on 2-Cluster data (a) and UMAP (b). (c) is the original cluster allocation.

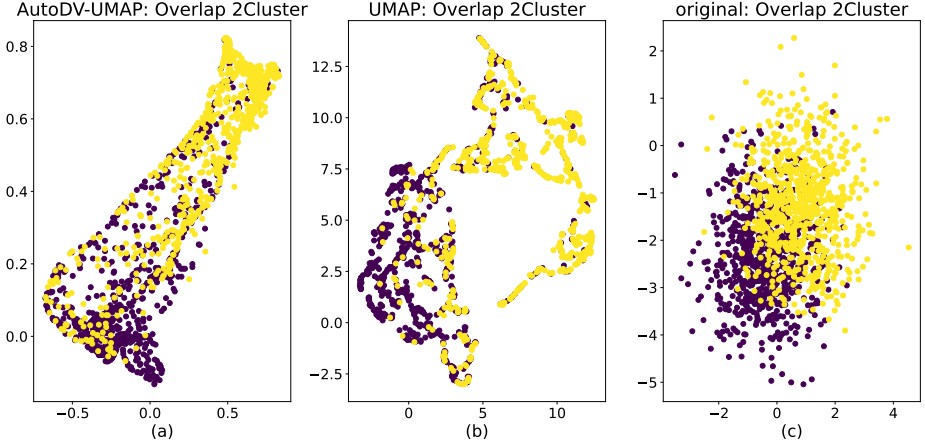

Figure 28: Visualization of 2-Cluster dataset with high level overlap using AutoDV-UMAP trained on 2-Cluster data (a) and UMAP (b). (c) is the original cluster allocation.

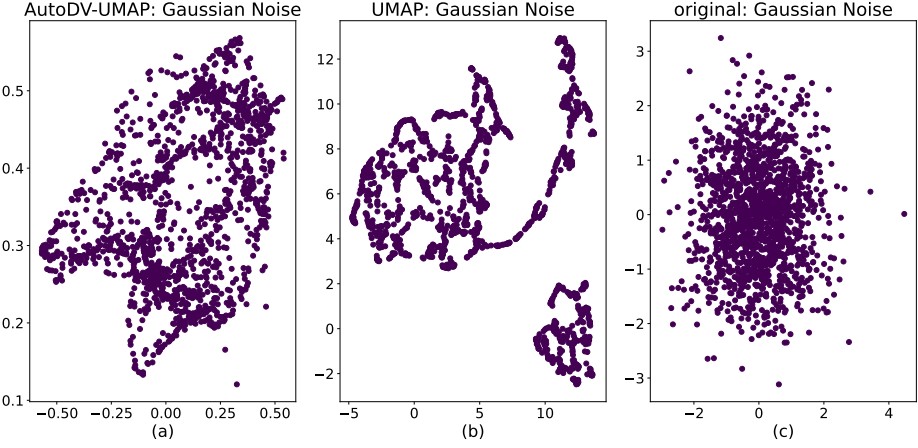

Figure 29: Visualization of 2D Gaussian noise using AutoDV-UMAP trained on 2-Cluster data (a) and UMAP (b). (c) is the original Gaussian noise allocation.

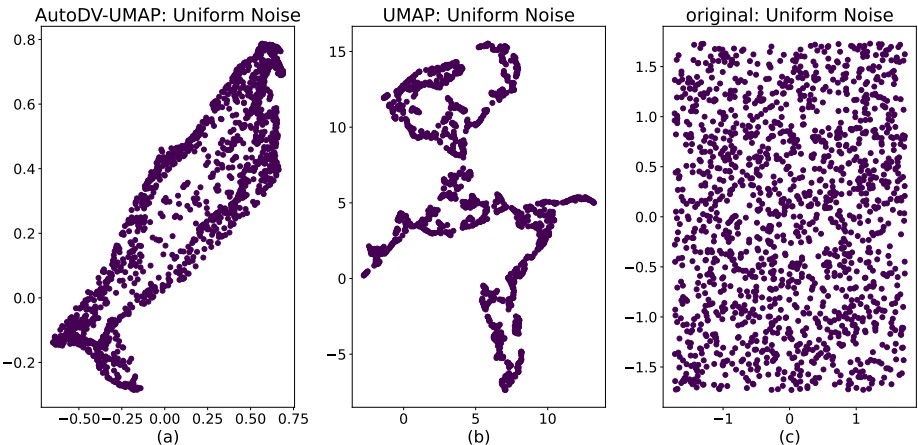

Figure 30: Visualization of uniform noise using AutoDV-UMAP trained on 2-Cluster data (a) and UMAP (b). (c) is the original uniform noise allocation.

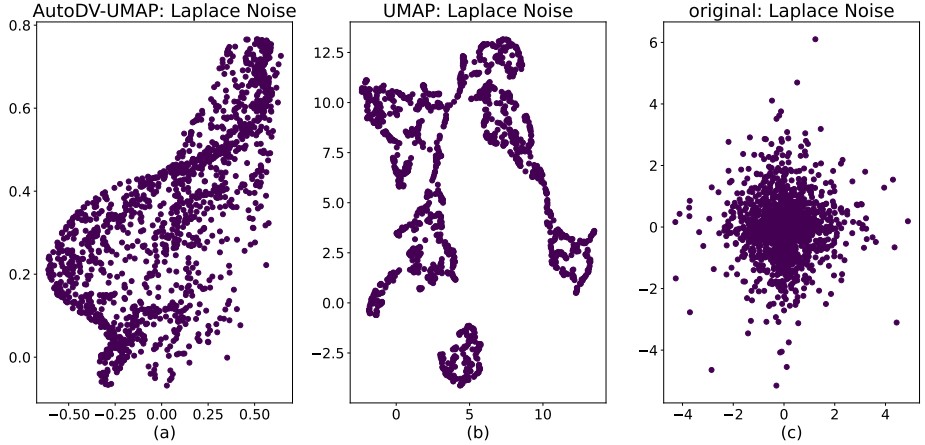

Figure 31: Visualization of Laplace noise using AutoDV-UMAP trained on 2-Cluster data (a) and UMAP (b). (c) is the original Laplace noise allocation.

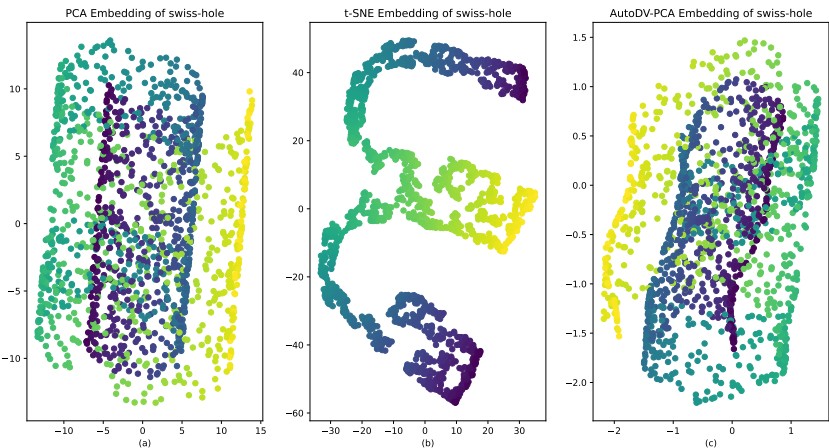

Figure 32: Visualization on a Swiss Hole dataset. (a) Performance of PCA. (b) Performance of default t-SNE. (right) Performance of AutoDV-PCA trained on Swiss Roll.

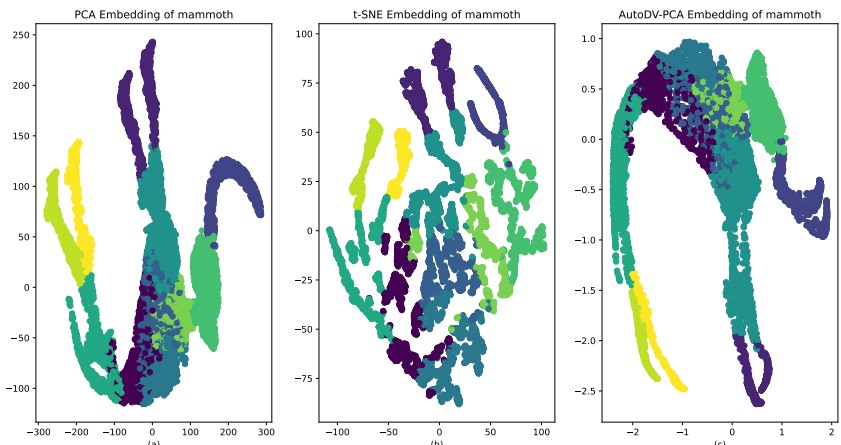

Figure 33: Visualization on Mammoth dataset. (a) Performance of PCA. (b) Performance of default t-SNE. (right) Performance of AutoDV-PCA trained on Swiss Roll.

## R    EXTEND TO LINEAR DIMENSION REDUCTION METHODS

We further explore how good AutoDV is if it is supervised by linear DR method, such as PCA. Due to limited time, we conduct the experiment using synthetic dataset similar to Q1 in Appendix Q. Specifically, we train an AutoDV-PCA model using Swiss Roll and test the performance on Swiss Hole and Mammoth datasets. The results are shown in Figure 32 and 33. It is seen that AutoDV-PCA has a similar result to the original PCA validating our proposed AutoDV framework.

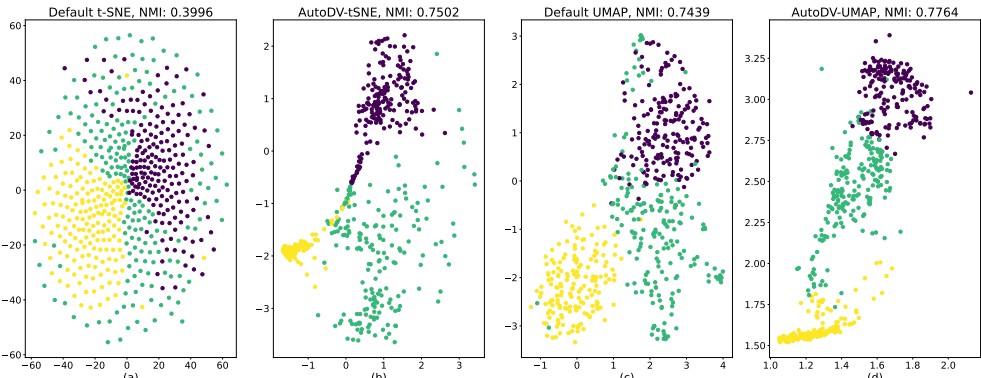

Figure 34: Visualization of Mouse Retina subset with 3 classes. (a) Default t-SNE, (b) AutoDV-tSNE, (c) Default UMAP, (d) AutoDV-UMAP.

## S  MORE VISUALIZATION ON GENE AND TABULAR DATA

We select some representative results in Gene and Tabular experiments, and compare the visualization with the default t-SNE and default UMAP. The results of Mouse Retina with 3 classes are in Figure 34. The results of PhiUSIIL with 2 classes are in Figure 35. The results of Wine with 3 classes are in Figure 36. It is seen that both AutoDV-tSNE and AutoDV-UMAP show a better cluster structure than default t-SNE and default UMAP. For datasets with more classes, we present the results of RT-IoT2022 with 12 classes and Forty Soybean Cultivars from Subsequent Harvests with 40 classes in Figure 37 and Figure 38, respectively.

In some cases, the proposed AutoDV can be better than the default t-SNE/UMAP or even than t-SNE*/UMAP*. It can be understood from 2 perspectives.

- Both t-SNE and UMAP build a single neighborhood graph (or similarity matrix) on the target dataset and optimize an embedding directly from that graph. Their behavior is therefore controlled by one neighborhood scale (perplexity or n_neighbors) and by the local sampling density of the current data, which makes the result sensitive to noise, density variations, and hyper-parameters, and limits their ability to capture complex geometric structures in real-world datasets. In contrast, AutoDV converts each training dataset into multiple k-scale graphs with different bandwidths and trains a neural network to map these multi-scale similarity patterns to a low-dimensional representation. Trained on a diverse set of datasets, this network learns a data-driven prior that reconciles different structural scales and is robust to spurious local connections. So, AutoDV yields more stable and expressive embeddings

- The local-noise experiment in Figures 24 and 25(in Appendix Q) illustrates this difference more concretely. t-SNE and UMAP treat all points symmetrically and do not distinguish "noise" from "normal" points; they only try to preserve high-dimensional neighbor relations. Local uniform noise points typically lie close to some cluster in the original space, so they enter the kNN graph as strong neighbors of the cluster points. In the t-SNE KL objective and the UMAP cross-entropy objective, breaking these high-dimensional neighbor edges incurs a large penalty, whereas accidentally adding extra neighbors is penalized much less. Therefore, it is always cheaper for default t-SNE/UMAP to pull the noisy samples into the clusters than to isolate them, which explains why the noise is visually mixed with the cluster cores in the default UMAP embeddings. In contrast, AutoDV is robust to such spurious local connections, and can correctly visualize the shifting caused by local noise. When facing datasets with more complex structure, such property can better reflect the pair-wise relative position in high dimension.

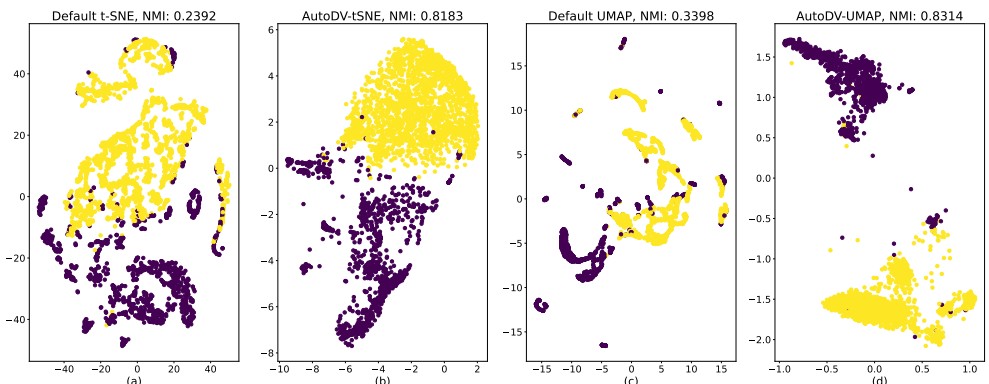

Figure 35: Visualization of PhiUSIIL with 2 classes. (a) Default t-SNE, (b) AutoDV-tSNE, (c) Default UMAP, (d) AutoDV-UMAP.

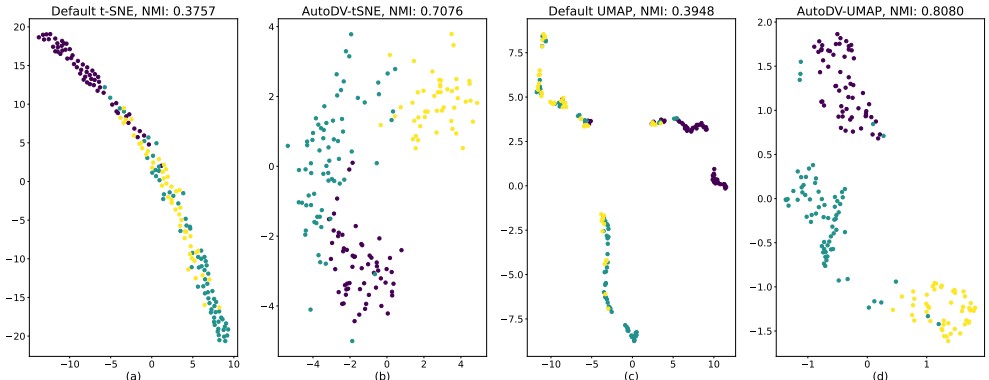

Figure 36: Visualization of Wine with 3 classes. (a) Default t-SNE, (b) AutoDV-tSNE, (c) Default UMAP, (d) AutoDV-UMAP.

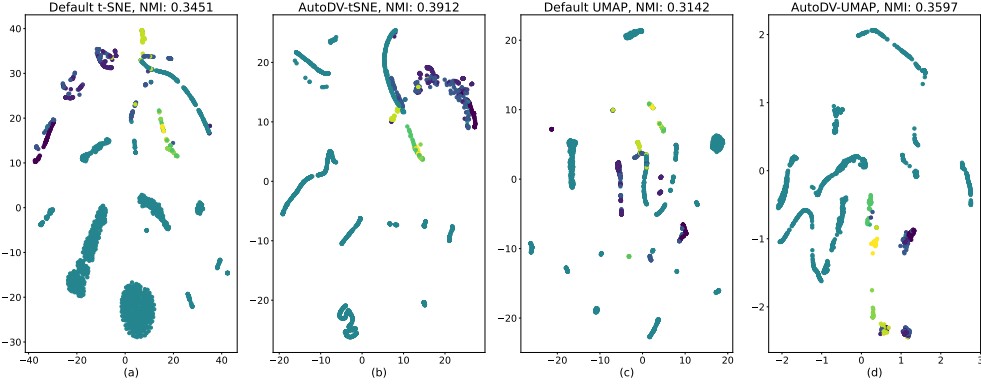

Figure 37: Visualization of RT-IoT2022 with 12 classes. (a) Default t-SNE, (b) AutoDV-tSNE, (c) Default UMAP, (d) AutoDV-UMAP.

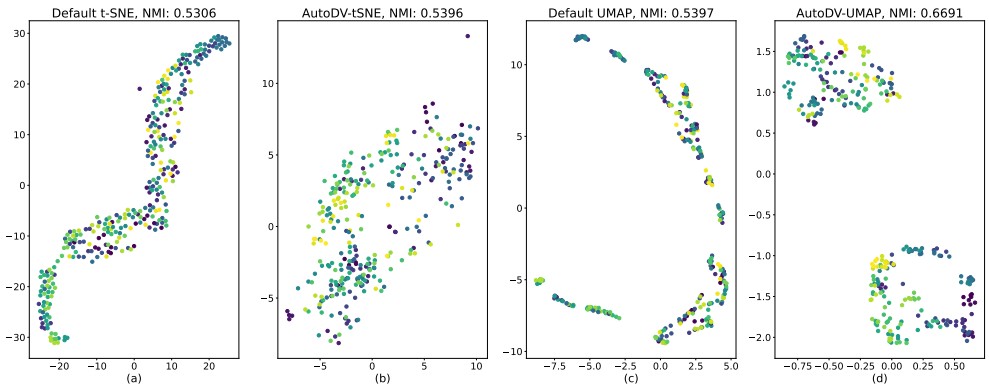

Figure 38: Visualization of Forty Soybean Cultivars from Subsequent Harvests with 40 classes. (a) Default t-SNE, (b) AutoDV-tSNE, (c) Default UMAP, (d) AutoDV-UMAP.

