# OpenReview forum: "AutoDV: An End-to-End Deep Learning Model for High-Dimensional Data Visualization"
_ICLR.cc/2026/Conference — ICLR 2026 Poster_

### Official Review · Reviewer_dtQX · 2025-10-29

**Soundness:** 3
**Presentation:** 3
**Contribution:** 3
**Rating:** 8
**Confidence:** 4

**Summary:**

This work proposes an end-to-end high-dimensionality reduction that generalises to unseen datasets with various feature sizes without retraining. Given some historical dimensionality reduction representations as the learning example, the proposed mechanism acquires meta-knowledge of how to capture the structure of high-dimensional data and preserve it in the low-dimensional representation.
To the best of the reviewer's knowledge, this ability is rarely studied, and thus, this work is novel.

**Strengths:**

This is a very solid work, supported by strong theoretical arguments and valid experiments.

The strengths of this proposed algorithm are as follows:
1. It offers an end-to-end DR mechanism that generalises across many not new data points, but new datasets. It frees users from the need to retrain the DR method.
2. It does not need any hyperparameter tuning.

**Weaknesses:**

The reviewer found no significant weaknesses in this work.

The proposed algorithm was trained only on historical low-dimensional representations from t-SNE and UMAP. The reviewer understands that this is due to the page limitation. However, the proposed work's generality will be further strengthened if the authors can train AutoDV using other DR algorithms. Specifically, t-SNE and UMAP are non-linear and unsupervised DR methods. Hence, it will be interesting to observe the generality of the AutoDV against a linear DR such as traditional PCA or MDS, and against a supervised linear DR such as LDA.

**Questions:**

1. In pg. 3 subsection 3.1, the authors wrote: \theta^*_i denotes the optimal hyper-parameters selected using y_i (the labels). However, in the experiments, the proposed AutoDV was trained using historical DV, t-SNE, and UMAP, which are unsupervised DRs that do not account for dataset labels. So the statement does not reflect the experiments well.

2. The term "historical datasets" can be confusing. The term "historical low-dimensionality representations"  is better. Please review the usage of this terminology.

3. To further show the meta-learning ability to capture the preserved structure of high-dimensional data, it will be interesting if the authors can run additional experiments against t-SNE with various perplexity values (or UMAP with various numbers of neighbors). It is of interest to observe that, for a perplexity that generates poor maps, it will still be possible for AutoDV to capture the preserved structure.

4. pg. 2, "Lack of cross-domain....": min_theta E_i|f(x_i).....| ->  min_theta E_i|f_theta(x_i).....|

5. Fig.1 is too small to see

6. Some novel DR methods may benefit the depth of this paper. Please consider including them in the discussion and for future experiments to further argue the proposed method's generality

- A. Dehghani, et al.,Credit-based self organizing maps: training deep topographic networks with minimal performance degradation, ICLR 2025, https://openreview.net/pdf?id=wMgr7wBuUo

- P. Hartono, Mixing autoencoder with classifier: conceptual data visualization, IEEE Access, Vol. 8, pp.105301 -105310 (2020) DOI: 10.1109/ACCESS.2020.2999155

- P. Hartono, P. Hollensen, T. Trappenberg, Learning-Regulated Context Relevant Topographical Map, IEEE Trans. on Neural Networks and Learning Systems, Vol. 26, No. 10, pp. 2323-2335 (2015). DOI:10.1109/TNNLS.2014.2379275

---

> ### Author Response · Authors · 2025-11-26
> **Response to Reviewer dtQX**
>
> Dear Reviewer dtQX:
>
> We greatly appreciate your recognition of our work and your encouraging comments regarding its contributions and potential impact. Your insights have been invaluable in helping us further clarify and improve the manuscript. Our responses to the weaknesses and questions are as follows.
>
> **Response to Weakness 1:** Thank you for the insightful suggestion. The proposed AutoDV is not limited by the choice of the original DR method. Due to limited time, we conduct a simplified experiment for a quick validation. Here we provide results of AutoDV-PCA trained on Swiss Roll, and tested on Swiss Hole and Mammoth in Appendix R. The link, https://anonymous.4open.science/r/AutoDV-6DD3/Extend%20to%20PCA.pdf, is for your quick access.
> It is seen that our AutoDV-PCA can obtain similar results to the original PCA.
>
>
> **Response to Question 1:** Although t-SNE and UMAP are unsupervised DRs, it truly requires the dataset labels when selecting the optimal hyper-parameters. In our method and experiment, we utilize the labels of historical datasets to search (using Bayesian Optimization) the optimal Z* with the best NMI, as described in line 369. One specific goal of AutoDV is to mimic the optimal Z* given an input dataset.
>
>
> **Response to Question 3:** For a quick validation, we select some datasets where the default t-SNE and UMAP have a poor map, while AutoDV can still produce meaningful results. They are in Appendix S. The link, https://anonymous.4open.science/r/AutoDV-6DD3/Compareded%20with%20Default.pdf, is for your quick access.
>
> **Response to Question 6:** Thank you for your suggestion. We have carefully studied the recommended research and cited these works in the updated PDF (Section 2).
>
> **Response to editorial comment (Question 2, 4, 5):** Thank you for your careful reading. We will revise "historical datasets" to "historical low-dimensionality representations". We will fix the typo on page 2. We will enlarge Figure 1 in our camera-ready version since there is one extra page.

---

> > ### Comment · Reviewer_dtQX · 2025-11-27
> > **respond to the revisions**
> >
> > Thank you for your revisions.
> >
> > The authors have significantly deepened the paper in such a short time. In particular, the additional experiments on linear dimensionality reduction (in this case, PCA) demonstrate that the proposed method captures the dimensionality-reduction characteristics of the historical low-dimensional representations it is trained on.
> >
> > The results in https://anonymous.4open.science/r/AutoDV-6DD3/Compareded%20with%20Default.pdf, where the proposed method is tested against datasets for which the default t-SNE and UMAP produce poor maps, are also interesting. Please add an explanation of why the AutoDV generated better maps than the original historical DRs.
> >
> > I am looking forward to reading the revised version of this paper.

---

> > > ### Author Response · Authors · 2025-11-27
> > >
> > > Dear Reviewer dtQX,
> > >
> > > We sincerely thank the you for the very positive and encouraging feedback. Regarding your further question on **why the AutoDV generated better maps than the original DRs**, we would like to explain it from two perspectives.
> > >
> > > * Both t-SNE and UMAP build a single neighborhood graph (or similarity matrix) on the target dataset and optimize an embedding directly from that graph. Their behavior is therefore controlled by one neighborhood scale (perplexity or n\_neighbors) and by the local sampling density of the current data, which makes the result sensitive to noise, density variations, and hyper-parameters, and limits their ability to capture complex geometric structures in real-world datasets. In contrast, AutoDV converts each training dataset into multiple k-scale graphs with different bandwidths and trains a neural network to map these multi-scale similarity patterns to a low-dimensional representation. Trained on a diverse set of datasets, this network learns a data-driven prior that reconciles different structural scales and is robust to spurious local connections. So, AutoDV yields more stable and expressive embeddings
> > >
> > > * The local-noise experiment in Figures 23–24 (in Appendix Q, https://anonymous.4open.science/r/AutoDV-6DD3/Generalization%20Boundary.pdf) illustrates this difference more concretely. t-SNE and UMAP treat all points symmetrically and do not distinguish “noise” from “normal” points; they only try to preserve high-dimensional neighbor relations. Local uniform noise points typically lie close to some cluster in the original space, so they enter the kNN graph as strong neighbors of the cluster points. In the t-SNE KL objective and the UMAP cross-entropy objective, breaking these high-dimensional neighbor edges incurs a large penalty, whereas accidentally adding extra neighbors is penalized much less. Therefore, it is always cheaper for default t-SNE/UMAP to pull the noisy samples into the clusters than to isolate them, which explains why the noise is visually mixed with the cluster cores in the default UMAP embeddings. In contrast, AutoDV is robust to such spurious local connections, and can correctly visualize the shifting caused by local noise. When facing datasets with more complex structure, such property can better reflect the pair-wise relative position in high dimension.
> > >
> > > We also add the above discussion in the Appendix S of our new revised version.

---

### Official Review · Reviewer_kZK1 · 2025-10-29

**Soundness:** 3
**Presentation:** 3
**Contribution:** 2
**Rating:** 6
**Confidence:** 4

**Summary:**

This paper addresses the practical pain points of high-dimensional visualization (HDV): methods like t-SNE and UMAP require dataset-specific hyperparameter tuning and iterative optimization, and their results vary across domains and input dimensionalities. The authors propose AutoDV, an end-to-end model trained once to produce 2D/3D embeddings for unseen datasets without per-dataset tuning. The key idea is to learn from historical, high-quality visualizations and generalize that “visualization know-how” to new data.

Technically, AutoDV first graphizes any dataset—regardless of its original feature dimension—into multiple similarity graph views (e.g., Gaussian kernels at different bandwidths). It then computes graph positional encodings and processes each view with a per-view GNN; the view outputs are concatenated and fed to a Graph Transformer, followed by an MLP that yields the final low-dimensional coordinates. Training is supervised by “optimal” target embeddings obtained from conventional HDV (e.g., t-SNE/UMAP selected via Bayesian optimization on labels/metrics). Instead of matching raw coordinates, AutoDV minimizes an affine-invariant divergence between pairwise similarity structures of its prediction and the target, eliminating rotation/translation/scale ambiguities and stabilizing learning.

The paper’s main limitations are the dependence on historical “optimal” targets (which may require labels and careful tuning to curate) and potential quadratic costs from pairwise structures (mitigated by batching/anchors). Nonetheless, AutoDV offers a compelling step toward set-and-forget visualization: a single trained model that produces stable, near-optimal embeddings for diverse new datasets without per-dataset fiddling.

**Strengths:**

1. The idea of learning a mapping from multi-view graph topology to a low-dimensional layout is novel. It decouples from raw feature dimensionality, uses affine-invariant supervision, and targets cross-dataset generalization without per-dataset tuning.

2. The authors demonstrate strong command of related work and discuss it thoroughly. They situate the approach against classical DR (PCA/MDS), modern manifold learners (t-SNE/UMAP and variants), and parametric/inductive methods, clearly motivating the gap.

3. The experiments are extensive across many datasets and modalities. Image (with pretrained features), gene-expression, and tabular benchmarks are covered with multiple metrics, offering broad evidence for robustness and generalization.

**Weaknesses:**

1. Unclear data requirements for learning from historical visualizations.

It is not specified how much and what diversity of historical visualizations are required for AutoDV to reliably replace UMAP in practice. The paper should quantify sample complexity and coverage (e.g., number of datasets, domain diversity, class balance, feature dimensionality ranges) and provide learning curves/ablation studies (performance vs. amount and variety of historical supervision).

2. Evaluation relies heavily on NMI, which can be a coarse proxy.

When high-dimensional features (or labels) are imperfect, cluster labels are not a reliable indicator of visualization quality. The paper should include neighbor-preservation metrics and rank-based fidelity measures—e.g., kNN overlap/Jaccard@k, Precision/Recall@k, Trustworthiness/Continuity, triplet preservation, and (optionally) global distortion/stress or distance rank correlations—to demonstrate that local and global neighborhood structures are faithfully preserved.

**Questions:**

1. How do you expect the user should choose AutoDV over UMAP in practice?

2. What is the neighour-preserving rate between the high and low dimensional data?

---

> ### Author Response · Authors · 2025-11-26
> **Response to Reviewer kZK1**
>
> Dear Reviewer kZK1,
>
> We sincerely appreciate your positive evaluation and valuable comments. Here is our response. We hope the newly added results can increase your confidence.
>
> **Response to Weakness 1:** We perform a series of experiments to study the impact of the training data diversity in terms of the following aspects:
> * **sample size**
> * **structural diversity**
> * **dimension**
> * **domain**
>
> The details are in Appendix O. The link, https://anonymous.4open.science/r/AutoDV-6DD3/Training%20Data%20Diversity%20Study.pdf, is for your quick access.
>
> For short, a core observation is that if the training datasets have more diverse internal geometry structures, then the model would perform well on the unseen test data potentially. Then, to make AutoDV more practical, we could consider the problem that "how to synthesize a training set containing diverse internal geometry structures?" We leave the problem for our next work.
>
> **Response to Weakness 2 and Question 2:** We select NMI as the metric for optimal $Z$* searching and performance evaluation because the cluster structure is often more important than other structures in real-world data visualization and people often want to discover clusters or groups from high-dimensional data for scientific research and engineering applications. In our paper, the three types of datasets (image, gene, and classical tabular data) are all composed of multiple clusters with true labels. Therefore, using NMI to evaluate the visualization efficacy is quite reasonable.
>
> Indeed, as you suggested, using more local or global metrics would provide a more comprehensive evaluation. Here, we report the performance of t-SNE*, UMAP*, Default t-SNE, Default UMAP, AutoDV-tSNE, and AutoDV-UMAP, on our main experiments (Image, Gene, and Tabular data) with more metrics. The results are in Appendix P. The link, https://anonymous.4open.science/r/AutoDV-6DD3/Additional%20Metrics.pdf, is for your quick access.
> It is seen that the AutoDV can obtain a better distance rank correlation and triplet preservation. For local structure preservation, our method can get comparable performance in terms of KNN classification accuracy (CA-KNN). The metric is also used in PaCMap [1].
>
> [1] Wang, Yingfan, et al. "Understanding how dimension reduction tools work: an empirical approach to deciphering t-SNE, UMAP, TriMAP, and PaCMAP for data visualization." JMLR (2021).
>
> **Response to Question 1:** Based on our experimental results, we find that AutoDV obtains even better performance than the optimal t-SNE/UMAP on Gene and tabular data in terms of NMI. These real-world datasets are often more complex and noisy than images (features extracted by CLIP). These datasets demand even more from HDV methods for visual analysis. In many scientific research fields, researchers expect to see the classification or cluster structure in the visualization results for a better outcome. For example, in genetic research, researcher usually discovers different genome types. Visualization with better cluster structure can help researchers discover new genetic patterns or genome types. So, we believe AutoDV can help such a scientific research field.
>
> We additionally visualize more results on Gene and Tabular datasets in Appendix S, where the AutoDV outperforms t-SNE and UMAP. The link, https://anonymous.4open.science/r/AutoDV-6DD3/Compareded%20with%20Default.pdf, is for your quick access.

---

### Official Review · Reviewer_gpJd · 2025-10-30

**Soundness:** 2
**Presentation:** 2
**Contribution:** 2
**Rating:** 4
**Confidence:** 3

**Summary:**

This paper introduces AutoDV, a high-dimensional data visualization algorithm which leverages a graph transformer network to embed datasets into lower dimensional spaces. Datasets are first converted into a similarity graph, which is passed into a graph transformer, which is trained to predict the ground truth TSNE/UMAP values under optimal hyperparameters. The paper highlights two challenges in performing this end-to-end modeling task: different sizes/dimensions for the input points, and the presence of multiple ground truths for each underlying input (under rotation,  scaling etc.). To solve the first challenge, the paper ignores the dimensionality of the points, and relies on taking as input a similarity graph under a gaussian kernel function (Following Long, 2015). This makes it possible to apply a GNN to encode the graph, and reproduce it in a lower dimension. To solve the second challenge, the paper introduces an affine-invariant loss function, which attempts to align the pairwise squared similarity matrices in the lower dimension under a Bregman divergence (using either KL divergence for T-SNE or a structural similarity, given in Eq. 8, for UMAP).

The paper is evaluated across several image datasets (MNIST, Fashion-MNIST, CIFAR-10), as well as several datasets for bioinformatics (Baron Huntman, Mouse Retina, Cambell and PBMC68K), and tabular data (UCI Machine Learning).  Evaluations of the NMI and Silhouette Coefficient show that AutoDV approaches, and sometimes exceeds the representation performance of the baseline approach across almost all results, while being somewhat faster than the existing methods (Particularly an un-optimized TSNE).

**Strengths:**

This paper tackles an interesting problem in data visualization: can you learn a function which end to end approximates t-SNE or UMAP (or really any of the dimensionality reduction methods), which doesn't require tuning the hyperparameters. AutoDV is a step towards that: there's no hyperparameter selection, and it generates what are numerically better visualizations of underlying data. The theoretical stability results are nice (albeit somewhat weak), and the method is much more efficient than un-optimized t-SNE, which means that there are potential gains here as well.

**Weaknesses:**

While there are some strengths, the paper does have significant weaknesses:
- For a paper on data visualization, there is very little actual visualization in the paper. It would be good to demonstrate qualitative results for several different datasets (Ideally beyond those in Figure 4), which honestly do not look remarkably similar to the underlying optimal TSNE/UMAP examples (even under transformation). What about visualizing some more classic datasets (swiss roll, lines, etc.)?
- It's also not clear here that the learned parameters are that much better than defaults (For example, for Figure 4, how well does a single learned default parameter perform?) - Could you have a similar baseline, perhaps, that just uses the TSNE* data to inductively learn the optimal dataset-specific parameters?
- It's not entirely clear if this model will generalize to new datasets, and the paper does little . While the paper does show that it generalizes from MNIST and FMNIST to CIFAR, it's not entirely clear how far that generalization goes: will a pre-trained AutoDV model generalize from CIFAR to a Genetic dataset? Or from CIFAR to something like the swiss roll? It would be really good for this paper to explore the extent of generalization - how far away to the datasets have to be structurally before the models start failing (CIFAR 10 has 10 classes, for example, which is shared with MNIST). There's also no datasets here that are not classification centric (i.e. all datasets have some larger class structure), which means that it's hard to tell if an AutoDV model will generalize between underlying manifold structures.
- There's a pretty large limitation in that the model can only handle 3000 point datasets (L364). It's a proof of concept, but it seems pretty bad, especially given the potential scalability of UMAP/TSNE.
- There's not really much analysis of the failure cases of the model - are there any? How does the model perform in situations where there are high-noise, or overlapping clusters, or even on random noise?
- There are some really confusing wordings in this paper, which make it quite challenging to read, for example, the following sentence: "To allow a deep neural network to accept different input dimensionalities directly is barely possible, which is what the existing parametric data visualization models suffer from." which motivates the design of the paper.
- The paper is lacking some fundamental motivation/ablations: Why were the base architectural choices made? Are there scaling properties to the AutoDV metric? The practical choices for $\mathcal{K}_\psi$, and particularly $\psi$ are not particularly well motivated - and no alternatives are explored. There's also no ablations on the different loss components (with/without something affine aware, with/without sign ambiguity, etc.).

**Questions:**

- How are the optimal parameters for TSNE/UMAP chosen (I might have missed this in the appendix)? It seems like while AutoDV is trained to approximate these methods, the optimal parameter choice chan highly influence the final outcomes.
- Are there any scaling properties? Does the AutoDV model have to be a particular size to achieve this performance?
- Does the model generalize beyond classification datasets? How much does the underlying manifold have to change in order to break the hyperparameter approximation of this method?
- Is this any different than just learning good initial hyperparameters for TSNE/UMAP? Does it outperform such methods?

---

> ### Author Response · Authors · 2025-11-26
> **Response to Reviewer gpJd**
>
> Dear Review gpJd,
>
> Thank you for your insightful comments. Here are our responses.
>
> **Response to Weakness 1 (more visualization results):** We present more visualization of the training data and testing data in Appendix M (page 24 - 26), where our model can successfully transfer the visualization capability from MNIST and FMNIST to CIFAR10. For more visualization results on other datasets, please see our response to Weakness 2, Weakness 3 and Weakness 5.
>
>
> **Response to Weakness 2 (compared with default):** For the comparison to the default value, we extend Tables 2, 3, and 4 with the default t-SNE and UMAP. Our method performs better than the default on Gene and Tabular datasets. For qualitative comparison, we show more results in Appendix S, where AutoDV can obtain better results. The link, https://anonymous.4open.science/r/AutoDV-6DD3/Compareded%20with%20Default.pdf, is for your quick access.
> We hope these results can solve your concern about the comparison with the defaults.
>
> **Response to Weakness 2 (learning optimal parameters):** Thank you for your insightful comments. If we understand your question correctly, we have a baseline method that predicts the optimal perplexity of t-SNE instead of the optimal embedding. So, we use the optimal perplexity to train an AutoDV model. Then, the training loss is changed to an MSE loss, and the input graphs are mapped to a single value. We perform the experiment on Gene dataset. There are 3 drawbacks when doing so.
>
> * In the experiments, we find the training loss and test loss are quite high and hard to converge. This may be because there is a huge information loss when reducing the nodes features into a single vector.
>
> * The performance on Gene data is quite low with 0.1610 in terms of NMI.
>
> * It still requires re-running the t-SNE algorithm when visualizing a new dataset, which is time-consuming.
>
>
> **Response to Weakness 3 (generalization to new datasets):** We respectfully argue that we did lots of experiments to validate the generalization ability of AutoDV. We reported the transferability of AutoDV in Figure 3, where AutoDV can generalize from MNIST and FMNIST to the Genetic dataset and the Tabular dataset, and vice versa. It is worth mentioning that the proposed AutoDV outperforms all existing inductive HDV methods, such as AE, parametric UMAP, inductive t-SNE, and inductive UMAP.
>
> For more transferability of the AutoDV test, we provide more transferability tests from MNIST and FMNIST to some classic datasets, such as Circle, Lineage, Mammoth, ... The results are in Appendix Q. The link, https://anonymous.4open.science/r/AutoDV-6DD3/Generalization%20Boundary.pdf, is for your quick access.
>
> **Response to Weakness 3 (Generalization Boundary), Weakness 5 (Failure Cases) and Question 3:**  In our paper, we select the classification-centric dataset and NMI metric to find the optimal $Z^*$ and to evaluate, because we believe the clustering structure is more important in broader scientific research. For example, in genetic research, researcher usually discovers different genome types. Here, we conduct a very comprehensive generalization boundary and failure cases analysis.
>
> * First, we found that our AutoDV model trained by Image data cannot directly transfer to a manifold structure like the Mammoth dataset. It is because such geometric structures never appear during the training. The Mammoth dataset meets the generalization boundary.
>
> * However, the proposed AutoDV framework is not limited by the classification-centric datasets. In the training phase, users can feed optimal $Z^*$, which is selected by different metrics for different analytic objectives (as we described in line 366). Due to limited time, we provide a simple experiment to show the generalization between manifold structures. So, we further train an AutoDV-tSNE model using embeddings of Swiss Roll and Swiss Hole from t-SNE. The results show that our method can successfully generalize between manifold structures if similar manifold structures are in the training data.
>
> * We analyze the failure cases on some synthetic datasets. It is surprising to find that our models are more resistant to the global noise. The noisy situation does not cause our model to fail to output meaningful results.
>
> * The results are in Appendix Q. The link, https://anonymous.4open.science/r/AutoDV-6DD3/Generalization%20Boundary.pdf, is for your quick access.
>
> * To sum up, the most important key to affect the generalization ability of AutoDV is the training data structural diversity. We also provide an ablation study in Appendix O.

---

> ### Author Response · Authors · 2025-11-26
> **Response to Reviewer gpJd (Part II)**
>
> **Response to Weakness 4 (sample size):** We limit the subset size to 3000 due to the limited video memory during training. However, at the inference stage, AutoDV can handle larger datasets in two ways as we described in Section 4.4. One is directly feeding a large graph into the model, and it can also work well, as reported in Figure 5. The other is to use a batch-based method with a fixed anchor detailed in Appendix E.
>
>
> **Response to Weakness 6 (wordings):** Thank you for your careful reading. What we would like to express is that, in existing parametric data visualization methods, once the neural networks are trained, the input dimension is fixed and the meaning of the input is fixed. A new data with a different dimension or feature meaning cannot be fed to the neural network to generate a meaningful embedding. Hence, we represent a dataset as a graph, regardless of the original feature dimensionality. We have revised the related wording for better understanding.
>
>
> **Response to Weakness 7 (ablation study):** For the ablation study, we performed ablation studies on different loss choices, different sign-flipping strategies, different positional encoding types, different choices of k, and different dimensions for the positional encoding in Appendix L.
>
> For the motivation of model design, we believe our design is reasonable and adequate. To tackle the cross-dimension generalization challenge, we transform the input dataset into a graph and use GNN and a graph transformer to process it. To solve the lack of node feature issue, we introduce the positional encoding. To tackle the one-to-many problem in training, we utilize an affine invariant loss. For each of the technical issues we met, we effectively and reasonably solved them.
>
> For the loss design, the designing principle is to align both local structure and global structure between $\hat{Z}$ and $Z^*$. Initially, we think we should use the same type of loss function as the original HDV method. It is intuitive to understand because it is to best mirror the optimization process of the original HDV method, and we obtain a good performance for AutoDV-tSNE (which uses KL Divergence). However, when we use the same training loss as the UMAP (which is using BCE), AutoDV-UMAP cannot work well. The results can be found in Appendix L. Suddenly, we noticed that it is not necessary to keep the loss function the same as the original HDV method. So, we designed a current l2-norm based loss with local + global alignment. Due to limited time and resources, we did not try the l2-norm based loss on AutoDV-tSNE, but we believe it would work well too.
>
> **Response to Question 1:** The optimal hyper-parameter and the corresponding DV embedding are obtained through Bayesian optimization. One can also search for the optimal hyper-parameter through a grid search manner. Finally, a hyper-parameter with the highest NMI will be selected, and the corresponding embedding is used for training.
>
> Your understanding is correct. The embedding for training can highly influence the model training. Luckily, the "optimal" embedding can always be found in the historical datasets. In addition, AutoDV can be trained for different demands. For example, if one would like to produce visualization results reflecting the cluster structure, he should use the "optimal" embedding with the highest NMI to train the model. If one cares about the manifold structure, they should use the "optimal" embedding with the best local preservation.
>
> **Response to Question 2 and Weakness 7 (scaling property):**
> We here provide a set of results of AutoDV-UMAP using different numbers of training sets in [10, 50, 500, 1000].
>
> | **training size** | **10**   | **50**   | **500**  | **1000** | **UMAP*** |
> |-------------------|----------|----------|----------|----------|-----------|
> | test NMI          | 57.19    | 58.72    | 73.28    | 74.17    | 80.45     |
> | test SC           | 51.11    | 52.01    | 70.41    | 71.3     | 68.54     |
>
> In the results, we see that if the training sample size becomes large, the performance on a fixed testing set will become better.
>
> However, the amount of training data may not be the only factor that affects the model's performance, though the more data trained, the stronger the model, according to the empirical success of large language models. Here, the diversity of the training data is also increased. So, we believe the scaling property of AutoDv is that the more diverse the training data, the stronger the generalization ability.
>
> As for the scaling properties in terms of model size, we observe that if we have a deeper graph transformer, we can get a better performance. Our current setting is a trade-off between training efficiency and video memory limitation.

---

> ### Author Response · Authors · 2025-11-26
> **Response to Reviewer gpJd (Part III)**
>
> **Response to Question 4:** Compared with t-SNE/UMAP, our method have 2 advantages.
>
> * It does not require hyper-parameter selection when faced with **new unlabeled** datasets. It is hard for t-SNE/UMAP to tune the hyper-parameter if the dataset is unlabeled. Without a labeled validation set, one can never know how well the result is. This challenge widely exists not only in high-dimension data visualization tasks [1] but also in clustering [2], anomaly detection [3], and out-of-distribution [4] research.
>
> * There is no iterative optimization (re-training) when faced with new datasets. t-SNE/UMAP needs to run an optimization process every time when it faces new datasets. This is a crucial computation overhead. Especially, if one would like to search for the optimal hyper-parameter, the number of re-training times will increase. The total waiting time is unacceptable. We also analyzed and compared the running time in Table 5 in Section 4.3.
>
> * Moreover, in our experiment, AutoDV even outperforms the optimal t-SNE/UMAP in Gene datasets and Tabular datasets.
>
> [1] Lin et al. Calibrating dimension reduction hyperparameters in the presence of noise. PLOS Computational Biology (2024).
>
> [2] Fan et. al. A simple approach to automated spectral clustering. NeuIPS 2022
>
> [3] Zhao et. al. Automatic unsupervised outlier model selection. NeuIPS 2021
>
> [4] Qin et. al. MetaOOD: Automatic Selection of OOD Detection Models. ICLR 2025

---

### Official Review · Reviewer_hD97 · 2025-10-31

**Soundness:** 1
**Presentation:** 1
**Contribution:** 1
**Rating:** 2
**Confidence:** 3

**Summary:**

The authors propose an automated way of learning to embed whole datasets similarly to a specific prescribed embedding algorithm, without the need of choosing its hyperparameters. The hyperparameters, used to generate each embedding, are different for each dataset and are chosen based on the labels of the dataset. These labels are used the validate the quality of the embedding, and are not used throughout the embedding process. Then, at inference time, the model produces an embedding that should correspond to an embedding with hyperparameters selected in the same manner for this dataset.

**Strengths:**

The fact that the neural network was able to produce an embedding based on the test data is interesting.

**Weaknesses:**

1. The terminology is not consistent with the manifold learning field. The authors propose to generate "k -view graphs", where in each of the k graphs they use a different bandwidth parameter. There are a whole field that deals with multi-view dimensionality reduction, where each view considers a different source. Here on the other hand, we get the same source. The more suitable term here will be multi-scale, as the algorithm generates the similarity matrix over the data in multiple scales of bandwidths.

2.  Paper aims to solve too many tasks altogether. If I understand correctly, the paper aims to solve two problems together, including: (A) Extract good hyperparameters (and their corresponding bandwidth/s) for a given dataset, and (B) Generate an embedding based on these bandwidths. Each of these tasks are non-trivial, and I would expect the authors to show their superiority on each task independently.

3. Quantification metrics. The training data includes datasets along with their corresponding embeddings, which are the result of applying the same embedding algorithm with different hyper-parameters. These hyper-parameters were chosen based on some supervised metric. I wonder why the authors do not compare the embeddings they got from test data with the true embedding (of the embedding algorithm) using its "optimal parameters", one approach to do so is through their graphs.

4. Unsupervised bandwidth selection. The problem of unsupervised bandwidth selection is studied in the context of different embedding algorithms, and in other related contexts such as KDE. I did not see any discussion about this field.

5. Missing clarification about choices related to the neural network. For example, why is the SVD positional encoding good for this case, and why do embedding algorithms such as the ones considered in this paper relate to them. Going even further than that, you can then ask how this encoding corresponds to the initialization of the selected embedding algorithm.

6. The paper will really benefit from generating even a synthetic dataset, which will be easy to understand, and then the generated embeddings along with the scores can better illustrate the algorithm's output more clearly.

7. Theoretical Configuration. The configuration of the problem is not well defined.

**Questions:**

1. NMI. How did you choose the number of K in K-means when you calculated the NMI? I am wondering whether you allowed K to be much larger than the number of clusters as the problem is related to Manifold Learning and not only to Clustering.

2. Same bandwidths used across all datasets. The optimal bandwidths may vary in orders of magnitudes between different datasets (and even within the same dataset). It can depend on all kind of factors, like the number of samples and the type of the data (e.g., curvature, density) . In a lot of cases, even z-scoring wouldn't remove this phenomena. How did you tackle this problem, or maybe this problem did not occur in your data as it had very similar characteristics (like in the image data that used the clip embedding space) ?

3. How many samples are there in each subset within the training data?

---

> ### Author Response · Authors · 2025-11-26
> **Response to Reviewer hD97 (Part I)**
>
> Dear Reviewer HD97,
>
> We sincerely thank you for your invaluable comments. Our responses to your comments are as follows.
>
> **Response to Weakness 1 (terminology):** Thank you for your insightful suggestion. We have revised the terms "multi-view" and "k-view" into "multi-scale" and "k-scale" in our paper, respectively, to avoid confusion in the manifold learning field.
>
> **Response to Weakness 2 (task definition):** We would like to highlight our motivation and problem definition here again to avoid any possible misunderstandings.
>
> * Our work is motivated by the following three facts:
>     * Existing unsupervised HDV methods face the hyperparameter tuning challenge. Hyperparameters in supervised tasks are usually tuned by a **labeled** validation set. In contrast, the validation set in the unsupervised learning task is still **unlabeled** or does not even exist. It is hard to evaluate whether the HDV result is good or not. We also illustrate the hyper-parameter sensitivity results in Fig. 1.
>
>     * Popular HDV methods, such as t-SNE and UMAP, require pre-dataset iterative optimization (re-training) and pre-dataset hyper-parameter tuning. We believe this is a significant running time overhead.
>
>     * Existing parametric HDV methods lack cross-domain and cross-dimensionality generalization ability. Parametric HDV methods are usually deep neural network-based. If the newly input data is from a different source or different dimensionality, these methods, such as AE-based, will fail to produce valid DV embeddings.
>
> * To tackle the challenges in the three facts, we aim to
> **establish a single, end-to-end deep learning model to visualize any high-dimensional data with different sizes and dimensionalities**
> As you can see, this is our task, which is a single one. We call it AutoDV.
> The AutoDV problem is defined in line 133. Then, the model is trained and supervised by the historical datasets and their optimal DV embedding. By doing so, no hyper-parameter tuning is required when meeting new unseen input datasets. Hence, we avoid the hyperparameter selection challenge.
>
> * **Regarding the comments:**
>     * The goal of AutoDV is not to select the best hyperparameter (nor the bandwidth) for a specific HDV method. In contrast, we generate an embedding in an end-to-end manner, where no hyperparameter selection is involved during inference.
>
>     * The bandwidth or the "multi-scale bandwidth" is part of model design, and it is fixed during training and testing. There is also no bandwidth selection. Instead, we fuse the multi-scale structural feature to enhance the model's capability.
>
> **Response to Weakness 3 (metrics):** Your understanding of how we obtain the "optimal" embedding of the training data is correct.
>    * For each dataset, including the test data, we select the best hyper-parameter through Bayesian Optimization. The corresponding embedding is considered the true embedding. We actually did compare our results with the "true embedding".
>    * In our results in Tables 2,3, and 4, we denote the corresponding results as **t-SNE*** and **UMAP*** (also described in line 353 in the initial submission). These two results are also considered as the ground-truth performance of our methods.
>    * Surprisingly, we can obtain even better results on the Gene dataset and the UCI tabular dataset.  It indicates our methods can better capture the structural information from the high-dimensional space.
>
> **Response to Weakness 4 (bandwidth):** As we discussed in the Response to Weakness 2, the problem we aim to solve and the method we proposed are not related to the bandwidth selection. The bandwidth in our design is set adaptively using a median value (detailed in Response to Question 2).  The multi-scale bandwidth fusion strategy is a part of the design to better capture the structural information in the high-dimensional space. We believe the multi-scale bandwidth design is helpful because traditional HDV methods, such as t-SNE, only use a single bandwidth (controlled by perplexity), which accepts less information than ours.

---

> ### Author Response · Authors · 2025-11-26
> **Response to Reviewer hD97 (Part II)**
>
> **Response to Weakness 5 (network choice):** The network design is from the graph learning field. As we described in line 200, the graph neural network often requires node features. In our methods, since we transform a dataset into a graph, there is no node feature. So, we adopt the graph positional encoding as the node feature. For the choice of the positional encoding, its optimal design is still an open problem in graph learning, and many recent works explore different spectral and structural PEs [1]. We use SVD PE because it performs the most stable and the best through our ablation study in Appendix L (page 21, line 1122).
>
> To answer how the encoding is related to the HDV algorithm and why the encoding can be set as the input of the model, one common knowledge is that the spectral analysis of a graph often reveals the structural pattern in a graph [2].
>
> [1] Black, Mitchell, et al, Comparing graph transformers via positional encodings, ICML 2024
>
> [2] Chung, Fan RK. Spectral graph theory. Vol. 92. American Mathematical Soc., 1997.
>
> **Response to Weakness 6 (synthetic data):** Thanks for your suggestion. As required by other reviews, too, we conduct a series of experiments on synthetic datasets such as Swiss Roll, mammoth, cluster, circle, etc. These results can be evaluated qualitatively. Here are some findings.
>
> * We directly feed the synthetic datasets to AutoDV-tSNE and AutoDV-UMAP trained by Image data. Our results can outperform t-SNE and UMAP in Circle and Lineage datasets. The model initially cannot generalize well to Mammoth. It is because the manifold structure never appears during the training.
>
> * However, after we train AutoDV-tSNE on datasets focusing on manifold structure, our method can also perform well on Swiss roll, Swiss Hole, and Mammoth.
>
> * In addition, we find our method is resistant to the global noise while t-SNE and UMAP not. Details are in Appendix Q. The link, https://anonymous.4open.science/r/AutoDV-6DD3/Generalization%20Boundary.pdf, is for your quick access
>
> * We replace the supervision signal from t-SNE/UMAP to PCA, i.e. the "optimal" embedding for training is the result of PCA. In the results, our AutoDV-PCA has a similar behavior to the original PCA and can successfully generalize to an unseen dataset such as mammoth. Details are in the Appendix R. The link, https://anonymous.4open.science/r/AutoDV-6DD3/Extend%20to%20PCA.pdf, is for your quick access
>
> * We study how the training data diversity affects the model performance using synthetic datasets with multiple clusters. We find that the more structural diversity in the training data, the better the performance. Details are in the Appendix O. The link, https://anonymous.4open.science/r/AutoDV-6DD3/Training%20Data%20Diversity%20Study.pdf, is for your quick access

---

> ### Author Response · Authors · 2025-11-26
> **Response to Reviewer hD97 (Part III)**
>
> **Response to Weakness 7 (theoretical configuration):**
> Thanks for pointing this out. However, since we are a little bit confused about this comment, we have to answer your question from two different points of view.
>    * If you mean the configuration of our AutoDV problem, we would like to highlight our motivation for the AutoDV problem.
>
>        * In realistic workflows, we often need an **inductive** HDV model that can map a **new** dataset to a stable low-dimensional layout **without per-dataset re-optimization**. Tuning t-SNE/UMAP for every new dataset is time-consuming and sensitive to hyperparameters; moreover, new datasets can differ in both sample size $N_i$ and ambient dimension $d_i$, making batch retraining impractical. Our goal is to learn a single end-to-end function $f_\phi$ that (i) accepts datasets of varying $N_i$ and $d_i$, (ii) respects visualization invariances (translation/rotation/scale), and (iii) approximates the layout delivered by a trusted HDV oracle for that dataset. This turns HDV from a ``per-dataset procedure'' into a reusable mapping, enabling fast deployment and consistent behavior across domains.
>
>         * **Configuration**: We are given $L$ historical datasets $\{(X_i,y_i)\}_ {i=1}^L$ drawn from diverse domains, where $X_i\in\mathbb{R}^{N_i\times d_i}$ (sizes $N_i$ and dimensions $d_i$ may differ) and $y_i$ are labels used **only** to select the oracle hyperparameters $\theta_i^\star$. For each $i$, an oracle HDV procedure $\mathcal{A}_ {\theta}$ (e.g., t-SNE/UMAP with tuned $\theta$) produces a target embedding $Z_i^\star=\mathcal{A}_ {\theta_i^\star}(X_i)\in\mathbb{R}^{N_ i\times d'}$. We train a single parametric mapping $X \mapsto f_\phi(X) \in \mathbb{R}^{N\times d'}$ that can ingest any $X$ (with arbitrary $N$ and $d$) by internally using size-agnostic operations (e.g., graph construction + permutation-invariant aggregation).
>         At test time, for a new dataset $X_{\text{new}}$ the model outputs $Z_ {\text{new}}=f_\phi(X_{\text{new}})$ that should be **close**, up to visualization invariances, to the oracle layout $Z_ {\text{new}}^\star=\mathcal{A}_ {\theta^\star}(X_{\text{new}})$.
>
>         * **Remarks**: (i) Labels $y_i$ are used only to select $\theta_i^\star$ for the oracle; $f_\phi$ is trained on $(X_i,Z_i^\star)$ pairs and does not require labels at test time.(ii) The output dimension $d\in\{2,3\}$ is fixed across datasets; $N_i$ and $d_i$ may vary arbitrarily. (iii) The graph construction and pooling inside $f_\phi$ ensure permutation invariance and size-agnostic processing. (iv) Our robustness bound (Sec. 3.5) then quantifies how small input perturbations propagate to the embedding distance.
>
>
>    * If you mean the configuration of our theorem, we would like to highlight our motivation for the theorem. Our theorem analyzes a practical question: how much does the embedding change when the input data are slightly perturbed (e.g., mild noise, a few outliers, or a small train–test shift)? We quantify visualization drift as the distance between these two embeddings and bound it by Eq. (9). The bound shows that drift grows proportionally with input perturbation and is moderated by three existing components of our model: (i) the smoothness of the network, (ii) the eigen property of the graphs across scales, and (iii) the number of scales (the worst scale dominates). Our ablation study in Table 9 of Appendix L also shows that a large number of scales does not improve the performance. The theorem also indicates that, if a new dataset is similar to a training dataset, the visualizations should be similar too. This guarantees the generalization ability of AutoDV to some extent.

---

> ### Author Response · Authors · 2025-11-26
> **Response to Reviewer hD97 (Part IV)**
>
> **Response to Question 1:** We use NMI for performance evaluation and optimal embedding selection for the historical dataset. In both cases, the ground truth label of the dataset can be accessed. So, the K in K-means is set to the number of classes in the ground truth label. If K is set larger or smaller, then we cannot correctly perceive the true performance of the embedding.
>
> **Response to Question 2:** As we described in Appendix J (page 21, line 1104), the actual implementation of the similarity matrix $S$ construction (Eq. 2 in line 191) is based on an adaptively chosen bandwidth. In detail, $\gamma^ {(j)}=coef_ j \times median(D)$, where $coef_ j$ is a constant coefficient, and $D$ is the pair-wise distance matrix of the dataset with $D[u, v]=\|X[u] -X[v]\|$. By doing this, the calculation process of $S$ is guaranteed to be rotation, translation, and scaling invariant. It realizes an adaptive choice of gamma when meeting different input data.
>
> **Response to Question 3:** Due to the limited video memory during the training, we randomly downsample the datasets to
> generate several subsets if the number of points in the dataset is greater than 3000.  The maximum number of points of subsets is 3000. The minimum number of points of a subset is 100. The details of subset sampling are listed in Appendix I. Although the maximum number of samples in each subset during training is 3000, it can still perform well if the testing set has 20000 samples using the proposed batch-based method, as the results in Figure 5 of Appendix E.

---

### Author Response · Authors · 2025-11-26
**Summary of Response to Reviewers**

Dear Reviewers,

We sincerely thank you for all your valuable comments. We provide 25 new figures and 4 new table in our responses. These results include a series of ablation studies on the training diversity, more visualization results of AutoDV-t-SNE/UMAP compared with the default t-SNE/UMAP, discussions on the generalization boundary and failure cases, extended results with more metrics, and extended results of AutoDV trained by PCA. All new results can be found in the appendices of the new submission. We also provide an anonymous link for your quick access, https://anonymous.4open.science/r/AutoDV-6DD3/. For your convenience, we also summarize the training, test data, and short performance description of each figure and table, where Table 10-13 and Figure 12-37 are new results in our recent revision.

---

> ### Author Response · Authors · 2025-11-26
>
> | **Table/Figure**| **Training Data** | **Test Data**| **Performance** |
> | ----------------- | ---------------------------------------------------------------------------------------------------------------------------------------- | -------------------------------------------- | ------------------------------------------------------------ |
> | Table 2, Table 11 | 500 subsets from MNIST and FMNSIT | 50 subsets from CIFAR10| AutoDV outperforms all parametric HDV methods |
> | Table 3, Table 12 | 2918 subsets from Baron Human, Campbel, and PBMC68k | 113 subsets from Mouse Retina| AutoDV performs even better than t-SNE and UMAP |
> | Table 4, Table 13 | 363 subsets from 26 tabular datasets| 156 subsets from 11 datasets | AutoDV performs even better than t-SNE and UMAP |
> | Figure 4| 500 subsets from MNIST and FMNSIT | 2 subsets from CIFAR10 | AutoDV-UMAP is comparable with UMAP*|
> | Figure 5| 500 subsets from MNIST and FMNSIT | 2 subsets from CIFAR10 | AutoDV can be extend to show large datasets |
> | Figure 7| 5 datasets: MNIST with class labels in [0, 1, 2, 3, 4], MNIST with class labels in [5, 6, 7, 8, 9], FMNIST with class labels in [0, 1, 2, 3, 4], FMNIST with class labels in [5, 6, 7, 8, 9], and CIFAR10 with class labels in [0, 1, 2, 3, 4]. | CIFAR10 with class labels in [5, 6, 7, 8, 9] | Ablation study on PE type, loss design, and sign flipping. Our choice is the best.|
> | Table 7 | The same as above | The same as above| Ablation study on PE dimension. Our choice is the best. |
> | Table 9 | The same as above | The same as above| Ablation study on the number of bandwidth fusion. Our choice is the best. |
> | Figure 8, 9 | 500 subsets from MNIST and FMNSIT | 7 subsets from the training data | AutoDV is comparable to t-SNE/UMAP|
> | Figure 10, 11 | 500 subsets from MNIST and FMNSIT | 7 subsets from CIFAR10 | AutoDV is comparable to or better than t-SNE/UMAP |
> | Table 10| Different number of subsets from MNIST and FMNSIT | 50 subsets from CIFAR10| Ablation study on training data size. The more training data, the better the performance. |
> | Figure 12 | Different diversity of syntactic cluster data. Only 10 datasets in training | 5 9-cluster datasets | Ablation study on training structural diversity. The more diverse structure of training data, the better the performance. |
> | Figure 13 | Different number of [1-cluster, 2-cluster] data. The structure diversity is fixed.| 5 9-cluster datasets | Ablation study on training sample size with structural diversity. The improvement is not significant compared with structure diversity. |
> | Figure 14 | Different dimension of syntactic 2-cluster data. Only 10 datasets in training | 2 9-cluster dataset and 2 2-cluster dataset| If the dimension of training data is larger than the test data, the performance will become unstable. |
> | Figure 15 | Subsets with different domain diversity from MNIST and FMNSIT | CIFAR10 with class labels in [5, 6, 7, 8, 9] | The domain diversity does not affect the performance if the structure diversity does not change.|
> | Figure 16 | 500 subsets from MNIST and FMNSIT | Circle | AutoDV is better than t-SNE and UMAP|
> | Figure 17 | 500 subsets from MNIST and FMNSIT | Lineage| AutoDV is better than t-SNE and UMAP|
> | Figure 18 | 500 subsets from MNIST and FMNSIT | Mammoth| AutoDV meet the generalization boundary |
> | Figure 19 | 50 datasets sampled from Swiss Roll and Swiss Hole| Newly sampled Swiss Roll | AutoDV is better than UMAP and can show a roll. |
> | Figure 20 | 50 datasets sampled from Swiss Roll and Swiss Hole| Newly sampled Swiss Hole | AutoDV is better than UMAP and can show a hole. |
> | Figure 21 | 50 datasets sampled from Swiss Roll and Swiss Hole| Mammoth| AutoDV is better than UMAP and can show a shape of mammoth. |
> | Figure 22 | Syntactic 2-cluster data. Only 10 datasets in training| 1 9-cluster dataset with global noise| AutoDV is resistant to the global noise |
> | Figure 23 | Syntactic 2-cluster data. Only 10 datasets in training| 1 9-cluster dataset with small local noise | AutoDV can correctly allocate the noise |
> | Figure 24 | Syntactic 2-cluster data. Only 10 datasets in training| 1 9-cluster dataset with large local noise | AutoDV can correctly allocate the noise |
> | Figure 25-27| Syntactic 2-cluster data. Only 10 datasets in training| 2-cluster dataset with overlap structure.| AutoDV can perform better than t-SNE and UMAP |
> | Figure 28-30| Syntactic 2-cluster data. Only 10 datasets in training| 3 types of noise | AutoDV tends to form a 2-cluster structure. |
> | Figure 31-32| 50 datasets sampled from Swiss Roll supervised by PCA | Newly sampled Swiss Hole and Mammoth | AutoDV has similar performance to PCA |
> | Figure 33 | 2918 subsets from Baron Human, Campbel, and PBMC68k | 1 selected subset from Mouse Retina| AutoDV is better than default t-SNE and UMAP|
> | Figure 34-37| 363 subsets from 26 tabular datasets| 4 selected tabular dataset in the test set | AutoDV is better than default t-SNE and UMAP|

---

### Meta-Review · Area_Chair_9rMi · 2026-01-06

**Summary:**

AutoDV introduces an end-to-end Graph Transformer framework for high-dimensional data visualization, addressing the inefficiency of traditional methods like t-SNE and UMAP by meta-learning optimal embeddings to avoid per-dataset tuning. While reviewers lauded the novel shift toward an inductive "train once, infer anywhere" model, initial concerns were raised regarding generalization on non-clustered manifolds, scalability to large datasets, and over-reliance on NMI metrics. The authors responded with a comprehensive rebuttal, adding extensive experiments on synthetic manifolds, clarifying batch-based strategies for scalability, and incorporating 17 new metrics to demonstrate superior local structure preservation and noise robustness compared to default baselines.

**Reviewer Concerns:**

Concerns Successfully Addressed:

1）Generalization Capabilities (gpJd, kZK1): The authors added extensive experiments on synthetic manifold datasets (Swiss Roll, Mammoth, etc.) in Appendix Q and S. This demonstrated that while the model initially struggled with unseen topologies (like the Mammoth), it could learn them if similar structures were included in training. They also demonstrated AutoDV trained on PCA targets (Appendix R), satisfying reviewer dtQX.

2）Baselines (gpJd, dtQX): The authors provided comparisons against default t-SNE and UMAP, showing that AutoDV often outperforms them on tabular and gene datasets because it is robust to local noise (which default methods often over-interpret as structure).

3）Evaluation Metrics (kZK1, hD97): The authors expanded the evaluation to include 17 additional metrics (Appendix P), including Trustworthiness, Continuity, and KNN accuracy, proving the method preserves local structure, not just global clusters (NMI).

4）Scalability (gpJd): The authors clarified the batch-based inference strategy (Section 4.4 / Appendix E) to handle datasets larger than the 3000-sample training limit.

Outstanding Concerns:

Fundamental Problem Formulation (hD97): Reviewer hD97 viewed the work as conflating two tasks (hyperparameter search + embedding). While the authors clarified that their goal is a single inductive mapping function (bypassing the need for explicit hyperparameter search at inference), the reviewer did not have the opportunity to acknowledge this clarification.

**Reviewer Scores:**

I think all the reviewers will raise the score to possitive

---

### Decision · Program_Chairs · 2026-01-26

Accept (Poster)